# One Ring to Rule Them All: Certifiably Robust Geometric Perception with Outliers

**Heng Yang and Luca Carlone**
Laboratory for Information and Decision Systems (LIDS)
Massachusetts Institute of Technology
{hankyang,lcarlone}@mit.edu

## Abstract

We propose the first general and practical framework to design *certifiable algorithms* for robust geometric perception in the presence of a large amount of outliers. We investigate the use of a *truncated least squares* (TLS) cost function, which is known to be robust to outliers, but leads to hard, nonconvex, and nonsmooth optimization problems. Our first contribution is to show that –for a broad class of geometric perception problems– TLS estimation can be reformulated as an optimization over the ring of polynomials and *Lasserre's hierarchy of convex moment relaxations* is empirically tight at the *minimum relaxation order* (*i.e.,* certifiably obtains the *global minimum* of the nonconvex TLS problem). Our second contribution is to exploit the structural sparsity of the objective and constraint polynomials and leverage *basis reduction* to significantly reduce the size of the semidefinite program (SDP) resulting from the moment relaxation, without compromising its tightness. Our third contribution is to develop scalable *dual optimality certifiers* from the lens of *sums-of-squares* (SOS) relaxation, that can compute the suboptimality gap and possibly certify global optimality of any candidate solution (*e.g.,* returned by fast heuristics such as RANSAC or graduated non-convexity). Our dual certifiers leverage *Douglas-Rachford Splitting* to solve a convex feasibility SDP. Numerical experiments across different perception problems, including single rotation averaging, shape alignment, 3D point cloud and mesh registration, and high-integrity satellite pose estimation, demonstrate the tightness of our relaxations, the correctness of the certification, and the scalability of the proposed dual certifiers to large problems, beyond the reach of current SDP solvers.[1]

## 1 Introduction

*Geometric perception*, estimating unknown geometric models (*e.g.,* rotations, poses, 3D structure) from visual measurements (*e.g.,* images and point clouds), is a fundamental problem in computer vision, robotics, and graphics. It finds extensive applications to object detection and localization [95, 98], motion estimation and 3D reconstruction [30, 104], simultaneous localization and mapping [23, 84], shape analysis [71, 78], virtual and augmented reality [58], and medical imaging [8].

A common formulation for geometric perception resorts to optimization to perform estimation:

$$\min_{\boldsymbol{x} \in \mathcal{X}} \quad \sum_{i=1}^{N} \rho\left(r\left(\boldsymbol{x}, \boldsymbol{y}_i\right)\right), \tag{1}$$

where $\boldsymbol{y}_i \in \mathcal{Y}, i = 1, \ldots, N$, are the visual measurements, $\boldsymbol{x} \in \mathcal{X} \subseteq \mathbb{R}^n$ is the to-be-estimated geometric model, $r : \mathcal{X} \times \mathcal{Y} \to \mathbb{R}_+$ is the *residual function* that quantifies the disagreement between each measurement $\boldsymbol{y}_i$ and the geometric model $\boldsymbol{x}$, and $\rho : \mathbb{R}_+ \to \mathbb{R}_+$ is the *cost function* that determines how residuals are penalized. When the distribution of the measurement noise is known, maximum likelihood estimation provides a systematic way to design $\rho$; for instance, assuming

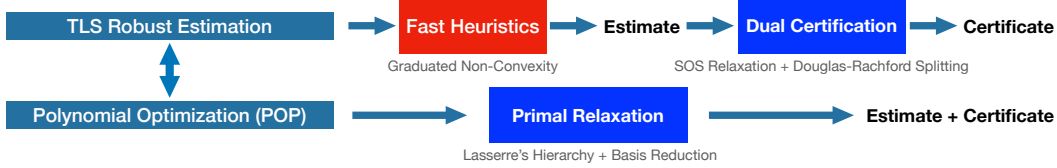

Figure 1: A general and practical framework for certifiably robust geometric perception with outliers.

Gaussian noise leads to the popular *least squares* cost function $\rho(r) = r^2$ [51, 81, 45]. However, in practice, *a large amount of* measurements, called *outliers*, depart from the assumed noise distribution (*e.g.,* due to sensor failure or incorrect data association). Therefore, a *robust* cost function, such as the $\ell_1$-norm [91], Huber [52], Geman-McClure [99], and truncated least squares [96], is necessary to prevent the outliers from corrupting the estimate. Both the constraints –defining the domain $\mathcal{X}$– and the objective function in (1) are typically nonconvex in geometric perception problems.

Solving geometric perception with *optimality guarantees* is of paramount importance for safety-critical and high-integrity applications such as autonomous driving and space robotics. Indeed, suboptimal solutions of (1) typically correspond to poor or outlier-contaminated estimates [99]. However, obtaining globally optimal solutions, particularly in the presence of outliers, remains a challenging task. Related work is divided into (i) *fast heuristics, e.g.,* RANSAC [37] and graduated non-convexity (GNC) [99], that are efficient but brittle against high outlier rates and offer no optimality guarantees, and (ii) *global solvers, e.g.,* Branch and Bound [54, 100], that guarantee optimality but run in worst-case exponential time. Recently, *certifiable algorithms* [9, 98, 19, 25] are rising as a new paradigm for solving geometric perception with both *a posteriori* optimality guarantees and polynomial-time complexity. A popular framework for constructing a certifiable algorithm requires (i) a *tight convex relaxation* of problem (1); (ii) a *fast heuristics* that computes a candidate solution to problem (1) with high probability of success; and (iii) a fast *duality-based certifier* that verifies if the candidate solution returned by the heuristics is globally optimal for the relaxation.[2] However, although a growing body of tight convex relaxations have been discovered for various instances of geometric perception *without* outliers [57, 20, 21, 84, 36, 106, 97, 82, 71, 27, 4, 39, 3, 42, 48, 94], only a few (problem-specific) tight relaxations exist for *outlier-robust* geometric perception [95, 96, 91, 62, 24].

**Contributions.** We contribute the first general and practical framework for designing certifiable algorithms for robust geometric perception with outliers (Fig. 1). Our first contribution is to show that common geometric perception problems with the truncated least squares (TLS) cost function can be reformulated as an optimization over the ring of polynomials, and *Lasserre's hierarchy of moment relaxations* [64, 65] is tight at the *minimum relaxation order*, despite the strong non-convexity and non-smoothness of the problem. Our second contribution is to propose a *basis reduction* technique, that exploits the structural *sparsity* of the polynomials and significantly reduces the size of the semidefinite programs (SDP) resulting from moment relaxation, while surprisingly maintaining tightness of the relaxation. These two contributions lead to the first set of *certifiably robust* solvers for a broad class of geometric perception problems. While scaling better than the standard moment relaxation, these solvers still rely on existing SDP solvers, whose runtime restricts their use to small-scale problems (*e.g., $N = 20$*). Therefore, our third contribution is to study the *dual* sums-of-squares (SOS) relaxation and design fast *dual optimality certifiers* that scale to realistic problem sizes (*e.g., $N = 100$*). Our certifiers leverage *Douglas–Rachford Splitting* (DRS), initialized by solving an SOS program with *correlative sparsity* [89, 90], to compute a *suboptimality* gap for any candidate solution, and possibly certify *global optimality* when the suboptimality is zero. Dual certifiers enhance existing heuristics (*e.g.,* RANSAC and GNC) with a fast certification that asserts the quality of their estimates and rejects failure cases, thus enhancing trustworthiness in safety-critical applications. We demonstrate our tight relaxations and fast certifiers on several perception problems including single rotation averaging [44, 66], image-based pose estimation (also called shape alignment) [99], point cloud registration [95], mesh registration [20], and in a satellite pose estimation application [28].

**Notation.** Let $\mathbb{R}[\boldsymbol{x}]$ be the ring of real-valued multivariate polynomials in $\{x_i\}_{i=1}^n$. Using standard notation [65], we denote every $f \in \mathbb{R}[\boldsymbol{x}]$ as $f = \sum_{\boldsymbol{\alpha} \in \mathcal{F}} c(\boldsymbol{\alpha}) \boldsymbol{x}^{\boldsymbol{\alpha}}$, where $\mathcal{F} \subseteq \mathbb{Z}_+^n$ is a finite set of nonnegative integer exponents, $c(\boldsymbol{\alpha})$ are real coefficients, and $\boldsymbol{x}^{\boldsymbol{\alpha}} \doteq x_1^{\alpha_1} x_2^{\alpha_2} \cdots x_n^{\alpha_n}$ are standard monomials. The degree of a monomial $\boldsymbol{x}^{\boldsymbol{\alpha}}$ is $\deg(\boldsymbol{x}^{\boldsymbol{\alpha}}) \doteq \sum_{i=1}^n \alpha_i$, and the degree of a polynomial $f$ is $\deg(f) = \max\{\deg(\boldsymbol{x}^{\boldsymbol{\alpha}}) : \boldsymbol{\alpha} \in \mathcal{F}\}$. We use $(\boldsymbol{x})_d$ (resp. $[\boldsymbol{x}]_d$) to denote the set of monomials

with degree $d$ (resp. with degree up to $d$). We use $m_n(d) \doteq \binom{n+d}{d}$ to denote the dimension of $[\boldsymbol{x}]_d$. Similarly, we use $[\boldsymbol{x}]_{\mathcal{F}} \doteq \{\boldsymbol{x}^{\boldsymbol{\alpha}} : \boldsymbol{\alpha} \in \mathcal{F}\}$ to denote the set of monomials with exponents in $\mathcal{F}$, and we use $m(\mathcal{F})$ to denote its dimension. We use $\mathcal{S}^n$ to denote the set of $n \times n$ symmetric matrices, and $\mathcal{S}_+^n$ for the set of symmetric positive semidefinite (PSD) matrices. We also write $\boldsymbol{A} \succeq 0$ to indicate $\boldsymbol{A} \in \mathcal{S}_+^n$. For $\boldsymbol{A} \in \mathcal{S}^n$ we use svec $(\boldsymbol{A})$ to denote its symmetric vectorization [87]. A polynomial $q \in \mathbb{R}[\boldsymbol{x}]$ is a sums-of-squares (SOS) polynomial if and only if $q$ can be written as $q = [\boldsymbol{x}]_{\mathcal{F}}^\mathsf{T} \boldsymbol{Q} [\boldsymbol{x}]_{\mathcal{F}}$ for some monomial basis $[\boldsymbol{x}]_{\mathcal{F}}$ and PSD matrix $\boldsymbol{Q} \succeq 0$, in which case $q \geq 0, \forall \boldsymbol{x} \in \mathbb{R}^n$.

## 2 Related Work

**Outlier-free Geometric Perception** algorithms can be divided into *minimal solvers* and *non-minimal solvers*. Minimal solvers assume *noiseless* measurements (*i.e.,* $r(\boldsymbol{x}, \boldsymbol{y}_i) = 0, \forall\ i$ in (1)) and use the minimum number of measurements necessary to estimate $\boldsymbol{x}$, which leads to solving a system of polynomial equations [79, 60, 40, 76]. Non-minimal solvers account for measurement noise and estimate $\boldsymbol{x}$ via nonlinear least squares (NLS), *i.e.,* $\rho(r) = r^2$ in (1). While in rare cases NLS can be solved in closed form [69, 70, 51, 7] or by solving the polynomial equations arising from the first-order optimality conditions [93, 59, 108], in general they lead to nonconvex problems and are attacked using local solvers [61, 2] or exponential-time methods (*e.g., Branch and Bound* [77, 46]).

*Certifiable algorithms* for outlier-free perception have recently emerged as an approach to compute globally optimal NLS solutions in polynomial time. These algorithms relax the NLS minimization into a convex optimization, using Shor's semidefinite relaxation for *quadratically constrained quadratic programs* [43, 68] or Lasserre's hierarchy of moment relaxations for *polynomial optimizations* [65]. By solving the SDP resulting from the convex relaxations, certifiable algorithms compute global solutions to NLS problems and provide a certificate of optimality, which usually depends on the rank of the SDP solution or the duality gap. Empirically tight convex relaxations have been discovered in pose graph optimization [25, 84], rotation averaging [36, 38], triangulation [4], 3D registration [20, 71, 27], absolute pose estimation [3], relative pose estimation [21, 106], hand-eye calibration [48] and 3D shape reconstruction from 2D landmarks [97]. More recently, theoretical analysis of when and why the relaxations are tight is also emerging [4, 36, 84, 31, 105, 27, 35, 53]. Tight relaxations also enable optimality certification (*i.e.,* checking if a given solution is optimal), which –in outlier-free perception– can be sometimes performed in closed form [25, 36, 41, 19, 22, 83, 32, 53]. Despite being certifiably optimal, these solvers assume all measurements are inliers (*i.e.,* have small noise), which rarely occurs in practice, and hence give poor estimates even in the presence of a single outlier. In stark contrast, this paper develops certifiable algorithms in the presence of large amounts of outliers.

**Robust Geometric Perception** algorithms can be divided into *fast heuristics* and *globally optimal solvers*. Two general frameworks for designing fast heuristics are RANSAC [37] and *graduated non-convexity* (GNC) [99, 14, 5]. RANSAC robustifies minimal solvers and acts as a fast heuristics to solve *consensus maximization* [29, 88], while GNC robustifies non-minimal solvers and acts as a fast heuristics to solve *M-estimation* (*i.e.,* using a robust cost function $\rho$ in (1)) [17]. Local optimization is also a popular and fast heuristics [26, 47, 85, 18, 1, 34] for the case where an initial guess is available. On the other hand, globally optimal solvers are typically designed using Branch and Bound [12, 80, 54, 56, 101], or boost robustness via a preliminary outlier-pruning scheme [95, 80].

*Certifiably robust algorithms* relax problem (1) with a robust cost into a tight convex optimization. While certain robust costs, such as the $\ell_1$-norm [91] and Huber [24], are already convex, they have low breakdown points (*i.e.,* they can be compromised by a single outlier) [103, 72]. A few problem-specific certifiably robust algorithms have been proposed to deal with high-breakdown-point formulations, such as the TLS cost [96, 95, 16, 62]. Even optimality certification becomes harder and problem-specific in the presence of outliers, due to the lack of a closed-form characterization of the dual variables [98]. In this paper, we introduce a *general* framework to design certifiably robust algorithms and optimality certifiers for a broad class of geometric perception problems with TLS cost.

## 3 Robust Geometric Perception as Polynomial Optimization

In this paper we develop certifiable algorithms to solve (1) for the case when the cost $\rho$ is a *truncated least squares* (TLS) cost:

$$f^\star = \min_{\boldsymbol{x} \in \mathcal{X}} \ \sum_{i=1}^N \min\left\{ \frac{r^2(\boldsymbol{x}, \boldsymbol{y}_i)}{\beta_i^2}, \bar{c}^2 \right\}, \tag{TLS}$$

where $\min\{\cdot, \cdot\}$ denotes the minimum between two scalars, $\beta_i$ is a known constant that can be used to model the inlier standard deviation (potentially different for each measurement $i$), and $\bar{c}$ is the maximum admissible residual for a measurement to be considered an inlier. Intuitively, problem (TLS) implements a nonlinear least squares where measurements with large residuals (*i.e.,* outliers) do not influence the estimate (*i.e.,* lead to a constant cost of $\bar{c}^2$). Problem (TLS) is known to be robust to large amounts of outliers [102, 72]. However, its global minimum $f^\star$ is hard to compute due to the non-convexity and non-smoothness of the cost (which adds to the typical non-convexity of the domain $\mathcal{X}$). In the following, we briefly review a few instantiations of robust geometric perception.

**Example 1 (Single Rotation Averaging [44])** *Given $N$ measurements of an unknown 3D rotation: $\boldsymbol{R}_i, i = 1, \dots, N$, single rotation averaging seeks to find the best* average *rotation $\boldsymbol{R}$. In this case, $\boldsymbol{x} = \boldsymbol{R} \in \mathrm{SO}(3)$, $\boldsymbol{y}_i = \boldsymbol{R}_i$, and the residual function can be chosen as $r(\boldsymbol{x}, \boldsymbol{y}_i) = \|\boldsymbol{R} - \boldsymbol{R}_i\|_F$ (the chordal distance between two rotations [44]), where $\|\cdot\|_F$ denotes the Frobenius norm.*

**Example 2 (Shape Alignment [99])** *Given a set of 3D points $\boldsymbol{B}_i \in \mathbb{R}^3$ and a set of 2D pixels $\boldsymbol{b}_i \in \mathbb{R}^2$ ($i = 1, \dots, N$), with putative correspondences $\boldsymbol{b}_i \leftrightarrow \boldsymbol{B}_i$, shape alignment seeks to find the best scale $s \in [0, \bar{s}]$ (where $\bar{s}$ is a given upper bound for the scale) and 3D rotation $\boldsymbol{R} \in \mathrm{SO}(3)$ of the point set, such that the 3D points project onto the corresponding pixels. In this case, $\boldsymbol{x} = (\boldsymbol{R}, s)$, $\boldsymbol{y}_i = (\boldsymbol{B}_i, \boldsymbol{b}_i)$ and the residual function is the reprojection error under the weak perspective camera model: $r(\boldsymbol{x}, \boldsymbol{y}_i) = \|\boldsymbol{b}_i - s\Pi\boldsymbol{R}\boldsymbol{B}_i\|$, where $\Pi = [1, 0, 0; 0, 1, 0] \in \mathbb{R}^{2 \times 3}$.*

**Example 3 (Point Cloud Registration [95])** *Given two sets of 3D points $\boldsymbol{a}_i, \boldsymbol{b}_i \in \mathbb{R}^3, i = 1, \dots, N$, with putative correspondences $\boldsymbol{a}_i \leftrightarrow \boldsymbol{b}_i$, point cloud registration seeks the best 3D rotation $\boldsymbol{R} \in \mathrm{SO}(3)$ and translation $\boldsymbol{t} \in \mathbb{R}^3$ to align them.[3] In this case, $\boldsymbol{x} = (\boldsymbol{R}, \boldsymbol{t})$, $\boldsymbol{y}_i = (\boldsymbol{a}_i, \boldsymbol{b}_i)$ and the residual function is the Euclidean distance between registered pairs of points: $r(\boldsymbol{x}, \boldsymbol{y}_i) = \|\boldsymbol{b}_i - \boldsymbol{R}\boldsymbol{a}_i - \boldsymbol{t}\|$.*

**Example 4 (Mesh Registration [20])** *Consider a 3D mesh $\{\boldsymbol{a}_i, \boldsymbol{u}_i\}_{i=1}^N$ and a 3D point cloud with estimated normals $\{\boldsymbol{b}_i, \boldsymbol{v}_i\}_{i=1}^N$, where $\boldsymbol{a}_i \in \mathbb{R}^3$ is an arbitrary point on a face of the mesh, and $\boldsymbol{u}_i$ is the unit normal of the same face, while $\boldsymbol{b}_i \in \mathbb{R}^3$ is a 3D point and $\boldsymbol{v}_i$ is the estimated unit normal at $\boldsymbol{b}_i$. Given putative correspondences $(\boldsymbol{a}_i, \boldsymbol{u}_i) \leftrightarrow (\boldsymbol{b}_i, \boldsymbol{v}_i)$, mesh registration seeks the best 3D rotation $\boldsymbol{R} \in \mathrm{SO}(3)$ and translation $\boldsymbol{t} \in \mathbb{R}^3$ to align the mesh with the point cloud.[3] In this case, $\boldsymbol{x} = (\boldsymbol{R}, \boldsymbol{t})$, $\boldsymbol{y}_i = (\boldsymbol{a}_i, \boldsymbol{u}_i, \boldsymbol{b}_i, \boldsymbol{v}_i)$, and the residual function is the weighted sum of the point-to-plane distance and normal-to-normal distance: $r^2(\boldsymbol{x}, \boldsymbol{y}_i) = \|(\boldsymbol{R}\boldsymbol{u}_i)^\mathsf{T}(\boldsymbol{b}_i - \boldsymbol{R}\boldsymbol{a}_i - \boldsymbol{t})\|^2 + w_i\|\boldsymbol{v}_i - \boldsymbol{R}\boldsymbol{u}_i\|^2$, where $w_i > 0$ is the relative weight between normal-to-normal distance and point-to-plane distance.*

The following proposition states that all the four examples above lead to (TLS) problems that can be cast as polynomial optimization problems (POPs).

**Proposition 5 (Geometric Perception as POP)** *Robust geometric perception (TLS), with residual functions as in Examples 1-4, is equivalent to the following polynomial optimization (POP):*

$$f^\star = \min_{\boldsymbol{p} \in \mathbb{R}^{\tilde{n}}} \quad f(\boldsymbol{p}) \tag{2}$$
$$\text{s.t.} \quad h_j(\boldsymbol{p}) = 0, j = 1, \dots, l_h,$$
$$1 \geq g_k(\boldsymbol{p}) \geq 0, k = 1, \dots, l_g,$$

*with $\tilde{n} \doteq n + N$, and $\boldsymbol{p} \doteq [\boldsymbol{x}^\mathsf{T}, \boldsymbol{\theta}^\mathsf{T}]^\mathsf{T} \in \mathbb{R}^{\tilde{n}}$, where $\boldsymbol{x} \in \mathcal{X}$ contains the to-be-estimated geometric model, and the vector of binary variables $\boldsymbol{\theta} \in \{\pm 1\}^N$ is such that $\theta_i = +1$ (resp. $\theta_i = -1$) when the $i$-th measurement $\boldsymbol{y}_i$ is estimated to be an inlier (resp. outlier). In this POP, $f$ is a polynomial in $\boldsymbol{p}$ with $\deg(f) \leq 3$, while $h_j, g_k$ are quadratic (degree-2) polynomials in $\boldsymbol{p}$ that are used to define the domains $\mathcal{X}$ and $\{\pm 1\}^N$. The polynomials $f, h_j, g_k$ possess the following structural properties:*

*(i) (objective function sparsity) $f$ can be written as a sum of $N$ polynomials $f_i, i = 1, \dots, N$, and each $f_i$ is a polynomial in $\boldsymbol{x}$ and $\theta_i$ of degree lower or equal to 3, i.e., $f = \sum_{i=1}^N f_i, f_i \in \mathbb{R}[\boldsymbol{x}, \theta_i], \deg(f_i) \leq 3$;*

*(ii) (constraints sparsity) let $\boldsymbol{h} \doteq \{h_j\}_{j=1}^{l_h}$ and $\boldsymbol{g} \doteq \{g_k\}_{k=1}^{l_g}$. Then, $\boldsymbol{g} \subset \mathbb{R}[\boldsymbol{x}]$ are polynomials in $\boldsymbol{x}$ (i.e., do not depend on $\boldsymbol{\theta}$). Moreover, $\boldsymbol{h}$ can be partitioned into $N + 1$ disjoint subsets: $\boldsymbol{h} = \boldsymbol{h}^\theta \cup \boldsymbol{h}^x$, with $\boldsymbol{h}^\theta = \cup_{i=1}^N \boldsymbol{h}^{\theta_i}$, where $\boldsymbol{h}^{\theta_i} \subset \mathbb{R}[\theta_i]$ are polynomials in $\theta_i$ (i.e., do not depend on $\boldsymbol{x}$ and $\theta_j, \forall j \neq i$), $\boldsymbol{h}^x \subset \mathbb{R}[\boldsymbol{x}]$ are polynomials in $\boldsymbol{x}$ (i.e., do not depend on $\boldsymbol{\theta}$);*

*(iii) (Archimedeanness) the feasible set $\mathcal{P}$ of the POP* (2) *is* Archimedean.[4]

The Supplementary Material provides a proof of Proposition 5 and the expressions of $f, h_j, g_k$ for Examples 1-4. Proposition 5 is based on three insights. First, each inner minimization $\min\{a, b\}$ $(a, b \in \mathbb{R})$ can be written as $\min_{\theta \in \{\pm 1\}} \frac{1+\theta}{2}a + \frac{1-\theta}{2}b$, which gives rise to the binary variables and leads to the objective sparsity in (i). Second, the constraint sets of $\boldsymbol{x}$ and each $\theta_i$ are mutually independent, and can be described by quadratic equality and inequality constraints, leading to the constraints sparsity in (ii). Third, the unknown variables, including $\boldsymbol{R} \in \mathrm{SO}(3), s \in [0, \bar{s}], \|\boldsymbol{t}\| \leq T$, and $\theta_i \in \{\pm 1\}$, live in compact domains described by polynomials, leading to the Archimedeanness property (iii).

# 4   The Primal View: Tight Moment Relaxation

In this section, we develop dense (Section 4.1) and sparse (Section 4.2) convex moment relaxations to the POP (2). The dense relaxation is a standard application of Lasserre's hierarchy [64, 65], while the sparse relaxation is based on a basis reduction that leverages the structural properties in Proposition 5.

## 4.1   Lasserre's Hierarchy

The following theorem describes Lasserre's hierarchy of dense moment relaxations for the POP (2).

**Theorem 6 (Dense Moment Relaxation [65])** *The dense moment relaxation at order $\kappa$ ($\geq 2$) for the POP* (2) *is the following SDP:*

$$p_\kappa^\star = \min_{\boldsymbol{z}_{2\kappa} \in \mathbb{R}^{m_{\tilde{n}}(2\kappa)}} \sum_{\boldsymbol{\alpha} \in \mathcal{F}} c(\boldsymbol{\alpha}) z_{\boldsymbol{\alpha}} \tag{3}$$

$$s.t. \quad z_{\boldsymbol{0}} = 1, \boldsymbol{M}_\kappa(\boldsymbol{z}_{2\kappa}) \succeq 0,$$
$$\boldsymbol{M}_{\kappa-1}(h_j \boldsymbol{z}_{2\kappa-2}) = \boldsymbol{0}, j = 1, \ldots, l_h,$$
$$\boldsymbol{M}_{\kappa-1}(g_k \boldsymbol{z}_{2\kappa-2}) \succeq 0, k = 1, \ldots, l_g.$$

*where $\boldsymbol{z}_{2\kappa} = \{z_{\boldsymbol{\alpha}}\} \in \mathbb{R}^{m_{\tilde{n}}(2\kappa)}$ is the vector of* moments *up to degree $2\kappa$, $c(\boldsymbol{\alpha})$ are the real coefficients of the objective function $f(\boldsymbol{p})$ corresponding to monomials $\boldsymbol{p}^{\boldsymbol{\alpha}}$ in* (2), $\boldsymbol{M}_\kappa(\boldsymbol{z}_{2\kappa}) \in \mathcal{S}^{m_{\tilde{n}}(\kappa)}$ *is the* moment matrix, *and $\boldsymbol{M}_{\kappa-1}(h_j \boldsymbol{z}_{2\kappa-2}), \boldsymbol{M}_{\kappa-1}(g_k \boldsymbol{z}_{2\kappa-2}) \in \mathcal{S}^{m_{\tilde{n}}(\kappa-1)}$ are the* localizing matrices.[5] *Let $\boldsymbol{z}_{2\kappa}^\star$ be the optimal solution of* (3)*, then the following holds true:*

*(i) (lower bound) $p_\kappa^\star$ is a lower bound for $f^\star$, i.e., $p_\kappa^\star \leq f^\star, \forall \kappa \geq 2$;*

*(ii) (finite convergence) $p_{\kappa_1}^\star \leq p_{\kappa_2}^\star$ for any $\kappa_1 \leq \kappa_2$, and $p_\kappa^\star = f^\star$ at some finite $\kappa$;*

*(iii) (optimality certificate) if $\mathrm{rank}\left(\boldsymbol{M}_\kappa(\boldsymbol{z}_{2\kappa}^\star)\right) = 1$, then $\boldsymbol{z}_\kappa^\star = [\boldsymbol{p}^\star]_\kappa$, where $\boldsymbol{p}^\star$ is the unique global minimizer of the POP* (2)*, and the relaxation is said to be* tight*;*

*(iv) (rounding and duality gap) if $\mathrm{rank}\left(\boldsymbol{M}_\kappa(\boldsymbol{z}_{2\kappa}^\star)\right) > 1$, let $\hat{\boldsymbol{p}}$ be a rounded estimate computed from a rank-1 approximation of $\boldsymbol{M}_\kappa(\boldsymbol{z}_{2\kappa}^\star)$,[5] and denote $\hat{f} = f(\hat{\boldsymbol{p}})$. Then, $p_\kappa^\star \leq f^\star \leq \hat{f}$ and we say that the relative duality gap is $\eta_\kappa = (\hat{f} - p_\kappa^\star)/\hat{f}$.*

Theorem 6 is a standard application of Lasserre's hierarchy [64] and the finite convergence result [75] to problem (2). Although Lasserre's hierarchy is guaranteed to be tight at some finite $\kappa$, the relaxation becomes computationally impractical for large $\kappa$. Therefore, it is desirable to obtain tight relaxations with small $\kappa$. In the Supplementary Material, we show that the dense moment relaxation is empirically tight at the *minimum* relaxation order $\kappa = 2$ for Examples 1-4, despite the fact that the POPs have both binary variables (a notoriously challenging setup [63]) and non-convex constraints $\boldsymbol{R} \in \mathrm{SO}(3)$.

## 4.2   Basis Reduction

Although the dense relaxation is tight at $\kappa = 2$, the size of the SDP (3) (*i.e.,* the size of the moment matrix $\boldsymbol{M}_\kappa(\boldsymbol{z}_{2\kappa})$ for $\kappa = 2$) is $\binom{n+N+2}{2}$, which grows *quadratically* in the number of measurements $N$ and quickly becomes intractable even for small $N$ (*e.g.,* $N = 20$). In this section, we exploit the

*monomial sparsity* of the POP (2) and use basis reduction to construct a sparse moment relaxation whose size grows linearly with $N$.

**Theorem 7 (Sparse Moment Relaxation)** *Define $[\boldsymbol{p}]_{\mathcal{B}} \doteq [1, \boldsymbol{x}^\mathsf{T}, \boldsymbol{\theta}^\mathsf{T}, (\boldsymbol{x})_2^\mathsf{T}, \boldsymbol{\theta}^\mathsf{T} \otimes \boldsymbol{x}^\mathsf{T}]^\mathsf{T}$ to be a reduced set of monomials, with $\mathcal{B}$ being the set of monomial exponents in $[\boldsymbol{p}]_{\mathcal{B}}$, i.e., $\mathcal{B} \doteq \{\boldsymbol{\alpha} \in \mathbb{Z}_+^{\tilde{n}} : \boldsymbol{p}^{\boldsymbol{\alpha}} \in [\boldsymbol{p}]_{\mathcal{B}}\}$. Similarly, define $[\boldsymbol{p}]_{\mathcal{B}_x} \doteq [1, \boldsymbol{x}^\mathsf{T}]^\mathsf{T}$ and let $\mathcal{B}_x$ be its set of exponents. Let $2\mathcal{B} \doteq \{\boldsymbol{\alpha} \in \mathbb{Z}_+^{\tilde{n}} : \boldsymbol{\alpha} = \boldsymbol{\alpha}_1 + \boldsymbol{\alpha}_2, \boldsymbol{\alpha}_1, \boldsymbol{\alpha}_2 \in \mathcal{B}\}$ (resp. $2\mathcal{B}_x$) be the Minkowski sum of $\mathcal{B}$ (resp. $\mathcal{B}_x$) with itself. Define $\boldsymbol{z}_{2\mathcal{B}} \in \mathbb{R}^{m(2\mathcal{B})}$ (resp. $\boldsymbol{z}_{2\mathcal{B}_x} \in \mathbb{R}^{m(2\mathcal{B}_x)}$) to be the vector of moments for all monomials in $[\boldsymbol{p}]_{2\mathcal{B}}$ (resp. $[\boldsymbol{p}]_{2\mathcal{B}_x}$), and $\boldsymbol{M}_{\mathcal{B}}(\boldsymbol{z}_{2\mathcal{B}}) \in \mathcal{S}^{m(\mathcal{B})}$ (resp. $\boldsymbol{M}_{\mathcal{B}_x}(\boldsymbol{z}_{2\mathcal{B}_x}) \in \mathcal{S}^{m(\mathcal{B}_x)}$) to be the moment matrix that assembles $\boldsymbol{z}_{2\mathcal{B}}$ (resp. $\boldsymbol{z}_{2\mathcal{B}_x}$) in rows and columns indexed by $[\boldsymbol{p}]_{\mathcal{B}}$ (resp. $[\boldsymbol{p}]_{\mathcal{B}_x}$). Then, the sparse moment relaxation is:*

$$p_{\mathcal{B}}^\star = \min_{\boldsymbol{z}_{2\mathcal{B}} \in \mathbb{R}^{m(2\mathcal{B})}} \sum_{\boldsymbol{\alpha} \in \mathcal{F}} c(\boldsymbol{\alpha}) z_{\boldsymbol{\alpha}} \tag{4}$$

$$\text{s.t.} \quad z_{\boldsymbol{0}} = 1, \boldsymbol{M}_{\mathcal{B}}(\boldsymbol{z}_{2\mathcal{B}}) \succeq 0,$$

$$\boldsymbol{M}_1(h\boldsymbol{z}_2) = \boldsymbol{0}, \forall h \in \boldsymbol{h}^x; \quad \boldsymbol{M}_{\mathcal{B}_x}(h\boldsymbol{z}_{2\mathcal{B}_x}) = \boldsymbol{0}, \forall h \in \boldsymbol{h}^\theta,$$

$$\boldsymbol{M}_1(g\boldsymbol{z}_2) \succeq 0, \forall g \in \boldsymbol{g},$$

*where $\boldsymbol{h}^x, \boldsymbol{h}^\theta, \boldsymbol{g}$ are defined as in Proposition 5. Moreover, we have $p_{\mathcal{B}}^\star \leq p_2^\star \leq f^\star$ and properties (iii)-(iv) in Theorem 6 hold for the sparse relaxation (4).*

The key idea behind Theorem 7 is to reduce the size of the SDP by only considering the reduced monomial basis $[\boldsymbol{p}]_{\mathcal{B}}$, which essentially removes all the monomials of the form $\theta_i \theta_j$ that do not appear in $f$ as per property (i) in Proposition 5. The size of the SDP (4) (*i.e.*, the size of $\boldsymbol{M}_{\mathcal{B}}(z_{2\mathcal{B}})$) is $m(\mathcal{B}) = \frac{(n+1)(n+2)}{2} + (1+n)N$, which grows linearly in $N$. In Section 6, we show that the sparse moment relaxation (4) is also tight, even in the presence of a large amount of outliers. Although there exist other efficient sparse variants [89, 92] of Lasserre's hierarchy that exploit correlative sparsity, in the Supplementary Material we show they break the tightness at the minimum relaxation order and produce poor estimates. Nevertheless, they can be used to bootstrap our dual certifiers (Section 5.2).

# 5 The Dual View: Fast Optimality Certification

Despite scaling linearly in $N$, the sparse relaxation (4) is still too large to be solved efficiently using current interior point methods (IPM) [74] when $N > 20$. On the other hand, fast heuristics such as graduated non-convexity [99] can compute globally optimal solutions to the POP (2) with high probability of success. In this section, we show that, by taking the dual perspective of sums-of-squares (SOS) relaxations, we can develop efficient *certifiers* to verify the optimality of a candidate solution $(\hat{\boldsymbol{p}}, \hat{f})$ for large $N$ (*e.g.*, $N = 100$), for which the SDP relaxation (4) is not even implementable.

## 5.1 Sums-of-Squares Relaxation

A candidate solution $(\hat{\boldsymbol{p}}, \hat{f})$ is globally optimal for the POP (2) if and only if $f(\boldsymbol{p}) - \hat{f} \geq 0, \forall \boldsymbol{p} \in \mathcal{P}$. However, testing *nonnegativity* of a polynomial on a constraint set is NP-hard [15], so instead we test if the polynomial is SOS on the constraint set and provide a sufficient condition for global optimality.

**Theorem 8 (Sufficient Condition for Global Optimality)** *Given any candidate solution $(\hat{\boldsymbol{p}}, \hat{f})$ to the POP (2), if the following optimization is* feasible *(i.e., has at least one solution):*

$$\text{find} \quad \boldsymbol{\lambda}_j^x \in \mathbb{R}^{m_{\tilde{n}}(2)}, \boldsymbol{\lambda}_j^\theta \in \mathbb{R}^{m_n(2)}, \boldsymbol{S}_0 \in \mathcal{S}_+^{m(\mathcal{B})}, \boldsymbol{S}_k \in \mathcal{S}_+^{m_{\tilde{n}}(1)} \tag{5}$$

$$\text{s.t.} \; f(\boldsymbol{p}) - \hat{f} - \sum_{h_j \in \boldsymbol{h}^x} h_j \left( [\boldsymbol{p}]_2^\mathsf{T} \boldsymbol{\lambda}_j^x \right) - \sum_{h_j \in \boldsymbol{h}^\theta} h_j \left( [\boldsymbol{x}]_2^\mathsf{T} \boldsymbol{\lambda}_j^\theta \right) = [\boldsymbol{p}]_{\mathcal{B}}^\mathsf{T} \boldsymbol{S}_0 \, [\boldsymbol{p}]_{\mathcal{B}} + \sum_{k=1}^{l_g} g_k \left( [\boldsymbol{p}]_1^\mathsf{T} \boldsymbol{S}_k \, [\boldsymbol{p}]_1 \right), \forall \boldsymbol{p}, \tag{6}$$

*then $\hat{f}$ (resp. $\hat{\boldsymbol{p}}$) is the global minimum (resp. global minimizer) of the POP (2). Moreover, problem (5) can be written compactly as a feasibility SDP:*

$$\text{find} \quad \boldsymbol{d}, \quad \text{s.t.} \quad \boldsymbol{d} \in \mathcal{K} \cap \mathcal{A}, \tag{7}$$

*where $\boldsymbol{d} = [(\boldsymbol{\lambda}_1^x)^\mathsf{T}, \ldots, (\boldsymbol{\lambda}_{|\boldsymbol{h}^x|}^x)^\mathsf{T}, (\boldsymbol{\lambda}_1^\theta)^\mathsf{T}, \ldots, (\boldsymbol{\lambda}_{|\boldsymbol{h}^\theta|}^\theta)^\mathsf{T}, svec\,(\boldsymbol{S}_1)^\mathsf{T}, \ldots, svec\,(\boldsymbol{S}_{l_g})^\mathsf{T}, svec\,(\boldsymbol{S}_0)^\mathsf{T}]^\mathsf{T}$ concatenates all variables in (5), $\mathcal{K}$ defines a convex cone, and $\mathcal{A} \doteq \{\boldsymbol{d} : \boldsymbol{Ad} = \boldsymbol{b}\}$ defines an affine subspace, where $\boldsymbol{b}$ is a vector and $\boldsymbol{A}$ is a matrix satisfying the* partial orthogonality *property [107, 13].*

In the Supplementary Material, we provide a proof of Theorem 8. Intuitively, if problem (5) is feasible, then for any $\boldsymbol{p} \in \mathcal{P}$, the left-hand side of (6) reduces to $f(\boldsymbol{p}) - \hat{f}$ (due to $h_j = 0$) and the right-hand side of (6) is nonnegative (due to $g_k \geq 0$, $\boldsymbol{S}_0, \boldsymbol{S}_k \succeq 0$), producing a certificate that $f(\boldsymbol{p}) \geq \hat{f}$. The SOS relaxation (5) also uses basis reduction and it is the dual of the sparse moment relaxation (4) [65] with the constraint that $\hat{f}$ is the global optimum. In SDP (7), the convex cone $\mathcal{K}$ corresponds to the PSD constraints in (5) and the affine subspace $\mathcal{A}$ corresponds to matching coefficients in the equality constraint (6). The partial orthogonality of $\boldsymbol{A}$ is a property for SDPs resulting from SOS relaxations and allows efficient *projection* onto the affine subspace $\mathcal{A}$ [107, 13].

## 5.2 Douglas-Rachford Splitting

In this section, we propose a first-order method based on *Douglas-Rachford Splitting* (DRS) [33, 55] to solve (7) at scale. DRS iteratively solves (7) by starting at an arbitrary initial point $\boldsymbol{d}_0$, and performing the following three-step updates (at each iteration $\tau \geq 0$):

$$(i)\; \boldsymbol{d}_\tau^{\mathcal{K}} = \mathrm{proj}_{\mathcal{K}}\left(\boldsymbol{d}_\tau\right), \quad (ii)\; \boldsymbol{d}_\tau^{\mathcal{A}} = \mathrm{proj}_{\mathcal{A}}\left(2\boldsymbol{d}_\tau^{\mathcal{K}} - \boldsymbol{d}_\tau\right), \quad (iii)\; \boldsymbol{d}_{\tau+1} = \boldsymbol{d}_\tau + \gamma_\tau\left(\boldsymbol{d}_\tau^{\mathcal{A}} - \boldsymbol{d}_\tau^{\mathcal{K}}\right), \quad (8)$$

where $\mathrm{proj}_{\mathcal{K}}$ (resp. $\mathrm{proj}_{\mathcal{A}}$) denotes the orthogonal projection onto $\mathcal{K}$ (resp. $\mathcal{A}$) and $\gamma_\tau$ is a parameter of the algorithm. The rationale behind the use of DRS to solve the feasibility SDP (7) is that, although finding $\boldsymbol{d} \in \mathcal{K} \cap \mathcal{A}$ is expensive (requires solving a large-scale SDP), finding $\boldsymbol{d} \in \mathcal{K}$ and $\boldsymbol{d} \in \mathcal{A}$ separately (*i.e.,* projecting onto $\mathcal{K}$ and $\mathcal{A}$ separately) is computationally inexpensive [49, 10, 50]. The following result shows how to certify optimality using the DRS iterations (8).

**Theorem 9** (DRS **for Optimality Certification**) *Consider the* DRS *iterations* (8). *Then the following properties hold true: (i) If the SDP* (7) *is feasible, then the sequence* $\{\boldsymbol{d}_\tau\}_{\tau \geq 0}$ *in* (8) *converges to a solution of* (7) *when* $0 < \gamma_\tau < 2$; *(ii) Let* $\varepsilon = (\hat{f} - f^\star)/\hat{f}$ *be the relative suboptimality between* $\hat{f}$ *and the global minimum* $f^\star$ *of the POP* (2)*, then each* DRS *iteration* (8) *gives a valid suboptimality upper bound* $\bar{\varepsilon}_\tau$, *i.e.,* $\varepsilon \leq \bar{\varepsilon}_\tau$, *and* $\bar{\varepsilon}_\tau$ *can be efficiently computed from* $\boldsymbol{d}_\tau^{\mathcal{A}}$.

A complete proof of Theorem 9 is given in the Supplementary Material. The intuition behind Theorem 9(i) is that, by using the two projections alternatively (thus, the name "*splitting*"), the DRS iterates (8) converge to a solution in $\mathcal{K} \cap \mathcal{A}$ if the intersection is nonempty. Moreover, even if the intersection is empty (*e.g.,* when $\hat{f}$ is not the global minimum), Theorem 9(ii) states that each DRS iteration is still able to *assess* the suboptimality of $\hat{f}$, which enables the *dual certifiers* to detect wrong candidate solutions (*cf.* Section 6). DRS converges faster than the vanilla *alternating projections to convex sets* used in [98] (*cf.* [11]). Moreover, we further boost convergence speed by initializing DRS with an initial point $\boldsymbol{d}_0$ computed by solving an inexpensive SOS program with *chordal sparsity* [89, 65] (see the Supplementary Material for implementation details).

# 6 Experiments

This section shows that (i) the sparse moment relaxation (4) is tight and can be used to solve small problems (*e.g.,* $N = 20$); (ii) our *dual optimality certifiers* are effective and scale to larger problems (*e.g.,* $N = 100$); (iii) our algorithms allow solving realistic satellite pose estimation problems.

**Implementation.** We model the sparse moment relaxation (4) using YALMIP [67] in Matlab and solve the resulting SDPs using MOSEK [6]. DRS is implemented in Matlab using $\gamma_\tau = 2$.[6]

**Setup.** We test primal relaxation and dual certification on random problem instances of Examples 1-4: single rotation averaging (SRA), shape alignment (SA), point cloud registration (PCR), and mesh registration (MR). At each Monte Carlo run, we randomly sample a ground truth model $\boldsymbol{x}$ and generate inliers by perturbing the measurements with Gaussian noise with standard deviation $\sigma$. We choose $\sigma = 3°$ in SRA, and $\sigma = 0.01$ in SA, PCR, and MR. Outliers are generated as arbitrary rotations or vectors (independent on $\boldsymbol{x}$). The relative weight between point-to-plane distance and normal-to-normal distance in MR is set to $w_i = 1, i = 1, \ldots, N$. The threshold in problem (TLS) is set to $\bar{c} = 1$ for all applications, and $\beta_i, i = 1, \ldots, N$, is set to be proportional to the inlier noise. The interested reader can find more details about the setup in the Supplementary Material.

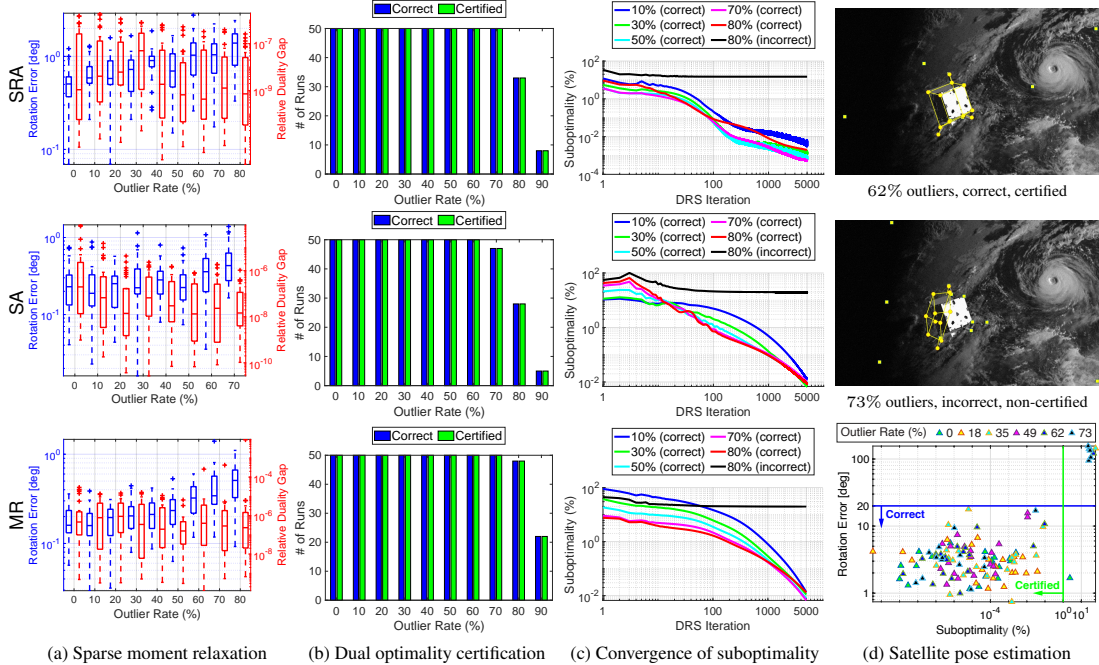

| (a) Sparse moment relaxation | (b) Dual optimality certification | (c) Convergence of suboptimality | (d) Satellite pose estimation |

Figure 2: Performance of certifiable algorithms. (a) Rotation estimation error (left axis) and relative duality gap (right axis) of the sparse moment relaxation (4). (b) Number of runs when the solution of GNC is correct (*i.e.,* rotation error less than $5°$) and number of runs when the solution of GNC is certified (*i.e.,* suboptimality below $1\%$). (c) Suboptimality gap versus DRS iterations, averaged over correct and incorrect runs. Top row: single rotation averaging (SRA), middle row: shape alignment (SA), bottom row: mesh registration (MR). (d) Qualitative and quantitative results for satellite pose estimation on the SPEED dataset [86].

**Primal Relaxation.** We first evaluate the performance of the sparse moment relaxation (4) under increasing outlier rates, with $N = 20$ measurements. Fig. 2(a) shows the boxplots of rotation estimation errors and relative duality gap for SRA (top), SA (middle), and MR (bottom) averaged over 30 Monte Carlo runs (results for PCR are qualitatively similar to MR and hence postponed to the Supplementary Material). The sparse moment relaxation is numerically tight (relative gap smaller than $10^{-3}$), with a single instance exhibiting a large gap (mesh registration, $80\%$ outliers). The figure also shows that the relaxation produces an accurate estimate in all tested instances. In the Supplementary Material, we show our primal relaxation is tight even under *adversarial outliers*.

**Dual Certification.** We test our dual optimality certifiers under increasing outlier rates, with $N = 100$ measurements. In each Monte Carlo run, we first use GNC [99] as a fast heuristics to compute a candidate solution to the POP (2), and then run the proposed dual certifiers (Theorem 9) to compute a suboptimality gap. Fig. 2(b) plots the number of runs when GNC returns the *correct* solutions (*i.e.,* with rotation error less than $5°$), and the number of runs when the solutions are *certified* (*i.e.,* have suboptimality below $1\%$). We can see that our dual certifiers can certify all correct solutions and reject all incorrect estimates (the blue and green bars always have same height, meaning that there are no false positives nor false negatives). Fig. 2(c) plots the average convergence history of the suboptimality gap versus the number of DRS iterations (in log-log scale). DRS drives the suboptimality below $1\%$ within 1000 iterations (within 100 iterations for SRA) if the solution is correct, while it reports a suboptimality larger than $10\%$ if the solution is incorrect. In the Supplementary Material, we show our certification outperforms the statistical Kolmogorov–Smirnov test [73].

**Which One is More Scalable?** Table 1 compares the scalability of the sparse relaxation and the dual certification for increasing number of measurements. Solving the large-scale SDP quickly becomes intractable for moderate $N$, while certification using DRS can scale to large number of measurements.

**Satellite Pose Estimation.** Satellite pose estimation using monocular vision is a crucial technology for many space operations [86, 28]. We use "Shape Alignment (Example 2)" to perform 6D pose estimation from satellite images in the SPEED dataset [86] (see Fig. 2(d)). Towards this goal, we first use the pre-trained network from [28] to detect 11 pixel measurements corresponding to 3D keypoints

| $N$ | SRA | | | SA | | | PCR | | | MR | | |
|---|---|---|---|---|---|---|---|---|---|---|---|---|
| | $m(\mathcal{B})$ | $t_{\text{relax}}$ | $t_{\text{certify}}$ | $m(\mathcal{B})$ | $t_{\text{relax}}$ | $t_{\text{certify}}$ | $m(\mathcal{B})$ | $t_{\text{relax}}$ | $t_{\text{certify}}$ | $m(\mathcal{B})$ | $t_{\text{relax}}$ | $t_{\text{certify}}$ |
| 20 | 255 | 151.65 | 0.73 | 168 | 35.52 | 2.00 | 351 | 763.59 | 16.34 | 351 | 750.67 | 8.43 |
| 50 | 555 | 38866 | 2.67 | 378 | 3287 | 7.88 | 741 | $>10^5$ | 84.86 | 741 | $>10^5$ | 60.76 |
| 100 | 1055 | ** | 8.35 | 728 | $>10^5$ | 37.44 | 1391 | ** | 357.48 | 1391 | ** | 165.87 |

Table 1: Relaxation and certification time (in seconds) for increasing $N$. $t_{\text{relax}}$ is the time for solving the sparse moment relaxation (4). $t_{\text{certify}}$ includes the time for computing the chordal initial guess and the time for DRS to drive the suboptimality below $1\%$. "$**$" denotes instances where MOSEK ran out of memory.

of the Tango satellite model. Because the network outputs fairly accurate detections (all inliers), we also replace $0\%, 18\%, 35\%, 49\%, 62\%,$ and $73\%$ *pairwise* inliers (see Supplementary Material) with random outliers to test more challenging instances. We show a correct and certified estimation with $62\%$ outliers in Fig. 2(d) top panel, and an incorrect and non-certified estimation with $73\%$ outliers in Fig. 2(d) middle panel. Fig. 2(d) bottom panel plots the statistics of the rotation error over 20 satellite images (showing the relation between suboptimality and estimation errors). We refer the reader to the Supplementary Material for a more comprehensive description of the tests and the results.

## 7    Conclusions

We have proposed a general framework for designing certifiable algorithms for a broad class of robust geometric perception problems. From the primal perspective, we apply Lasserre's hierarchy of moment relaxations, together with basis reduction, to construct tight semidefinite relaxations to nonconvex robust estimation problems. From the dual perspective, we use SOS relaxation to convert the certification of a given candidate solution to a convex feasibility SDP, and then we leverage Douglas-Rachford Splitting to solve the feasibility SDP and compute a suboptimality for the candidate solution. Our primal relaxation is tight, and our dual certification is correct and scalable.

## Broader Impact

Geometric perception plays an essential role in robotics applications ranging from autonomous driving, robotic manipulation, autonomous flight, to robotic search and rescue. Occasional perception failures are almost inevitable while operating in the wild (*e.g.,* due to sensor malfunction, or incorrect data association resulting from neural networks or hand-crafted feature matching techniques). These failures, if not detected and handled properly, have detrimental effects, especially in safety-critical and high-integrity applications (*e.g.,* they may put passengers at risk in autonomous driving or damage a satellite in space applications). Existing perception algorithms (*e.g.,* fast heuristics) can fail without notice. On the contrary, the certifiable algorithms presented in this work perform geometric perception with optimality guarantees and act as safeguards to detect perception failures. The development of certifiable algorithms has the potential to enhance the robustness of perception systems, inform system certification and monitoring, and increase the trustworthiness of autonomous systems.

On the negative side, the use of certifiable algorithms as an enabler for robots and autonomous systems inherits the shortcomings connected to the misuse of these technologies. The use of autonomous systems in military applications as well as the impact of robotics and automation on the (human) workforce must be carefully pondered to ensure a positive societal impact.

## Acknowledgments

The authors would like to thank Jie Wang, Victor Magron, and Jean B. Lasserre for the discussion about Lasserre's hierarchy of moment and SOS relaxations; Alp Yurtsever, Suvrit Sra, and Yang Zheng for the discussion about using first-order methods to solve large-scale SDPs; Bo Chen and Tat-Jun Chin for kindly sharing the data for satellite pose estimation; and the anonymous reviews.

This work was partially funded by ARL DCIST CRA W911NF-17-2-0181, ONR RAIDER N00014-18-1-2828, and the Lincoln Laboratory "Resilient Perception in Degraded Environments" program.

## Footnotes

[1]Code available at https://github.com/MIT-SPARK/CertifiablyRobustPerception.

[2]Global optimality of the relaxation implies global optimality of problem (1) when the relaxation is tight.

[3] For mathematical convenience, we assume the translation is bounded by a known value $T$, *i.e.,* $\|\boldsymbol{t}\| \leq T$.

[4]Archimedeanness is a stronger condition than compactness, see [15, Definition 3.137, p. 115].

[5]We refer the non-expert reader to [65] for a comprehensive introduction to moment relaxations, and provide extra definitions and accessible examples in the Supplementary Material.

[6]The limiting case of $\gamma_\tau = 2$ for DRS is commonly referred to as the *Peaceman-Rachford Splitting* (PRS) [33]. Although theoretically PRS could diverge, we found it worked well for all our applications.

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
