[Supplementary Material]

# One Ring to Rule Them All: Certifiably Robust Geometric Perception with Outliers

## *Supplementary Material*

**Heng Yang and Luca Carlone**
Laboratory for Information and Decision Systems (LIDS)
Massachusetts Institute of Technology
{hankyang,lcarlone}@mit.edu

**Notation.** Besides the notation already defined in Section 1 of the main text, we define the following extra notation. A polynomial $q \in \mathbb{R}[\boldsymbol{x}]$ is a sums-of-squares (SOS) polynomial if and only if $q$ can be written as $q = [\boldsymbol{x}]_{\mathcal{F}}^{\mathsf{T}} \boldsymbol{Q} [\boldsymbol{x}]_{\mathcal{F}}$ for some monomial basis $[\boldsymbol{x}]_{\mathcal{F}}$ and PSD matrix $\boldsymbol{Q} \succeq 0$, in which case $q \geq 0, \forall \boldsymbol{x} \in \mathbb{R}^n$. We use $\Sigma [\boldsymbol{x}]_{2\mathcal{F}}$ to denote the set of SOS polynomials parametrized by the monomial basis $[\boldsymbol{x}]_{\mathcal{F}}$. In particular, when $[\boldsymbol{x}]_{\mathcal{F}} = [\boldsymbol{x}]_d$ is the full standard monomial basis of degree up to $d$, we use $\Sigma [\boldsymbol{x}]_{2d}$ to denote the set of SOS polynomials with degree up to $2d$. Moreover, $\Sigma [\boldsymbol{x}] \subset \mathbb{R}[\boldsymbol{x}]$ is the set of all SOS polynomials (with arbitrary degrees). For a constraint set $\mathcal{X}$ defined by polynomial equality and inequality constraints $\mathcal{X} \doteq \{\boldsymbol{x} : h_j(\boldsymbol{x}) = 0, j = 1, \dots, l_h; g_k(\boldsymbol{x}) \geq 0, k = 1, \dots, l_g\}$, the set $\mathcal{X}$ is said to be Archimedean if there exist $M > 0, \lambda_j \in \mathbb{R}[\boldsymbol{x}], j = 1, \dots, l_h$, and $s_k \in \Sigma [\boldsymbol{x}], k = 1, \dots, l_g$, such that $M - \|\boldsymbol{x}\|^2 = \sum_{j=1}^{l_h} \lambda_j h_j + \sum_{k=1}^{l_g} s_k g_k$, which immediately implies that $\|\boldsymbol{x}\|^2 \leq M$ and the set $\mathcal{X}$ is compact [3, Definition 3.137, p. 115].

## A1    Proof of Proposition 5 (Geometric Perception as POP)

*Proof.* To tackle the non-smoothness of the inner minimization "$\min\{\cdot, \cdot\}$" in problem (TLS), we first reformulate problem (TLS) as:

$$f^{\star} = \min_{\substack{\boldsymbol{x} \in \mathcal{X} \\ \theta_i \in \{\pm 1\}, i=1,\dots,N}} \sum_{i=1}^{N} \frac{1+\theta_i}{2\beta_i^2} r^2(\boldsymbol{x}, \boldsymbol{y}_i) + \frac{1-\theta_i}{2} \bar{c}^2, \qquad (A1)$$

where we have used the fact that "$\min\{a, b\}$" is equivalent to an optimization over a binary variable: $\min\{a, b\} = \min_{\theta \in \{\pm 1\}} \frac{1+\theta}{2} a + \frac{1-\theta}{2} b$ (where $\theta = +1$ when $a < b$ and $\theta = -1$ when $a > b$). Intuitively, if the $i$-th measurement $\boldsymbol{y}_i$ is an inlier (*i.e.*, $r^2 \leq \bar{c}^2 \beta_i^2$), then $\theta_i = +1$ and the corresponding term in (A1) reduces to least squares; if $\boldsymbol{y}_i$ is an outlier (*i.e.*, $r^2 > \bar{c}^2 \beta_i^2$), then $\theta_i = -1$ and the corresponding term in (A1) becomes a constant $\bar{c}^2$, whence the outlier is irrelevant to the optimization. Since we have introduced $N$ binary variables to the optimization (A1), we denote $\boldsymbol{p} = \left[\boldsymbol{x}^{\mathsf{T}}, \boldsymbol{\theta}^{\mathsf{T}}\right]^{\mathsf{T}} \in \mathbb{R}^{\tilde{n}}$ as the new set of variables, where $\boldsymbol{\theta} \doteq [\theta_1, \dots, \theta_N]^{\mathsf{T}} \in \{\pm 1\}^N$ is the vector of binary variables and $\tilde{n} \doteq n + N$ is the number of variables. Then we make two immediate observations: (1) denote $f_i(\boldsymbol{p}) = \frac{1+\theta_i}{2\beta_i^2} r^2(\boldsymbol{x}, \boldsymbol{y}_i) + \frac{1-\theta_i}{2} \bar{c}^2$, then $f_i(\boldsymbol{p}) \in \mathbb{R}[\boldsymbol{x}, \theta_i]$ is only a polynomial of $\boldsymbol{x}$ and $\theta_i$ and the objective function of (A1) can be written as the finite sum of $f_i$'s: $f(\boldsymbol{p}) = \sum_{i=1}^{N} f_i(\boldsymbol{p})$ (*i.e.*, claim (i) in Proposition 5). (2) The binary constraints $\theta_i \in \{\pm 1\}, i = 1, \dots, N$ are equivalent to quadratic polynomial equality constraints $h^{\theta_i} \doteq 1 - \theta_i^2 = 0, i = 1, \dots, N$, and obviously each $h^{\theta_i} \in \mathbb{R}[\theta_i]$ is only a polynomial in $\theta_i$. For simplicity, we denote $\boldsymbol{h}^{\theta} = \{h^{\theta_i}\}_{i=1}^N$ (*i.e.*, claim (ii) in Proposition 5).

Next we will show that – for Examples 1-4– (1) $r^2(\boldsymbol{x}, \boldsymbol{y}_i)$ is a polynomial in $\boldsymbol{x}$ with $\deg\left(r^2(\boldsymbol{x}, \boldsymbol{y}_i)\right) \leq 2$ (and hence, $\deg\left(f_i(\boldsymbol{p})\right) \leq 3$), (2) the constraint set $\boldsymbol{x} \in \mathcal{X}$ can be written as quadratic polynomial

inequality and equality constraints, and (3) the feasible set is Archimedean (*i.e.,* claim (iii) in Proposition 5).

**Example 1 (Single Rotation Averaging).** We develop the residual function:

$$r^2(\boldsymbol{x}, \boldsymbol{y}_i) = \|\boldsymbol{R} - \boldsymbol{R}_i\|_F^2 = \text{tr}\left((\boldsymbol{R} - \boldsymbol{R}_i)^\mathsf{T}(\boldsymbol{R} - \boldsymbol{R}_i)\right) = \text{tr}\left(2\mathbf{I}_3\right) - 2\text{tr}\left(\boldsymbol{R}_i^\mathsf{T}\boldsymbol{R}^\mathsf{T}\right) = 6 - 2\boldsymbol{y}_i^\mathsf{T}\boldsymbol{r}, \quad \text{(A2)}$$

where we have denoted $\boldsymbol{r} \doteq \text{vec}\left(\boldsymbol{R}\right) \in \mathbb{R}^9$ as the vectorization of the unknown rotation matrix $\boldsymbol{R}$, and $\boldsymbol{y}_i \doteq \text{vec}\left(\boldsymbol{R}_i\right) \in \mathbb{R}^9$ as the vectorization of the measurements $\boldsymbol{R}_i$. From eq. (A2) it is clear that $\deg\left(r^2(\boldsymbol{x}, \boldsymbol{y}_i)\right) = 1$. The constraint set for single rotation averaging is $\boldsymbol{R} \in \text{SO}(3)$, which is known to be equivalent to a set of (redundant) quadratic polynomial equality constraints [25].

**Lemma A1 (Quadratic Constraints for** $\text{SO}(3)$ **[20, 25]).** *For any matrix* $\boldsymbol{R} \in \mathbb{R}^{3\times 3}$, $\boldsymbol{R} \in \text{SO}(3)$ *is equivalent to the following set of 15 quadratic polynomial equality constraints* $\boldsymbol{h}^r = \{h_i^r\}_{i=1}^{15}$:

$$\begin{cases} \text{Orthonormality: } h_1^r{=}1{-}\|\boldsymbol{r}_1\|^2, \ h_2^r{=}1{-}\|\boldsymbol{r}_2\|^2, \ h_3^r{=}1{-}\|\boldsymbol{r}_3\|^2, \ h_4^r{=}\boldsymbol{r}_1^\mathsf{T}\boldsymbol{r}_2, \ h_5^r{=}\boldsymbol{r}_2^\mathsf{T}\boldsymbol{r}_3, \ h_6^r{=}\boldsymbol{r}_3^\mathsf{T}\boldsymbol{r}_1 \\ \text{Right-handedness: } h_{7,8,9}^r{=}\boldsymbol{r}_1{\times}\boldsymbol{r}_2{-}\boldsymbol{r}_3, \ h_{10,11,12}^r{=}\boldsymbol{r}_2{\times}\boldsymbol{r}_3{-}\boldsymbol{r}_1, \ h_{13,14,15}^r{=}\boldsymbol{r}_3{\times}\boldsymbol{r}_1{-}\boldsymbol{r}_2 \end{cases} \quad \text{(A3)}$$

*where* $\boldsymbol{r}_i \in \mathbb{R}^3, i = 1, 2, 3$ *denotes the $i$-th column[1] of* $\boldsymbol{R}$ *and "$\times$" represents vector cross product.*

Therefore, we have $\boldsymbol{h} = \boldsymbol{h}^x \cup \boldsymbol{h}^\theta$ with $\boldsymbol{h}^x \doteq \boldsymbol{h}^r$, and $\boldsymbol{g} = \varnothing$ for single rotation averaging. To show the Archimedeanness of the feasible set $\mathcal{P} \doteq \{\boldsymbol{p} : h(\boldsymbol{p}) = 0, \forall h \in \boldsymbol{h}, 1 \geq g(\boldsymbol{p}) \geq 0, \forall g \in \boldsymbol{g}\}$, we note that:

$$3 + N - \|\boldsymbol{p}\|^2 = \sum_{i=1}^{3} 1 \cdot h_i^r + \sum_{i=1}^{N} 1 \cdot h^{\theta_i} = 0, \quad \text{(A4)}$$

which implies that $\|\boldsymbol{p}\|^2 \leq N + 3$ and the feasible set $\mathcal{P}$ is equipped with a polynomial certificate for compactness.

**Example 2 (Shape Alignment).** Directly developing the residual function $r^2(\boldsymbol{x}, \boldsymbol{y}_i) = \|\boldsymbol{b}_i - s\Pi\boldsymbol{R}\boldsymbol{B}_i\|^2$ leads to a quartic polynomial (degree 4) in $s$ and $\boldsymbol{R}$, which is not suitable for moment relaxation because it would increase the minimum relaxation order $\kappa$ [14]. Therefore, we perform a change of variables and let $\bar{\boldsymbol{R}} = s\Pi\boldsymbol{R}$:

$$\bar{\boldsymbol{R}} = s \begin{bmatrix} 1 & 0 & 0 \\ 0 & 1 & 0 \end{bmatrix} \begin{bmatrix} \boldsymbol{r}_1^\mathsf{T} \\ \boldsymbol{r}_2^\mathsf{T} \\ \boldsymbol{r}_3^\mathsf{T} \end{bmatrix} = \begin{bmatrix} s\boldsymbol{r}_1^\mathsf{T} \\ s\boldsymbol{r}_2^\mathsf{T} \end{bmatrix} \doteq \begin{bmatrix} \bar{\boldsymbol{r}}_1^\mathsf{T} \\ \bar{\boldsymbol{r}}_2^\mathsf{T} \end{bmatrix}, \quad \text{(A5)}$$

where $\boldsymbol{r}_i^\mathsf{T} \in \mathbb{R}^3$ denotes the $i$-th row of the rotation matrix $\boldsymbol{R}$ and we have denoted $\bar{\boldsymbol{r}}_i = s\boldsymbol{r}_i, i = 1, 2$ as the product of $s$ and $\boldsymbol{r}_i$. Now using Lemma A1, we can see that $s \in [0, \bar{s}]$ and $\boldsymbol{R} \in \text{SO}(3)$ is equivalent to the following constraints on $\bar{\boldsymbol{r}} \doteq \text{vec}\left(\bar{\boldsymbol{R}}^\mathsf{T}\right) = \left[\bar{\boldsymbol{r}}_1^\mathsf{T}, \bar{\boldsymbol{r}}_2^\mathsf{T}\right]^\mathsf{T}$:

$$\boldsymbol{h}^{\bar{r}} = \left\{h_1^r = \|\bar{\boldsymbol{r}}_1\|^2 - \|\bar{\boldsymbol{r}}_2\|^2, h_2^r = \bar{\boldsymbol{r}}_1^\mathsf{T}\bar{\boldsymbol{r}}_2\right\}, \quad \boldsymbol{g}^{\bar{r}} = \left\{1 - \frac{\|\bar{\boldsymbol{r}}_1\|^2 + \|\bar{\boldsymbol{r}}_2\|^2}{2\bar{s}^2}\right\}. \quad \text{(A6)}$$

Therefore, we have $\boldsymbol{h} = \boldsymbol{h}^x \cup \boldsymbol{h}^\theta$ with $\boldsymbol{h}^x \doteq \boldsymbol{h}^{\bar{r}}$, and $\boldsymbol{g} = \boldsymbol{g}^{\bar{r}}$ for shape alignment.[2] To prove the feasible set is Archimedean, we write the following polynomial certificate for compactness:

$$2\bar{s}^2 + N - \|\boldsymbol{p}\|^2 = 2\bar{s}^2 \cdot g^{\bar{r}} + \sum_{i=1}^{N} 1 \cdot h^{\theta_i} \geq 0. \quad \text{(A7)}$$

**Example 3 (Point Cloud Registration).** We develop the residual function:

$$r^2(\boldsymbol{x}, \boldsymbol{y}_i) = \|\boldsymbol{b}_i - \boldsymbol{R}\boldsymbol{a}_i - \boldsymbol{t}\|^2 = \|\boldsymbol{t}\|^2 - 2\boldsymbol{b}_i^\mathsf{T}\boldsymbol{t} - 2\boldsymbol{b}_i^\mathsf{T}\boldsymbol{R}\boldsymbol{a}_i + 2\boldsymbol{t}^\mathsf{T}\boldsymbol{R}\boldsymbol{a}_i + \|\boldsymbol{a}_i\|^2 + \|\boldsymbol{b}_i\|^2$$
$$= \|\boldsymbol{t}\|^2 - 2\boldsymbol{b}_i^\mathsf{T}\boldsymbol{t} - 2\left(\boldsymbol{a}_i^\mathsf{T} \otimes \boldsymbol{b}_i^\mathsf{T}\right)\boldsymbol{r} + 2\left(\boldsymbol{a}_i^\mathsf{T} \otimes \boldsymbol{t}^\mathsf{T}\right)\boldsymbol{r} + \|\boldsymbol{a}_i\|^2 + \|\boldsymbol{b}_i\|^2, \quad \text{(A8)}$$

where $\boldsymbol{r} \doteq \mathrm{vec}\,(\boldsymbol{R}) \in \mathbb{R}^9$ is the vectorization of $\boldsymbol{R}$ and "$\otimes$" denotes the Kronecker product. Clearly, $\deg\left(r^2(\boldsymbol{x}, \boldsymbol{y}_i)\right) = 2$ from eq. (A8). For the constraint set of $(\boldsymbol{R}, \boldsymbol{t})$, we have the 15 quadratic equality constraints from Lemma A1 for $\boldsymbol{R} \in \mathrm{SO}(3)$, and we have $g^t = 1 - \frac{\|\boldsymbol{t}\|^2}{T^2}$ for $\boldsymbol{t}$ (the translation is bounded by a known value $T$). Therefore, for point cloud registration, we have $\boldsymbol{h} = \boldsymbol{h}^x \cup \boldsymbol{h}^\theta$ with $\boldsymbol{h}^x \doteq \boldsymbol{h}^r$ and $\boldsymbol{g} = \{g^t\}$. The Archimedeanness of the constraint set can be seen from the following inequality:

$$T^2 + N - \|\boldsymbol{p}\|^2 = T^2 \cdot g^t + \sum_{i=1}^{N} 1 \cdot h^{\theta_i} \geq 0. \tag{A9}$$

**Example 4 (Mesh Registration).** To make the residual function $r^2(\boldsymbol{x}, \boldsymbol{y}_i)$ a quadratic polynomial, we perform the following change of variables and develop the residual function:

$$r^2(\boldsymbol{x}, \boldsymbol{y}_i) = \left\| (\boldsymbol{R}\boldsymbol{u}_i)^\mathsf{T} (\boldsymbol{b}_i - \boldsymbol{R}\boldsymbol{a}_i - \boldsymbol{t}) \right\|^2 + w_i \|\boldsymbol{v}_i - \boldsymbol{R}\boldsymbol{u}_i\|^2 \tag{A10}$$

$$= \left\| \boldsymbol{u}_i^\mathsf{T} \left(\boldsymbol{R}^\mathsf{T} \boldsymbol{b}_i - \boldsymbol{a}_i - \boldsymbol{R}^\mathsf{T} \boldsymbol{t}\right) \right\|^2 + w_i \|\boldsymbol{v}_i - \boldsymbol{R}\boldsymbol{u}_i\|^2 \tag{A11}$$

$$\overset{\tilde{\boldsymbol{R}} \doteq \boldsymbol{R}^T, \tilde{\boldsymbol{t}} \doteq \boldsymbol{R}^T \boldsymbol{t}}{=} \left\| \boldsymbol{u}_i^\mathsf{T} \tilde{\boldsymbol{R}} \boldsymbol{b}_i - \boldsymbol{u}_i^\mathsf{T} \boldsymbol{a}_i - \boldsymbol{u}_i^\mathsf{T} \tilde{\boldsymbol{t}} \right\|^2 + w_i \left\| \boldsymbol{v}_i - \tilde{\boldsymbol{R}}^\mathsf{T} \boldsymbol{u}_i \right\|^2 \tag{A12}$$

$$= \tilde{\boldsymbol{t}}^\mathsf{T} \left(\boldsymbol{u}_i \otimes \boldsymbol{u}_i^\mathsf{T}\right) \tilde{\boldsymbol{t}} + \tilde{\boldsymbol{r}}^\mathsf{T} \left(\boldsymbol{b}_i \boldsymbol{b}_i^\mathsf{T} \otimes \boldsymbol{u}_i \boldsymbol{u}_i^\mathsf{T}\right) \tilde{\boldsymbol{r}} - 2\mathrm{vec}\left(\boldsymbol{u}_i \boldsymbol{a}_i^\mathsf{T} \boldsymbol{u}_i \boldsymbol{b}_i^\mathsf{T}\right)^\mathsf{T} \tilde{\boldsymbol{r}} + 2\boldsymbol{u}_i^\mathsf{T} \boldsymbol{a}_i \boldsymbol{u}_i^\mathsf{T} \tilde{\boldsymbol{t}}$$
$$- 2\tilde{\boldsymbol{r}}^\mathsf{T} \left(\boldsymbol{b}_i \otimes \boldsymbol{u}_i \boldsymbol{u}_i^\mathsf{T}\right) \tilde{\boldsymbol{t}} - 2w_i \mathrm{vec}\left(\boldsymbol{u}_i \boldsymbol{v}_i^\mathsf{T}\right)^\mathsf{T} \tilde{\boldsymbol{r}} + \left(\boldsymbol{u}_i^\mathsf{T} \boldsymbol{a}_i\right)^2 + w_i \left(\|\boldsymbol{v}_i\|^2 + \|\boldsymbol{u}_i\|^2\right), \tag{A13}$$

where $\tilde{\boldsymbol{R}} \doteq \boldsymbol{R}^\mathsf{T} \in \mathrm{SO}(3)$ and $\tilde{\boldsymbol{t}} \doteq \boldsymbol{R}^\mathsf{T} \boldsymbol{t} \in \mathbb{R}^3$ is the new set of unknown rotation and translation. In addition, $\|\boldsymbol{t}\| \leq T$ if and only if $\|\tilde{\boldsymbol{t}}\| \leq T$ because the rotation matrix preserves the norm of $\boldsymbol{t}$. The original $(\boldsymbol{R}, \boldsymbol{t})$ can be recovered from $\left(\tilde{\boldsymbol{R}}, \tilde{\boldsymbol{t}}\right)$ by:

$$\boldsymbol{R} = \tilde{\boldsymbol{R}}^\mathsf{T}, \quad \boldsymbol{t} = \tilde{\boldsymbol{R}}^\mathsf{T} \tilde{\boldsymbol{t}}. \tag{A14}$$

The constraints for $\left(\tilde{\boldsymbol{R}}, \tilde{\boldsymbol{t}}\right)$ is the same as what we developed for point cloud registration: $\boldsymbol{h} = \boldsymbol{h}^x \cup \boldsymbol{h}^\theta$ with $\boldsymbol{h}^x \doteq \boldsymbol{h}^{\tilde{r}} \doteq \boldsymbol{h}^r$, and $\boldsymbol{g} \doteq \boldsymbol{g}^{\tilde{t}} \doteq \{g^t\}$. Therefore, the Archimedeanness of the feasible set follows from eq. (A9). This concludes the proof for Proposition 5. ∎

## A2 Explanation and Example for Theorem 6 (Dense Moment Relaxation)

In this section, we provide a brief but self-contained explanation to shed light on Lasserre's hierarchy of dense moment relaxations in Theorem 6 (Section A2.1). We also give an accessible example to demonstrate the application of the hierarchy to a simple but illustrative problem, namely 2D single rotation averaging (Section A2.2).

### A2.1 Explanation

Our explanation of Lasserre's hierarchy is adapted from [14, 13]. Let $\mu\,(\boldsymbol{p})$ be a probability measure supported on the feasible set $\mathcal{P}$ of the POP (2), and let $\Omega\,(\mathcal{P})$ be the set of all possible probability measures on $\mathcal{P}$. Then the POP (2) can be rewritten as a generalized moment problem.

**Theorem A2 (POP as the Moment Problem [13, Proposition 2.1]).** *Let the feasible set of the POP (2) be $\mathcal{P}$, then the POP is equivalent to the following optimization:*

$$f_\mu^\star = \min_{\mu \in \Omega(\mathcal{P})} \int f(\boldsymbol{p}) d\mu, \tag{A15}$$

*in the sense that:*

*(i) $f_\mu^\star = f^\star$;*

*(ii) if $\boldsymbol{p}^\star$ is a (potentially not unique) global minimizer of the POP (2), then $\mu^\star = \delta_{\boldsymbol{p}^\star}$ is a global minimizer of the moment problem (A15), where $\delta_{\boldsymbol{p}^\star}$ is the Dirac measure at $\boldsymbol{p}^\star$;*

*(iii)* assuming the POP (2) has a (potentially not unique) global minimizer with global minimum $f^\star$, then for every optimal solution $\mu^\star$ of the moment problem (A15), $f(\boldsymbol{p}) = f^\star$, $\mu^\star$-almost everywhere (i.e., $\mu^\star (\{\boldsymbol{p} : f(\boldsymbol{p}) \neq f^\star\}) = 0$ and $\mu^\star$ is supported only on the global minimizers of the POP);

*(iv)* if $\boldsymbol{p}^\star$ is the unique global minimizer of the POP (2), then $\mu^\star = \delta_{\boldsymbol{p}^\star}$ is the unique global minimizer of the moment problem (A15).

Although the moment problem (A15) is convex [14], it is infinite-dimensional and still intractable. Therefore, the crux of making the optimization tractable is to relax the infinite-dimensional problem into a finite-dimensional one. Towards this goal, we introduce the notion of moments, moment matrices and localizing matrices.

**Definition A3** (**Moments, Moment Matrices, Localizing Matrices [14, Chapter 3]**). *Given a probability measure $\mu$ supported on $\mathcal{P} \subset \mathbb{R}^{\tilde{n}}$, its moment of order $\boldsymbol{\alpha} \in \mathbb{Z}_+^{\tilde{n}}$ is the scalar $z_{\boldsymbol{\alpha}} \doteq \int_{\mathcal{P}} \boldsymbol{p}^{\boldsymbol{\alpha}} d\mu = \mathbb{E}_\mu [\boldsymbol{p}^{\boldsymbol{\alpha}}] \in \mathbb{R}$, i.e., the integral (expected value) of the monomial $\boldsymbol{p}^{\boldsymbol{\alpha}}$ over the set $\mathcal{P}$ w.r.t. $\mu$. In particular, if $\boldsymbol{\alpha} = \boldsymbol{0}$, then $\boldsymbol{p}^{\boldsymbol{\alpha}} = p_1^{\alpha_1} \cdots p_{\tilde{n}}^{\alpha_{\tilde{n}}} = 1$, and $z_{\boldsymbol{0}} = 1$. Now let $\boldsymbol{z} = (z_{\boldsymbol{\alpha}})$ be an infinite sequence of moments (the order $\boldsymbol{\alpha}$ can be unbounded), we define the linear functional $L_{\boldsymbol{z}} : \mathbb{R}[\boldsymbol{p}] \to \mathbb{R}$:*

$$f(\boldsymbol{p}) = \sum_{\boldsymbol{\alpha} \in \mathcal{F}} c(\boldsymbol{\alpha}) \boldsymbol{p}^{\boldsymbol{\alpha}} \mapsto L_{\boldsymbol{z}}(f) = \sum_{\boldsymbol{\alpha} \in \mathcal{F}} c(\boldsymbol{\alpha}) z_{\boldsymbol{\alpha}}, \tag{A16}$$

*that maps a polynomial $f$ to a real number $L_{\boldsymbol{z}}(f)$ by replacing the monomials of $f$ with corresponding moments. With this linear functional, the moment sequence of degree up to $2\kappa$ is simply:*

$$\boldsymbol{z}_{2\kappa} \doteq L_{\boldsymbol{z}}([\boldsymbol{p}]_{2\kappa}) \in \mathbb{R}^{m_{\tilde{n}}(2\kappa)}, \tag{A17}$$

*where the linear functional $L_{\boldsymbol{z}}$ applies component-wise to the vector of monomials $[\boldsymbol{p}]_{2\kappa}$, and the moment matrix of degree $\kappa$ is:*

$$\boldsymbol{M}_\kappa(\boldsymbol{z}_{2\kappa}) \doteq L_{\boldsymbol{z}}\left([\boldsymbol{p}]_\kappa [\boldsymbol{p}]_\kappa^\mathsf{T}\right) \in \mathcal{S}^{m_{\tilde{n}}(\kappa)}, \tag{A18}$$

*where $L_{\boldsymbol{z}}$ also applies component-wise to the monomial matrix $[\boldsymbol{p}]_\kappa [\boldsymbol{p}]_\kappa^\mathsf{T}$, and $\boldsymbol{M}_\kappa(\boldsymbol{z}_{2\kappa})$ essentially assembles the vector of moments $\boldsymbol{z}_{2\kappa}$ into a symmetric matrix. Finally, given a polynomial $h \in \mathbb{R}[\boldsymbol{p}]$, we definite the localizing matrix of order $\kappa$ with respect to $\boldsymbol{z}$ and $h$ to be:*

$$\boldsymbol{M}_\kappa(h\boldsymbol{z}_{2\kappa}) \doteq L_{\boldsymbol{z}}\left(h \cdot \left([\boldsymbol{p}]_\kappa [\boldsymbol{p}]_\kappa^\mathsf{T}\right)\right) \in \mathcal{S}^{m_{\tilde{n}}(\kappa)}, \tag{A19}$$

*where $L_{\boldsymbol{z}}$ applies component-wise, and $h \cdot \left([\boldsymbol{p}]_\kappa [\boldsymbol{p}]_\kappa^\mathsf{T}\right)$ means multiplying $h$ with each entry of the monomial matrix $[\boldsymbol{p}]_\kappa [\boldsymbol{p}]_\kappa^\mathsf{T}$.*

With this definition, we can see that the cost function of the moment problem (A15) is a linear function of the moments:

$$\int f(\boldsymbol{p}) d\mu = \int \sum_{\boldsymbol{\alpha} \in \mathcal{F}} c(\boldsymbol{\alpha}) \boldsymbol{p}^{\boldsymbol{\alpha}} d\mu = \sum_{\boldsymbol{\alpha} \in \mathcal{F}} c(\boldsymbol{\alpha}) \int \boldsymbol{p}^{\boldsymbol{\alpha}} d\mu = \sum_{\boldsymbol{\alpha} \in \mathcal{F}} c(\boldsymbol{\alpha}) z_{\boldsymbol{\alpha}}. \tag{A20}$$

Therefore, instead of finding the probability measure $\mu$ directly in the infinite-dimensional space $\Omega(\mathcal{P})$ as written in eq. (A15), we can equivalently search for the sequence of (possibly finite number of) moments $\boldsymbol{z}$ and then recover the measure $\mu$ from the moments $\boldsymbol{z}$. However, not every sequence of moments has a representing measure. In fact, in order to have a representing measure, the moment sequence has to satisfy the following conditions.

**Theorem A4** (**Necessary and Sufficient Condition for Representing Measure [14, Theorem 3.8(b), p. 63]**). *Let $\boldsymbol{z} = (z_{\boldsymbol{\alpha}})$ be a given infinite sequence of moments, and let $\mathcal{P}$ be an Archimedean constraint set defined by the polynomial equality and inequality constraints in the POP (2). Then, the sequence $\boldsymbol{z}$ has a representing measuring on $\mathcal{P}$ if and only if:*

$$\forall \kappa \in \mathbb{N}: \ \boldsymbol{M}_\kappa(\boldsymbol{z}_{2\kappa}) \succeq 0; \ \boldsymbol{M}_\kappa(h\boldsymbol{z}_{2\kappa}) = \boldsymbol{0}, \forall h \in \boldsymbol{h}; \ \boldsymbol{M}_\kappa(g\boldsymbol{z}_{2\kappa}) \succeq 0, \forall g \in \boldsymbol{g}. \tag{A21}$$

Enforcing the PSD constraints in (A21) for every $\kappa \in \mathbb{N}$ (potentially unbounded) is intractable due to the infinite moment sequence $\boldsymbol{z}$. Therefore, a natural strategy is to enforce the constraints for a fix order $\kappa$ (called the relaxation order), which is precisely the optimization (3) in Theorem 6.[3] Problem (3) is a *relaxation* of the moment problem (A15) because the constraints at a fixed $\kappa$ only provide a *necessary* condition for the existence of a representing measure, which in turn implies that the global minimum of the relaxation, $p_\kappa^\star$, is a *lower bound* of the global minimum of the moment problem (A15) (and thus the POP (2)):

$$p_\kappa^\star \leq f_\mu^\star = f^\star. \tag{A22}$$

Lasserre's hierarchy is a hierarchy of moment relaxations with increasing relaxation orders $\kappa_1 < \kappa_2 < \ldots$ (and increasing lower bounds $p_{\kappa_1}^\star \leq p_{\kappa_2}^\star < \ldots$ ) until the relaxation is tight (*i.e.,* $p_\kappa^\star = f_\mu^\star$). In general, Lasserre's hierarchy may achieve tightness only *asymptotically* (*i.e.,* $p_\kappa^\star \to f_\mu^\star$ as $\kappa \to \infty$). However, when the feasible set $\mathcal{P}$ is Archimedean, Nie [18] showed that the hierarchy terminates at a finite relaxation order, which is the case for our POP (2) arising from a broad class of geometric perception problems (*cf.* claim (iii) in Proposition 5).

Now a natural question is, how can one determine when the relaxation is tight (and terminate the hierarchy) without knowing the true global minimum $f^\star$? In other words, how to compute an *optimality certificate*, and possibly recover the global minimizers of the POP (2) from the solution of the moment relaxation (3)? Both questions boil down to checking if the solution of the moment relaxation, $\boldsymbol{z}_{2\kappa}^\star$, has a representing measure on $\mathcal{P}$, which is known as the *truncated $\mathbb{K}$-moment problem* [7]. The following theorem states a sufficient condition.

**Theorem A5 (Sufficient Condition for Truncated $\mathbb{K}$-Moment Problem [14, Theorem 3.11, p. 66]).** *Let $\mathcal{P}$ be the feasible set of the POP (2), where both $h_j$ and $g_k$ are quadratic polynomials. Let $\boldsymbol{z}_{2\kappa}^\star$ be the solution of the moment relaxation (3). Then $\boldsymbol{z}_{2\kappa}^\star$ admits an $r$-atomic representing measure supported on $\mathcal{P}$, with $r = \mathrm{rank}\left(\boldsymbol{M}_{\kappa-1}\left(\boldsymbol{z}_{2\kappa-2}^\star\right)\right)$, if:*

$$\mathrm{rank}\left(\boldsymbol{M}_{\kappa-1}\left(\boldsymbol{z}_{2\kappa-2}^\star\right)\right) = \mathrm{rank}\left(\boldsymbol{M}_\kappa\left(\boldsymbol{z}_{2\kappa}^\star\right)\right). \tag{A23}$$

Theorem A5 is a special case of Theorem 3.11 in [14], where we have used the fact that $\mathcal{P}$ are defined by quadratic polynomials.[4] In particular, for the POP arising from geometric perception problems, at the minimum relaxation order $\kappa = 2$, we usually have $\mathrm{rank}\left(\boldsymbol{M}_\kappa\left(\boldsymbol{z}_{2\kappa}^\star\right)\right) = 1$, which immediately implies that $r = \mathrm{rank}\left(\boldsymbol{M}_{\kappa-1}\left(\boldsymbol{z}_{2\kappa-2}^\star\right)\right) = \mathrm{rank}\left(\boldsymbol{M}_\kappa\left(\boldsymbol{z}_{2\kappa}^\star\right)\right) = 1$ (because $\boldsymbol{M}_{\kappa-1}\left(\boldsymbol{z}_{2\kappa-2}^\star\right)$ is a nonzero principal sub-matrix of $\boldsymbol{M}_\kappa\left(\boldsymbol{z}_{2\kappa}^\star\right)$), and $\boldsymbol{z}_{2\kappa}^\star$ admits a 1-atomic representing measure $\mu = \delta_{\boldsymbol{p}^\star}$. Therefore, from Theorem A2, we have $\boldsymbol{p}^\star$ is the unique global minimizer of the POP (2).[5] Additionally, for $\mu = \delta_{\boldsymbol{p}^\star}$, it is straightforward to verify that $\boldsymbol{z}_{2\kappa}^\star = [\boldsymbol{p}^\star]_{2\kappa}$ from eq. (A17), and $\boldsymbol{p}^\star$ can be directly read off from the moments.

**Rounding and Relative Duality Gap**. When the sufficient condition eq. (A23) does not hold and the moment matrix $\boldsymbol{M}_\kappa\left(\boldsymbol{z}_{2\kappa}^\star\right)$ has rank larger than 1, we can first perform spectral decomposition on $\boldsymbol{M}_\kappa\left(\boldsymbol{z}_{2\kappa}^\star\right)$, and extract its eigenvector corresponding to the largest eigenvalue, denoted as $\boldsymbol{v}_\kappa \in \mathbb{R}^{m_{\tilde{n}}(\kappa)}$. Then we normalize $\boldsymbol{v}_\kappa$'s first entry, $\boldsymbol{v}_\kappa(1)$, to be 1 by: $\boldsymbol{v}_\kappa \leftarrow \frac{\boldsymbol{v}_\kappa}{\boldsymbol{v}_\kappa(1)}$. If the relaxation were tight, then $\boldsymbol{v}_\kappa = [\boldsymbol{p}^\star]_\kappa$ is a vector of moments up to degree $\kappa$ and $\boldsymbol{p}^\star$ can be directly read off from $\boldsymbol{v}_\kappa$. However, since the relaxation is not tight, we obtain a feasible point $\hat{\boldsymbol{p}}$ by:

$$\hat{\boldsymbol{p}} = \mathrm{proj}_\mathcal{P}\left(\boldsymbol{v}_\kappa\left(\boldsymbol{p}\right)\right), \tag{A24}$$

where $\boldsymbol{v}_\kappa\left(\boldsymbol{p}\right)$ denotes the entries of $\boldsymbol{v}_\kappa$ at indices corresponding to the locations of $\boldsymbol{p}$ in $[\boldsymbol{p}]_\kappa$, and $\mathrm{proj}_\mathcal{P}$ performs the projection onto the feasible set $\mathcal{P}$ (see Section A2.3 for details). Let $\hat{f} = f(\hat{\boldsymbol{p}})$, we have the following inequality:

$$p_\kappa^\star \leq f^\star \leq \hat{f}, \tag{A25}$$

where the first inequality follows from eq. (A22), and the second inequality holds because $f^\star$ is the global minimum of the POP (2). The relative duality gap can be computed as:

$$\eta_\kappa = \frac{\hat{f} - p_\kappa^\star}{\hat{f}}. \tag{A26}$$

A smaller $\eta_\kappa$ implies a tighter relaxation and $\eta_\kappa = 0$ if and only if the relaxation is tight.

## A2.2 Example: 2D Single Rotation Averaging

To make our explanation of Lasserre's hierarchy in Section A2.1 more accessible, we show an application of the hierarchy to 2D single rotation averaging, because the dimension of $x$ is small and the constraint set $\mathcal{X}$ is simple to characterize. 2D single rotation averaging follows the definition of 3D single rotation averaging in Example 1, except that the measurements $R_i, i = 1, \ldots, N$ and the unknown geometric model $R$ are 2D rotation matrices, *i.e.,* $R \in \mathrm{SO}(2)$. In this case, we describe a 2D rotation matrix using:

$$R \in \mathrm{SO}(2) \iff R = \begin{bmatrix} x_1 & -x_2 \\ x_2 & x_1 \end{bmatrix}, \quad \text{s.t.} \quad h^x = 1 - x_1^2 - x_2^2 = 0. \tag{A27}$$

We then choose $N = 2$, leading to two binary variables $\theta_1$ and $\theta_2$. Denote $x = [x_1, x_2]^\mathsf{T}$, $\theta = [\theta_1, \theta_2]^\mathsf{T}$, and $p = [x^\mathsf{T}, \theta^\mathsf{T}]^\mathsf{T}$, the POP (2) for 2D single rotation averaging with $N = 2$ is:

$$\min_{p=[x_1,x_2,\theta_1,\theta_2]^\mathsf{T} \in \mathbb{R}^4} f(p) \tag{A28}$$

$$\text{s.t.} \quad h^x = 1 - x_1^2 - x_2^2 = 0, \tag{A29}$$

$$h^{\theta_1} = 1 - \theta_1^2 = 0, \tag{A30}$$

$$h^{\theta_2} = 1 - \theta_2^2 = 0, \tag{A31}$$

where the objective function can be computed from eq. (A1).

**Moment matrices**. To describe the dense moment relaxation (3) at $\kappa = 2$, we first form the moment matrices $M_1(z_2)$ and $M_2(z_4)$. Towards this goal, let us write the vector of monomials $[p]_1$ and $[p]_2$:

$$[p]_1 = [1, x_1, x_2, \theta_1, \theta_2]^\mathsf{T} \in \mathbb{R}^5, \tag{A32}$$

$$[p]_2 = [1, x_1, x_2, \theta_1, \theta_2, x_1^2, x_1x_2, x_1\theta_1, x_1\theta_2, x_2^2, x_2\theta_1, x_2\theta_2, \theta_1^2, \theta_1\theta_2, \theta_2^2]^\mathsf{T} \in \mathbb{R}^{15}. \tag{A33}$$

For notation simplicity, we use $z_{p^\alpha}$, instead of $z_\alpha$, to denote the moment of order $\alpha$. For example, $z_{x_1x_2} \doteq \int_{\mathcal{P}} x_1x_2 d\mu$ for some probability measure $\mu$ supported on $\mathcal{P}$. Then the vector of moments $z_1$ and $z_2$ directly follow from the vector of monomials in eq. (A32) and (A33):

$$z_1 = [z_1, z_{x_1}, z_{x_2}, z_{\theta_1}, z_{\theta_2}]^\mathsf{T} \in \mathbb{R}^5, \tag{A34}$$

$$z_2 = \left[z_1, z_{x_1}, z_{x_2}, z_{\theta_1}, z_{\theta_2}, z_{x_1^2}, z_{x_1x_2}, z_{x_1\theta_1}, z_{x_1\theta_2}, z_{x_2^2}, z_{x_2\theta_1}, z_{x_2\theta_2}, z_{\theta_1^2}, z_{\theta_1\theta_2}, z_{\theta_2^2}\right] \in \mathbb{R}^{15}. \tag{A35}$$

The vector of moments of degree up to 4, $z_4 \in \mathbb{R}^{m_4(4)=70}$ can be written in a similar way. We omit its full expression here, because it will soon appear in the moment matrix $M_2(z_4)$. Then we are ready to form the moment matrix of order 1, with rows and columns indexed by $[p]_1$:

$$M_1(z_2) = L_z\left([p]_1[p]_1^\mathsf{T}\right) = \begin{array}{c} \\ 1 \\ x_1 \\ x_2 \\ \theta_1 \\ \theta_2 \end{array} \begin{array}{c} 1 \quad\; x_1 \quad\;\; x_2 \quad\;\; \theta_1 \quad\;\; \theta_2 \\ \begin{bmatrix} z_1 & z_{x_1} & z_{x_2} & z_{\theta_1} & z_{\theta_2} \\ \star & z_{x_1^2} & z_{x_1x_2} & z_{x_1\theta_1} & z_{x_1\theta_2} \\ \star & \star & z_{x_2^2} & z_{x_2\theta_1} & z_{x_2\theta_2} \\ \star & \star & \star & z_{\theta_1^2} & z_{\theta_1\theta_2} \\ \star & \star & \star & \star & z_{\theta_2^2} \end{bmatrix} \end{array}, \tag{A36}$$

where we see that the moments appearing in $M_1(z_2)$ are exactly $z_2$ (compare upper triangular entries in (A36) with (A35), thus the expression $M_1(z_2)$). Similarly, we form the moment matrix of order 2, with rows and columns indexed by $[p]_2$:

$$\boldsymbol{M}_2\left(\boldsymbol{z}_4\right) = L_{\boldsymbol{z}}\left([\boldsymbol{p}]_2\,[\boldsymbol{p}]_2^\top\right) = \tag{A37}$$

$$
\begin{array}{c|ccccccccccccccc}
 & 1 & x_1 & x_2 & \theta_1 & \theta_2 & x_1^2 & x_1x_2 & x_1\theta_1 & x_1\theta_2 & x_2^2 & x_2\theta_1 & x_2\theta_2 & \theta_1^2 & \theta_1\theta_2 & \theta_2^2 \\
\hline
1 & z_1 & z_{x_1} & z_{x_2} & z_{\theta_1} & z_{\theta_2} & z_{x_1^2} & z_{x_1x_2} & z_{x_1\theta_1} & z_{x_1\theta_2} & z_{x_2^2} & z_{x_2\theta_1} & z_{x_2\theta_2} & z_{\theta_1^2} & z_{\theta_1\theta_2} & z_{\theta_2^2} \\
x_1 & \star & z_{x_1^2} & z_{x_1x_2} & z_{x_1\theta_1} & z_{x_1\theta_2} & z_{x_1^3} & z_{x_1^2x_2} & z_{x_1^2\theta_1} & z_{x_1^2\theta_2} & z_{x_1x_2^2} & z_{x_1x_2\theta_1} & z_{x_1x_2\theta_2} & z_{x_1\theta_1^2} & z_{x_1\theta_1\theta_2} & z_{x_1\theta_2^2} \\
x_2 & \star & \star & z_{x_2^2} & z_{x_2\theta_1} & z_{x_2\theta_2} & z_{x_1^2x_2} & z_{x_1x_2^2} & z_{x_1x_2\theta_1} & z_{x_1x_2\theta_2} & z_{x_2^3} & z_{x_2^2\theta_1} & z_{x_2^2\theta_2} & z_{x_2\theta_1^2} & z_{x_2\theta_1\theta_2} & z_{x_2\theta_2^2} \\
\theta_1 & \star & \star & \star & z_{\theta_1^2} & z_{\theta_1\theta_2} & z_{x_1^2\theta_1} & z_{x_1x_2\theta_1} & z_{x_1\theta_1^2} & z_{x_1\theta_1\theta_2} & z_{x_2^2\theta_1} & z_{x_2\theta_1^2} & z_{x_2\theta_1\theta_2} & z_{\theta_1^3} & z_{\theta_1^2\theta_2} & z_{\theta_1\theta_2^2} \\
\theta_2 & \star & \star & \star & \star & z_{\theta_2^2} & z_{x_1^2\theta_2} & z_{x_1x_2\theta_2} & z_{x_1\theta_1\theta_2} & z_{x_1\theta_2^2} & z_{x_2^2\theta_2} & z_{x_2\theta_1\theta_2} & z_{x_2\theta_2^2} & z_{\theta_1^2\theta_2} & z_{\theta_1\theta_2^2} & z_{\theta_2^3} \\
x_1^2 & \star & \star & \star & \star & \star & z_{x_1^4} & z_{x_1^3x_2} & z_{x_1^3\theta_1} & z_{x_1^3\theta_2} & z_{x_1^2x_2^2} & z_{x_1^2x_2\theta_1} & z_{x_1^2x_2\theta_2} & z_{x_1^2\theta_1^2} & z_{x_1^2\theta_1\theta_2} & z_{x_1^2\theta_2^2} \\
x_1x_2 & \star & \star & \star & \star & \star & \star & z_{x_1^2x_2^2} & z_{x_1^2x_2\theta_1} & z_{x_1^2x_2\theta_2} & z_{x_1x_2^3} & z_{x_1x_2^2\theta_1} & z_{x_1x_2^2\theta_2} & z_{x_1x_2\theta_1^2} & {\color{blue}z_{x_1x_2\theta_1\theta_2}} & z_{x_1x_2\theta_2^2} \\
x_1\theta_1 & \star & \star & \star & \star & \star & \star & \star & z_{x_1^2\theta_1^2} & z_{x_1^2\theta_1\theta_2} & z_{x_1x_2^2\theta_1} & z_{x_1x_2\theta_1^2} & {\color{blue}z_{x_1x_2\theta_1\theta_2}} & z_{x_1\theta_1^3} & z_{x_1\theta_1^2\theta_2} & z_{x_1\theta_1\theta_2^2} \\
x_1\theta_2 & \star & \star & \star & \star & \star & \star & \star & \star & z_{x_1^2\theta_2^2} & z_{x_1x_2^2\theta_2} & {\color{blue}z_{x_1x_2\theta_1\theta_2}} & z_{x_1x_2\theta_2^2} & z_{x_1\theta_1^2\theta_2} & z_{x_1\theta_1\theta_2^2} & z_{x_1\theta_2^3} \\
x_2^2 & \star & \star & \star & \star & \star & \star & \star & \star & \star & z_{x_2^4} & z_{x_2^3\theta_1} & z_{x_2^3\theta_2} & z_{x_2^2\theta_1^2} & z_{x_2^2\theta_1\theta_2} & z_{x_2^2\theta_2^2} \\
x_2\theta_1 & \star & \star & \star & \star & \star & \star & \star & \star & \star & \star & z_{x_2^2\theta_1^2} & z_{x_2^2\theta_1\theta_2} & z_{x_2\theta_1^3} & z_{x_2\theta_1^2\theta_2} & z_{x_2\theta_1\theta_2^2} \\
x_2\theta_2 & \star & \star & \star & \star & \star & \star & \star & \star & \star & \star & \star & z_{x_2^2\theta_2^2} & z_{x_2\theta_1^2\theta_2} & z_{x_2\theta_1\theta_2^2} & z_{x_2\theta_2^3} \\
\theta_1^2 & \star & \star & \star & \star & \star & \star & \star & \star & \star & \star & \star & \star & z_{\theta_1^4} & z_{\theta_1^3\theta_2} & z_{\theta_1^2\theta_2^2} \\
\theta_1\theta_2 & \star & \star & \star & \star & \star & \star & \star & \star & \star & \star & \star & \star & \star & z_{\theta_1^2\theta_2^2} & z_{\theta_1\theta_2^3} \\
\theta_2^2 & \star & \star & \star & \star & \star & \star & \star & \star & \star & \star & \star & \star & \star & \star & z_{\theta_2^4}
\end{array}
\tag{A38}
$$

where the upper triangular entries are exactly $\boldsymbol{z}_4$, the vector of moments up to degree $4$ (thus the expression $\boldsymbol{M}_2\left(\boldsymbol{z}_4\right)$). Moreover, the moment matrix is called a *generalized Hankel matrix* because a moment of order $\boldsymbol{\alpha}$ can appear multiple times in the matrix. For example, the moment $z_{x_1x_2\theta_1\theta_2}$ (highlighted in blue) appears three times in the upper triangular part of $\boldsymbol{M}_2\left(\boldsymbol{z}_4\right)$.

**Localizing matrices.** Using the moment matrix of order 1, $\boldsymbol{M}_1\left(\boldsymbol{z}_2\right)$, the localizing matrix for $h^x = 1 - x_1^2 - x_2^2 = 0$ (eq. (A29)) is:

$$\boldsymbol{M}_1\left(h^x \boldsymbol{z}_2\right) = L_{\boldsymbol{z}}\left(h^x [\boldsymbol{p}]_1\,[\boldsymbol{p}]_1^\top\right) = \tag{A39}$$

$$
\begin{array}{c|ccccc}
 & 1-x_1^2-x_2^2 & x_1-x_1^3-x_1x_2^2 & x_2-x_1^2x_2-x_2^3 & \theta_1-x_1^2\theta_1-x_2^2\theta_1 & \theta_2-x_1^2\theta_2-x_2^2\theta_2 \\
\hline
1 & z_1-z_{x_1^2}-z_{x_2^2} & z_{x_1}-z_{x_1^3}-z_{x_1x_2^2} & z_{x_2}-z_{x_1^2x_2}-z_{x_2^3} & z_{\theta_1}-z_{x_1^2\theta_1}-z_{x_2^2\theta_1} & z_{\theta_2}-z_{x_1^2\theta_2}-z_{x_2^2\theta_2} \\
x_1 & \star & z_{x_1^2}-z_{x_1^4}-z_{x_1^2x_2^2} & z_{x_1x_2}-z_{x_1^3x_2}-z_{x_1x_2^3} & z_{x_1\theta_1}-z_{x_1^3\theta_1}-z_{x_1x_2^2\theta_1} & z_{x_1\theta_2}-z_{x_1^3\theta_2}-z_{x_1x_2^2\theta_2} \\
x_2 & \star & \star & z_{x_2^2}-z_{x_1^2x_2^2}-z_{x_2^4} & z_{x_2\theta_1}-z_{x_1^2x_2\theta_1}-z_{x_2^3\theta_1} & z_{x_2\theta_2}-z_{x_1^2x_2\theta_2}-z_{x_2^3\theta_2} \\
\theta_1 & \star & \star & \star & z_{\theta_1^2}-z_{x_1^2\theta_1^2}-z_{x_2^2\theta_1^2} & z_{\theta_1\theta_2}-z_{x_1^2\theta_1\theta_2}-z_{x_2^2\theta_1\theta_2} \\
\theta_2 & \star & \star & \star & \star & z_{\theta_2^2}-z_{x_1^2\theta_2^2}-z_{x_2^2\theta_2^2}
\end{array}
\tag{A40}
$$

where the columns are indexed by $[\boldsymbol{p}]_1$, and the rows are indexed by $h^x \cdot [\boldsymbol{p}]_1$. The localizing matrix for $h^{\theta_1} = 1 - \theta_1^2 = 0$ (eq. (A30)) is:

$$\boldsymbol{M}_1\left(h^{\theta_1} \boldsymbol{z}_2\right) = L_{\boldsymbol{z}}\left(h^{\theta_1} [\boldsymbol{p}]_1\,[\boldsymbol{p}]_1^\top\right) = \tag{A41}$$

$$
\begin{array}{c|ccccc}
 & 1-\theta_1^2 & x_1-x_1\theta_1^2 & x_2-x_2\theta_1^2 & \theta_1-\theta_1^3 & \theta_2-\theta_1^2\theta_2 \\
\hline
1 & z_1-z_{\theta_1^2} & z_{x_1}-z_{x_1\theta_1^2} & z_{x_2}-z_{x_2\theta_1^2} & z_{\theta_1}-z_{\theta_1^3} & z_{\theta_2}-z_{\theta_1^2\theta_2} \\
x_1 & \star & z_{x_1^2}-z_{x_1^2\theta_1^2} & z_{x_1x_2}-z_{x_1x_2\theta_1^2} & z_{x_1\theta_1}-z_{x_1\theta_1^3} & z_{x_1\theta_2}-z_{x_1\theta_1^2\theta_2} \\
x_2 & \star & \star & z_{x_2^2}-z_{x_2^2\theta_1^2} & z_{x_2\theta_1}-z_{x_2\theta_1^3} & z_{x_2\theta_2}-z_{x_2\theta_1^2\theta_2} \\
\theta_1 & \star & \star & \star & z_{\theta_1^2}-z_{\theta_1^4} & z_{\theta_1\theta_2}-z_{\theta_1^3\theta_2} \\
\theta_2 & \star & \star & \star & \star & z_{\theta_2^2}-z_{\theta_1^2\theta_2^2}
\end{array},
\tag{A42}
$$

where the columns are indexed by $[\boldsymbol{p}]_1$, and the rows are indexed by $h^{\theta_1} \cdot [\boldsymbol{p}]_1$. The localizing matrix for $h^{\theta_2} = 1 - \theta_2^2 = 0$ (eq. (A31)) is:

$$\boldsymbol{M}_1 \left( h^{\theta_2} \boldsymbol{z}_2 \right) = L_{\boldsymbol{z}} \left( h^{\theta_2} [\boldsymbol{p}]_1 [\boldsymbol{p}]_1^{\mathsf{T}} \right) = \tag{A43}$$

$$
\begin{array}{c}
\begin{array}{cccccc}
 & {\color{red}1-\theta_2^2} & {\color{red}x_1-x_1\theta_2^2} & {\color{red}x_2-x_2\theta_2^2} & {\color{red}\theta_1-\theta_1\theta_2^2} & {\color{red}\theta_2-\theta_2^3}
\end{array} \\
\begin{array}{c}
{\color{red}1} \\ {\color{red}x_1} \\ {\color{red}x_2} \\ {\color{red}\theta_1} \\ {\color{red}\theta_2}
\end{array}
\begin{bmatrix}
z_1-z_{\theta_2^2} & z_{x_1}-z_{x_1\theta_2^2} & z_{x_2}-z_{x_2\theta_2^2} & z_{\theta_1}-z_{\theta_1\theta_2^2} & z_{\theta_2}-z_{\theta_2^3} \\
\star & z_{x_1^2}-z_{x_1^2\theta_2^2} & z_{x_1x_2}-z_{x_1x_2\theta_2^2} & z_{x_1\theta_1}-z_{x_1\theta_1\theta_2^2} & z_{x_1\theta_2}-z_{x_1\theta_2^3} \\
\star & \star & z_{x_2^2}-z_{x_2^2\theta_2^2} & z_{x_2\theta_1}-z_{x_2\theta_1\theta_2^2} & z_{x_2\theta_2}-z_{x_2\theta_2^3} \\
\star & \star & \star & z_{\theta_1^2}-z_{\theta_1^2\theta_2^2} & z_{\theta_1\theta_2}-z_{\theta_1\theta_2^3} \\
\star & \star & \star & \star & z_{\theta_2^2}-z_{\theta_2^4}
\end{bmatrix},
\end{array}
\tag{A44}
$$

where the columns are indexed by $[\boldsymbol{p}]_1$, and the rows are indexed by $h^{\theta_2} \cdot [\boldsymbol{p}]_1$.

**Dense Moment Relaxation.** With the expressions of the moment matrices and localizing matrices, the dense moment relaxation at $\kappa = 2$ for 2D single rotation averaging is:

$$
\begin{aligned}
p_2^\star = \min_{\boldsymbol{z}_4 \in \mathbb{R}^{70}} \quad & \sum_{\boldsymbol{\alpha} \in \mathcal{F}} c(\boldsymbol{\alpha}) z_{\boldsymbol{p}^{\boldsymbol{\alpha}}} \\
\text{s.t.} \quad & \boldsymbol{M}_2 (\boldsymbol{z}_4) \succeq 0 \quad \textit{(cf. eq. (A38))}, \\
& \boldsymbol{M}_1 (h^x \boldsymbol{z}_2) = \boldsymbol{0} \quad \textit{(cf. eq. (A40))}, \\
& \boldsymbol{M}_1 \left( h^{\theta_1} \boldsymbol{z}_2 \right) = \boldsymbol{0} \quad \textit{(cf. eq. (A42))}, \\
& \boldsymbol{M}_2 \left( h^{\theta_2} \boldsymbol{z}_2 \right) = \boldsymbol{0} \quad \textit{(cf. eq. (A44))}.
\end{aligned}
\tag{A45}
$$

Now it is clearly that problem (A45) is an SDP because the entries of the moment matrix $\boldsymbol{M}_2 (\boldsymbol{z}_4)$, and the localizing matrices $\boldsymbol{M}_1 (h^x \boldsymbol{z}_2) , \boldsymbol{M}_1 \left( h^{\theta_1} \boldsymbol{z}_2 \right) , \boldsymbol{M}_1 \left( h^{\theta_2} \boldsymbol{z}_2 \right)$ depend *linearly* on the optimization variables $\boldsymbol{z}_4$, and the objective function is also a linear function of $\boldsymbol{z}_4$.

**Remark A6** (**Moment Relaxation for POP *vs.* SDP Relaxation for QCQP**). *The expert reader may now see connections between the moment relaxation for POPs and Shor's SDP relaxation for quadratically constrained quadratic programs (QCQP): the moment relaxation can be seen as first performing a change of variables so that the POP becomes a QCQP (i.e., using $[\boldsymbol{p}]_2$ as the new set of variables, the POP can be seen as a QCQP because $[\boldsymbol{p}]_2$ contains monomials of degree higher than 1), and then apply standard SDP relaxations with redundant constraints. The redundant constraints come from (i) the new set of variables $[\boldsymbol{p}]_2$ are not mutually independent, e.g., $x_1x_2 = x_1 \cdot x_2$, and hence the moment matrix $\boldsymbol{M}_2 (\boldsymbol{z}_4)$ possess linear equality constraints, e.g., the term $z_{x_1x_2\theta_1\theta_2}$ appears multiple times; (ii) combinations of equality constraints, e.g., $h^x = 0$ and $h^{\theta_1} = 0$ implies $h^x \cdot h^{\theta_1} = 0$. Therefore, converting a POP to a QCQP and then applying SDP relaxation (see [4, 24] for two examples) has two drawbacks: first, carefully listing the complete set of redundant constraints can be time-consuming; second, it is challenging to handle inequality constraints. On the contrary, the localizing matrices in moment relaxation provide a systematic way to include all redundant equality and inequality constraints.*

### A2.3 Projection onto $\mathcal{P}$

Here we discuss how to project an estimate to the feasible set of the (POP): this is required to implement the rounding procedure described in eq. (A24). Denote $\boldsymbol{p}_v = [\boldsymbol{x}_v^{\mathsf{T}}, \boldsymbol{\theta}_v^{\mathsf{T}}]^{\mathsf{T}} = \boldsymbol{v}_\kappa (\boldsymbol{p})$ as the entries of $\boldsymbol{v}_\kappa$ at indices corresponding to the locations of $\boldsymbol{p}$ in $[\boldsymbol{p}]_\kappa$ (recall that $\boldsymbol{v}_\kappa$ is obtained from the spectral decomposition of a moment matrix $\boldsymbol{M}_\kappa (\boldsymbol{z}_4)$ with rank larger than 1, and in general $\boldsymbol{p}_v \notin \mathcal{P}$). To project $\boldsymbol{p}_v$ onto $\mathcal{P}$ (eq. (A24)), we need to project $\boldsymbol{x}_v$ onto $\mathcal{X}$ and project $\boldsymbol{\theta}_v$ onto $\{\pm 1\}_{i=1}^N$.

**Project $\boldsymbol{\theta}_v$ onto $\{\pm 1\}_{i=1}^N$.** The projection of $\boldsymbol{\theta}_v$ onto the set of binary variables $\{\pm 1\}_{i=1}^N$, denoted $\hat{\boldsymbol{\theta}}$, is straightforward:

$$\left[ \hat{\boldsymbol{\theta}} \right]_i = \text{proj}_{\{\pm 1\}} \left( [\boldsymbol{\theta}_v]_i \right) = \text{sgn} \left( [\boldsymbol{\theta}_v]_i \right), i = 1, \dots, N, \tag{A46}$$

where $[\cdot]_i$ denotes the $i$-th entry of a vector and $\text{sgn} (\cdot)$ denotes the sign function.

**Project $\boldsymbol{x}_v$ onto $\mathcal{X}$.** Because Examples 1-4 have different feasible sets $\mathcal{X}$ for the geometric models, the projections are different.

**Example 1 (Single Rotation Averaging).** $\boldsymbol{x} = \boldsymbol{R} \in \mathrm{SO}(3)$, so the projection is:

$$\hat{\boldsymbol{R}} = \mathrm{proj}_{\mathrm{SO}(3)}\left(\boldsymbol{x}_v\right) = \boldsymbol{U} \mathrm{diag}\left(1, 1, \det\left(\boldsymbol{U}\right) \cdot \det\left(\boldsymbol{V}\right)\right) \boldsymbol{V}^{\mathsf{T}}, \tag{A47}$$

where $\boldsymbol{x}_v = \boldsymbol{U}\boldsymbol{S}\boldsymbol{V}^{\mathsf{T}}, \boldsymbol{U}, \boldsymbol{V} \in \mathrm{O}(3)$ is the singular value decomposition for $\boldsymbol{x}_v$ (first reshape $\boldsymbol{x}_v \in \mathbb{R}^9$ into a $3 \times 3$ matrix) [9].

**Example 2 (Shape Alignment).** $\boldsymbol{x} = s\Pi\boldsymbol{R}, s \in [0, \bar{s}], \boldsymbol{R} \in \mathrm{SO}(3)$, so the projection is:

$$\hat{s} = \min\left\{\bar{s}, \frac{\sigma_1 + \sigma_2}{2}\right\}, \hat{\boldsymbol{R}} = \begin{bmatrix} \boldsymbol{r}_1^{\mathsf{T}} \\ \boldsymbol{r}_2^{\mathsf{T}} \\ (\boldsymbol{r}_1 \times \boldsymbol{r}_2)^{\mathsf{T}} \end{bmatrix}, \tag{A48}$$

where $\sigma_1, \sigma_2, \boldsymbol{r}_1, \boldsymbol{r}_2$ come from the singular value decomposition of $\boldsymbol{x}_v$ (first reshape $\boldsymbol{x}_v$ into a $2 \times 3$ matrix):

$$\boldsymbol{x}_v = \boldsymbol{U} \begin{bmatrix} \sigma_1 & 0 & 0 \\ 0 & \sigma_2 & 0 \end{bmatrix} \boldsymbol{V}^{\mathsf{T}}, \boldsymbol{U} \in \mathrm{O}(2), \boldsymbol{V} \in \mathrm{O}(3), \quad \begin{bmatrix} \boldsymbol{r}_1^{\mathsf{T}} \\ \boldsymbol{r}_2^{\mathsf{T}} \end{bmatrix} = \boldsymbol{U} \begin{bmatrix} 1 & 0 & 0 \\ 0 & 1 & 0 \end{bmatrix} \boldsymbol{V}^{\mathsf{T}}. \tag{A49}$$

**Example 3-4 (Point Cloud Registration and Mesh Registration).** $\boldsymbol{x} = (\boldsymbol{R}, \boldsymbol{t}), \boldsymbol{R} \in \mathrm{SO}(3), \|\boldsymbol{t}\| \leq T$, so the projection is:

$$\hat{\boldsymbol{R}} = \mathrm{proj}_{\mathrm{SO}(3)}\left(\boldsymbol{x}_v^r\right), \hat{\boldsymbol{t}} = \min\left\{\left\|\boldsymbol{x}_v^t\right\|, T\right\} \frac{\boldsymbol{x}_v^t}{\left\|\boldsymbol{x}_v^t\right\|}, \tag{A50}$$

where $\boldsymbol{x}_v^r$ denotes the entries of $\boldsymbol{x}_v$ corresponding to $\boldsymbol{R}$, and $\boldsymbol{x}_v^t$ denotes the entries of $\boldsymbol{x}_v$ corresponding to $\boldsymbol{t}$.

## A3 Proof of Theorem 7 (Sparse Moment Relaxation)

*Proof.* Let us first show that the sparse moment relaxation (4) is indeed a valid relaxation, *i.e.,* (a) the sparse set of monomials $[\boldsymbol{p}]_{2\mathcal{B}}$ contains all the monomials in the objective function $f(\boldsymbol{p})$ of the POP (2) (otherwise, the objective function of the relaxation (4) is not equivalent to the objective function of the POP (2)); and (b) the sparse set of moments $\boldsymbol{z}_{2\mathcal{B}}$ contains all the moments appearing in the localizing matrices of (4) (otherwise, the optimization contains undefined variables). To see (a), from the sparsity of the objective function $f$ (*cf.* property (i) of Proposition 5), we know that $f = \sum_{i=1}^{N} f_i$ and each $f_i$ at most contains monomials of type $[\boldsymbol{x}]_2$, $\theta_i$ and $\theta_i \cdot [\boldsymbol{x}]_2$ (*cf.* the expression of $f_i$ in eq. (A1)), all of which are included in the sparse set of monomials $[\boldsymbol{p}]_{2\mathcal{B}}$ (recall $[\boldsymbol{p}]_{\mathcal{B}} = [1, \boldsymbol{x}^{\mathsf{T}}, \boldsymbol{\theta}^{\mathsf{T}}, (\boldsymbol{x})_2^{\mathsf{T}}, \boldsymbol{\theta}^{\mathsf{T}} \otimes \boldsymbol{x}^{\mathsf{T}}]^{\mathsf{T}}$). To see (b), we observe that $h^x$ and $g$ only contain monomials $[\boldsymbol{x}]_2$, and $\boldsymbol{z}_2$ only contain monomials $[\boldsymbol{p}]_2$, so the product $[\boldsymbol{x}]_2 \otimes [\boldsymbol{p}]_2$ is included in $[\boldsymbol{p}]_{2\mathcal{B}}$. Hence, the moments in the localizing matrices $\boldsymbol{M}_1\left(h^x \boldsymbol{z}_2\right)$ and $\boldsymbol{M}_1\left(g \boldsymbol{z}_2\right)$ are included in the moment vector $\boldsymbol{z}_{2\mathcal{B}}$. Similarly, $h^\theta$ only contains monomials $[\boldsymbol{\theta}]_2$, and $\boldsymbol{z}_{2\mathcal{B}_x}$ only contains monomials $[\boldsymbol{x}]_2$, so the product $[\boldsymbol{\theta}]_2 \otimes [\boldsymbol{x}]_2$ is included in $[\boldsymbol{p}]_{2\mathcal{B}}$. Hence, the moments in the localizing matrices $\boldsymbol{M}_{\mathcal{B}_x}\left(h^\theta \boldsymbol{z}_{2\mathcal{B}_x}\right)$ are also included in the moment vector $\boldsymbol{z}_{2\mathcal{B}}$.

**Lower Bound.** Then we prove that $p_{\mathcal{B}}^\star \leq p_2^\star$, *i.e.,* the optimal value of the sparse relaxation (4) is a lower bound of the optimal value of the dense relaxation (3) at order $\kappa = 2$. We prove this by showing that the feasible set of the dense relaxation is contained in the feasible set of the sparse relaxation. To see this, consider $\boldsymbol{z}_4$ as an arbitrary point in the feasible set of the dense relaxation (3), *i.e.,* $\boldsymbol{z}_4$ satisfies $z_0 = 1, \boldsymbol{M}_2\left(\boldsymbol{z}_4\right) \succeq 0, \boldsymbol{M}_1\left(h\boldsymbol{z}_2\right) = \boldsymbol{0}, \forall h \in \boldsymbol{h}$, and $\boldsymbol{M}_1\left(g\boldsymbol{z}_2\right) \succeq 0, \forall g \in \boldsymbol{g}$. Then $\boldsymbol{z}_{2\mathcal{B}} \subset \boldsymbol{z}_4$, the sub-vector of $\boldsymbol{z}_4$ corresponding to the sparse set of monomials $[\boldsymbol{p}]_{2\mathcal{B}}$, must be feasible for the sparse relaxation (4). This is because $\boldsymbol{M}_{\mathcal{B}}\left(\boldsymbol{z}_{2\mathcal{B}}\right) \succeq 0$ must hold as $\boldsymbol{M}_{\mathcal{B}}\left(\boldsymbol{z}_{2\mathcal{B}}\right)$ is a principal sub-matrix of the full moment matrix $\boldsymbol{M}_2\left(\boldsymbol{z}_4\right)$; $\boldsymbol{M}_{\mathcal{B}_x}\left(h\boldsymbol{z}_{2\mathcal{B}_x}\right) = \boldsymbol{0}$ must hold as $\boldsymbol{M}_{\mathcal{B}_x}\left(h\boldsymbol{z}_{2\mathcal{B}_x}\right)$ is a principal sub-matrix of the full localizing matrix $\boldsymbol{M}_1\left(h\boldsymbol{z}_2\right), \forall h \in \boldsymbol{h}^\theta$; and $\boldsymbol{M}_1\left(h\boldsymbol{z}_2\right)$ and $\boldsymbol{M}_1\left(g\boldsymbol{z}_2\right)$ are the same as the localizing matrices in the dense relaxation.

**Rounding and Relative Duality Gap.** Property (iv) of the dense relaxation still holds for the sparse relaxation. Because $p_{\mathcal{B}}^\star \leq p_2^\star$ and $p_2^\star \leq f^\star$, we have $p_{\mathcal{B}}^\star \leq f^\star$ is also a valid lower bound for $f^\star$, the true optimal objective value of the POP. Additionally, since the sparse set of monomials $[\boldsymbol{p}]_{\mathcal{B}}$ still contains $[\boldsymbol{p}]_1$, the degree-1 monomials of $\boldsymbol{x}$ and $\boldsymbol{\theta}$, one can use the same rounding method (*i.e.,* spectral decomposition and the projection onto the feasible set $\mathcal{P}$ in (A24)) to obtain a feasible

solution $\hat{p}$, which gives a value $\hat{f} = f(\hat{p})$ that satisfies $p_{\mathcal{B}}^\star \leq f^\star \leq \hat{f}$. The relative duality gap can then be calculated similar to eq. (A26) as:

$$\eta_{\mathcal{B}} = \frac{\hat{f} - p_{\mathcal{B}}^\star}{\hat{f}}, \tag{A51}$$

where a smaller $\eta_{\mathcal{B}}$ implies a tighter relaxation and $\eta_{\mathcal{B}} = 0$ if and only if the relaxation is tight.

**Optimality Certificate**. Showing property (iii) of the dense relaxation also holds for the sparse relaxation is non-trivial because Theorem A5 does not hold for an arbitrary sparse moment matrix (*i.e.,* a moment matrix with rows and columns indexed by a sparse set of monomials $[\boldsymbol{p}]_{\mathcal{B}} \subset [\boldsymbol{p}]_2$). Therefore, we will show that the sparse moment matrix $\boldsymbol{M}_{\mathcal{B}}(\boldsymbol{z}_{2\mathcal{B}})$ can be *extended* to a full moment matrix $\boldsymbol{M}_2(\boldsymbol{z}_4)$ when $\mathrm{rank}(\boldsymbol{M}_{\mathcal{B}}(\boldsymbol{z}_{2\mathcal{B}})) = 1$. Let us first introduce the notion of a flat extension.

**Definition A7 (Flat Extension).** *Given two moment matrices $\boldsymbol{M}_{\mathcal{B}}(\boldsymbol{z}_{2\mathcal{B}})$ and $\boldsymbol{M}_{\mathcal{A}}(\boldsymbol{z}_{2\mathcal{A}})$, with $\mathcal{B} \subset \mathcal{A}$ (recall that the rows and columns of $\boldsymbol{M}_{\mathcal{B}}(\boldsymbol{z}_{2\mathcal{B}})$ (resp. $\boldsymbol{M}_{\mathcal{A}}(\boldsymbol{z}_{2\mathcal{A}})$) are indexed by monomials $[\boldsymbol{p}]_{\mathcal{B}}$ (resp. $[\boldsymbol{p}]_{\mathcal{A}}$)), then $\boldsymbol{M}_{\mathcal{A}}(\boldsymbol{z}_{2\mathcal{A}})$ is said to be a flat extension of $\boldsymbol{M}_{\mathcal{B}}(\boldsymbol{z}_{2\mathcal{B}})$ if $\boldsymbol{M}_{\mathcal{B}}(\boldsymbol{z}_{2\mathcal{B}})$ coincides with the sub-matrix of $\boldsymbol{M}_{\mathcal{A}}(\boldsymbol{z}_{2\mathcal{A}})$ indexed by $[\boldsymbol{p}]_{\mathcal{B}}$ and $\mathrm{rank}(\boldsymbol{M}_{\mathcal{B}}(\boldsymbol{z}_{2\mathcal{B}})) = \mathrm{rank}(\boldsymbol{M}_{\mathcal{A}}(\boldsymbol{z}_{2\mathcal{A}}))$.*

Our goal is to show that $\boldsymbol{M}_{\mathcal{B}}(\boldsymbol{z}_{2\mathcal{B}})$ admits a flat extension $\boldsymbol{M}_2(\boldsymbol{z}_4)$ when $\mathrm{rank}(\boldsymbol{M}_{\mathcal{B}}(\boldsymbol{z}_{2\mathcal{B}})) = 1$, so that $\mathrm{rank}(\boldsymbol{M}_2(\boldsymbol{z}_4)) = 1$ is also true, in which case we recover the dense moment relaxation and obtain an optimality certificate. To do so, we will show that the sparse moment matrix $\boldsymbol{M}_{\mathcal{B}}(\boldsymbol{z}_{2\mathcal{B}})$ satisfies the *generalized flat extension theorem* in [16], stated below.

**Theorem A8 (Generalized Flat Extension [16, Theorem 1.4]).** *Given a monomial basis $[\boldsymbol{p}]_{\mathcal{C}}$, define its* closure *to be the set:*

$$[\boldsymbol{p}]_{\mathcal{C}^+} \doteq [\boldsymbol{p}]_{\mathcal{C}} \cup \left( \cup_{i=1}^{\tilde{n}} p_i [\boldsymbol{p}]_{\mathcal{C}} \right) \doteq \{ \boldsymbol{p}^{\boldsymbol{\alpha}}, p_1 \boldsymbol{p}^{\boldsymbol{\alpha}}, \ldots, p_{\tilde{n}} \boldsymbol{p}^{\boldsymbol{\alpha}} | \boldsymbol{\alpha} \in \mathcal{C} \} . \tag{A52}$$

*For example, let $\tilde{n} = 3$, and $[\boldsymbol{p}]_{\mathcal{C}} = [p_1]$, then $[\boldsymbol{p}]_{\mathcal{C}^+} = [p_1, p_1^2, p_1 p_2, p_1 p_3]$. In addition, the monomial set $[\boldsymbol{p}]_{\mathcal{C}}$ is said to be* connected to 1 *if $1 \in [\boldsymbol{p}]_{\mathcal{C}}$ and every monomial $\boldsymbol{p}^{\boldsymbol{\alpha}}$ can be written as $\boldsymbol{p}^{\boldsymbol{\alpha}} = p_{i_1} p_{i_2} \cdots p_{i_k}$ with $p_{i_1}, p_{i_1} p_{i_2}, \ldots, p_{i_1} p_{i_2} \cdots p_{i_k} \in [\boldsymbol{p}]_{\mathcal{C}}$. For example $[1, p_1, p_1 p_2]$ is connected to 1, but $[1, p_1 p_2]$ is not. Then the generalized flat extension theorem states:*

*If $[\boldsymbol{p}]_{\mathcal{C}}$ is connected to 1, and $\boldsymbol{M}_{\mathcal{C}^+}(\boldsymbol{z}_{2\mathcal{C}^+})$ is a flat extension of $\boldsymbol{M}_{\mathcal{C}}(\boldsymbol{z}_{2\mathcal{C}})$, then $\boldsymbol{M}_{\mathcal{C}^+}(\boldsymbol{z}_{2\mathcal{C}^+})$ admits a unique flat extension $\boldsymbol{M}_{\kappa}(\boldsymbol{z}_{2\kappa})$ for any $\kappa \geq \alpha_{\max}$, where $\alpha_{\max} \doteq \max\{|\boldsymbol{\alpha}|: \boldsymbol{\alpha} \in \mathcal{C}^+\}$ denotes the maximum degree of the monomials in $[\boldsymbol{p}]_{\mathcal{C}^+}$.*

Using Theorem A8, let $[\boldsymbol{p}]_{\mathcal{C}} \doteq [\boldsymbol{p}]_{\mathcal{B}_x} = [1, \boldsymbol{x}^{\mathsf{T}}]^{\mathsf{T}}$, then obviously $[\boldsymbol{p}]_{\mathcal{C}}$ is connected to 1. The closure of $[\boldsymbol{p}]_{\mathcal{C}}$ is $[\boldsymbol{p}]_{\mathcal{C}^+} = [\boldsymbol{p}]_{\mathcal{B}} = [1, \boldsymbol{x}^{\mathsf{T}}, \boldsymbol{\theta}^{\mathsf{T}}, (\boldsymbol{x})_2, \boldsymbol{\theta}^{\mathsf{T}} \otimes \boldsymbol{x}^{\mathsf{T}}]^{\mathsf{T}}$. If $\mathrm{rank}(\boldsymbol{M}_{\mathcal{B}}(\boldsymbol{z}_{2\mathcal{B}})) = 1$, then $\mathrm{rank}(\boldsymbol{M}_{\mathcal{B}}(\boldsymbol{z}_{2\mathcal{B}})) = \mathrm{rank}(\boldsymbol{M}_{\mathcal{B}_x}(\boldsymbol{z}_{2\mathcal{B}_x})) = 1$ and $\boldsymbol{M}_{\mathcal{B}}(\boldsymbol{z}_{2\mathcal{B}})$ is a flat extension of $\boldsymbol{M}_{\mathcal{B}_x}(\boldsymbol{z}_{2\mathcal{B}_x})$. Therefore, $\boldsymbol{M}_{\mathcal{B}}(\boldsymbol{z}_{2\mathcal{B}})$ admits a flat extension $\boldsymbol{M}_{\kappa}(\boldsymbol{z}_{2\kappa})$ for any $\kappa \geq 2$. In particular, setting $\kappa = 2$, we recover a dense moment matrix $\boldsymbol{M}_2(\boldsymbol{z}_4)$ with $\mathrm{rank}(\boldsymbol{M}_2(\boldsymbol{z}_4)) = 1$. It remains to show that the moments $\boldsymbol{z}_4$ (obtained from the flat extension) satisfy the constraints on the localizing matrices in the dense relaxation (3). The only different constraint between the dense relaxation (3) at $\kappa = 2$ and the sparse relaxation (4) is that, the constraint $\boldsymbol{M}_1(h\boldsymbol{z}_2) = \boldsymbol{0}$ has been relaxed to $\boldsymbol{M}_1(h\boldsymbol{z}_{2\mathcal{B}_x}) = \boldsymbol{0}$ for $h \in \boldsymbol{h}^{\theta}$. However, we observe that the top-left entry of $\boldsymbol{M}_1(h\boldsymbol{z}_{2\mathcal{B}_x})$ is $z_1 - z_{\theta_i^2}$ when $h = h^{\theta_i} = 1 - \theta_i^2$, and $z_1 - z_{\theta_i^2} = 0$ implies $z_{\theta_i^2} = z_{\theta_i}^2 = 1$ (due to $\mathrm{rank}(\boldsymbol{M}_2(\boldsymbol{z}_4)) = 1$), which implies $z_{\theta_i} = \pm 1$ and the solution is indeed binary and supported on $\mathcal{P}$ and must satisfy $\boldsymbol{M}_1(h\boldsymbol{z}_2) = \boldsymbol{0}$. ∎

# A4 Proof of Theorem 8 (Sufficient Condition for Global Optimality)

*Proof.* If problem (5) is feasible, then for any $\boldsymbol{p} \in \mathcal{P}$, we have:

$$[\boldsymbol{p}]_{\mathcal{B}}^{\mathsf{T}} \boldsymbol{S}_0 [\boldsymbol{p}]_{\mathcal{B}} \geq 0, \quad [\boldsymbol{p}]_1^{\mathsf{T}} \boldsymbol{S}_k [\boldsymbol{p}]_1 \geq 0, \forall k = 1, \ldots, l_g, \tag{A53}$$

because $\boldsymbol{S}_0$ and $\boldsymbol{S}_k, k = 1, \ldots, l_g$, are PSD matrices (*i.e.,* $[\boldsymbol{p}]_{\mathcal{B}}^{\mathsf{T}} \boldsymbol{S}_0 [\boldsymbol{p}]_{\mathcal{B}}$ and $[\boldsymbol{p}]_1^{\mathsf{T}} \boldsymbol{S}_k [\boldsymbol{p}]_1$ are SOS polynomials). In addition, $g_k(\boldsymbol{p}) \geq 0, \forall \boldsymbol{p} \in \mathcal{P}$ by construction of the inequality constraints of the POP (2). Therefore, the right-hand side of eq. (6) is nonnegative. On the other hand, the left-hand side of eq. (6) reduces to $f(\boldsymbol{p}) - \hat{f}$ for any $\boldsymbol{p} \in \mathcal{P}$ due to the equality constraints $h_j(\boldsymbol{p}) = 0, \forall h_j \in \boldsymbol{h}^x$

and $h_j(\boldsymbol{p}) = 0, \forall h_j \in \boldsymbol{h}^\theta$ of the POP (2). Combining these two observations, we have $f(\boldsymbol{p}) - \hat{f} \geq 0$ for any $\boldsymbol{p} \in \mathcal{P}$, which implies that $\hat{f}$ is the global minimum of the POP and $\hat{\boldsymbol{p}}$ is the corresponding global minimizer.

Next we show how to convert problem (5) into the compact SDP formulation in (7), by writing every polynomial in (6) as a sum of products between coefficients (parametrized by the unknowns $\boldsymbol{\lambda}_j^x, \boldsymbol{\lambda}_j^\theta, \boldsymbol{S}_0$ and $\boldsymbol{S}_k$) and the monomial basis $[\boldsymbol{p}]_{2\mathcal{B}}$.

**Right-hand Side of** (6). We start from the right-hand side of (6). To do so, we first write the monomial outer product $[\boldsymbol{p}]_\mathcal{B} [\boldsymbol{p}]_\mathcal{B}^\mathsf{T}$ as:

$$[\boldsymbol{p}]_\mathcal{B} [\boldsymbol{p}]_\mathcal{B}^\mathsf{T} = \sum_{\boldsymbol{\alpha} \in 2\mathcal{B}} \boldsymbol{W}_{\boldsymbol{\alpha}}^0 \boldsymbol{p}^{\boldsymbol{\alpha}}, \tag{A54}$$

where $\boldsymbol{W}_{\boldsymbol{\alpha}}^0 \in \mathcal{S}^{m(\mathcal{B})}$ is the "0/1" *monomial indicator matrix* with rows and columns indexed by $[\boldsymbol{p}]_\mathcal{B}$, whose entries are defined as:

$$\left[\boldsymbol{W}_{\boldsymbol{\alpha}}^0\right]_{\boldsymbol{p}^{\boldsymbol{\alpha}_1}, \boldsymbol{p}^{\boldsymbol{\alpha}_2}} = \begin{cases} 1 & \text{if } \boldsymbol{\alpha}_1 + \boldsymbol{\alpha}_2 = \boldsymbol{\alpha} \\ 0 & \text{otherwise} \end{cases}. \tag{A55}$$

Using the expression in (A54), we can write the SOS polynomial $s_0 \doteq [\boldsymbol{p}]_\mathcal{B}^\mathsf{T} \boldsymbol{S}_0 [\boldsymbol{p}]_\mathcal{B}$ as:

$$s_0 \doteq [\boldsymbol{p}]_\mathcal{B}^\mathsf{T} \boldsymbol{S}_0 [\boldsymbol{p}]_\mathcal{B} = \left\langle \boldsymbol{S}_0, [\boldsymbol{p}]_\mathcal{B} [\boldsymbol{p}]_\mathcal{B}^\mathsf{T} \right\rangle = \left\langle \boldsymbol{S}_0, \sum_{\boldsymbol{\alpha} \in 2\mathcal{B}} \boldsymbol{W}_{\boldsymbol{\alpha}}^0 \boldsymbol{p}^{\boldsymbol{\alpha}} \right\rangle = \sum_{\boldsymbol{\alpha} \in 2\mathcal{B}} \left\langle \boldsymbol{S}_0, \boldsymbol{W}_{\boldsymbol{\alpha}}^0 \right\rangle \boldsymbol{p}^{\boldsymbol{\alpha}}, \tag{A56}$$

where $\langle \boldsymbol{A}, \boldsymbol{B} \rangle \doteq \operatorname{tr}(\boldsymbol{A}\boldsymbol{B})$ denotes the inner product between two symmetric matrices $\boldsymbol{A}, \boldsymbol{B} \in \mathcal{S}^n$. Similarly, for each outer product $g_k [\boldsymbol{p}]_1 [\boldsymbol{p}]_1^\mathsf{T}$, we write them as:

$$g_k [\boldsymbol{p}]_1 [\boldsymbol{p}]_1^\mathsf{T} = \sum_{\boldsymbol{\alpha} \in 2\mathcal{B}} \boldsymbol{W}_{\boldsymbol{\alpha}}^k \boldsymbol{p}^{\boldsymbol{\alpha}}, k = 1, \dots, l_g, \tag{A57}$$

where $\boldsymbol{W}_{\boldsymbol{\alpha}}^k \in \mathcal{S}^{m_{\tilde{n}}(1)}$ is the *monomial coefficient matrix* with rows and columns indexed by $[\boldsymbol{p}]_1$, whose entries are defined as:

$$\left[\boldsymbol{W}_{\boldsymbol{\alpha}}^k\right]_{\boldsymbol{p}^{\boldsymbol{\alpha}_1}, \boldsymbol{p}^{\boldsymbol{\alpha}_2}} = c_k(\boldsymbol{\alpha} - \boldsymbol{\alpha}_1 - \boldsymbol{\alpha}_2), \tag{A58}$$

where $c_k(\boldsymbol{\alpha} - \boldsymbol{\alpha}_1 - \boldsymbol{\alpha}_2)$ denotes the coefficient of $g_k$ corresponding to the monomial $\boldsymbol{p}^{\boldsymbol{\alpha} - \boldsymbol{\alpha}_1 - \boldsymbol{\alpha}_2}$. Note that $\boldsymbol{W}_{\boldsymbol{\alpha}}^k$ is not an "0/1" matrix due to the multiplication of $g_k$ with the monomial outer product $[\boldsymbol{p}]_1 [\boldsymbol{p}]_1^\mathsf{T}$. Using the expression in (A57), we can write the nonnegative polynomials $s_k \doteq g_k [\boldsymbol{p}]_1^\mathsf{T} \boldsymbol{S}_k [\boldsymbol{p}]_1, k = 1, \dots, l_g$, as:

$$s_k \doteq g_k [\boldsymbol{p}]_1^\mathsf{T} \boldsymbol{S}_k [\boldsymbol{p}]_1 = \left\langle \boldsymbol{S}_k, g_k [\boldsymbol{p}]_1 [\boldsymbol{p}]_1^\mathsf{T} \right\rangle = \left\langle \boldsymbol{S}_k, \sum_{\boldsymbol{\alpha} \in 2\mathcal{B}} \boldsymbol{W}_{\boldsymbol{\alpha}}^k \boldsymbol{p}^{\boldsymbol{\alpha}} \right\rangle = \sum_{\boldsymbol{\alpha} \in 2\mathcal{B}} \left\langle \boldsymbol{S}_k, \boldsymbol{W}_{\boldsymbol{\alpha}}^k \right\rangle \boldsymbol{p}^{\boldsymbol{\alpha}}. \tag{A59}$$

Eq. (A56) and (A59) have written the right-hand side of (6) as a sum of products, where each product is between a monomial $\boldsymbol{p}^{\boldsymbol{\alpha}}$ and a coefficient, $\left\langle \boldsymbol{S}_0, \boldsymbol{W}_{\boldsymbol{\alpha}}^0 \right\rangle$ or $\left\langle \boldsymbol{S}_k, \boldsymbol{W}_{\boldsymbol{\alpha}}^k \right\rangle$, parametrized by the unknown PSD matrices $\boldsymbol{S}_0$ and $\boldsymbol{S}_k, k = 1, \dots, l_g$.

**Left-hand Side of** (6). We now perform similar algebra for the left-hand side of (6). We write $f(\boldsymbol{p}) - \hat{f}$ as:

$$f(\boldsymbol{p}) - \hat{f} = \sum_{\boldsymbol{\alpha} \in 2\mathcal{B}} c_f(\boldsymbol{\alpha}) \boldsymbol{p}^{\boldsymbol{\alpha}}, \tag{A60}$$

where $c_f(\boldsymbol{\alpha})$ denotes the coefficient of $f(\boldsymbol{p}) - \hat{f}$ corresponding to the monomial $\boldsymbol{p}^{\boldsymbol{\alpha}}$. We write $h_j [\boldsymbol{p}]_2, h_j \in \boldsymbol{h}^x$ as:

$$h_j [\boldsymbol{p}]_2 = \sum_{\boldsymbol{\alpha} \in 2\mathcal{B}} \boldsymbol{w}_{\boldsymbol{\alpha}}^{x_j} \boldsymbol{p}^{\boldsymbol{\alpha}}, \tag{A61}$$

where $\boldsymbol{w}_{\boldsymbol{\alpha}}^{x_j} \in \mathbb{R}^{m_{\tilde{n}}(2)}$ is a vector of coefficients indexed by $[\boldsymbol{p}]_2$, whose entries are defined as:

$$\left[\boldsymbol{w}_{\boldsymbol{\alpha}}^{x_j}\right]_{\boldsymbol{p}^{\boldsymbol{\alpha}_1}} = c_{h_j^x}(\boldsymbol{\alpha} - \boldsymbol{\alpha}_1), \tag{A62}$$

where $c_{h_j^x}(\boldsymbol{\alpha} - \boldsymbol{\alpha}_1)$ is the coefficient of $h_j \in \boldsymbol{h}^x$ corresponding to the monomial $\boldsymbol{p}^{\boldsymbol{\alpha}-\boldsymbol{\alpha}_1}$. Using the expression in (A61), we can write $q_j^x \doteq h_j \left[\boldsymbol{p}\right]_2^{\mathsf{T}} \boldsymbol{\lambda}_j^x, \forall h_j \in \boldsymbol{h}^x$, as:

$$q_j^x \doteq h_j \left[\boldsymbol{p}\right]_2^{\mathsf{T}} \boldsymbol{\lambda}_j^x = \left\langle \boldsymbol{\lambda}_j^x, h_j \left[\boldsymbol{p}\right]_2 \right\rangle = \left\langle \boldsymbol{\lambda}_j^x, \sum_{\boldsymbol{\alpha} \in 2\mathcal{B}} \boldsymbol{w}_{\boldsymbol{\alpha}}^{x_j} \boldsymbol{p}^{\boldsymbol{\alpha}} \right\rangle = \sum_{\boldsymbol{\alpha} \in 2\mathcal{B}} \left\langle \boldsymbol{\lambda}_j^x, \boldsymbol{w}_{\boldsymbol{\alpha}}^{x_j} \right\rangle \boldsymbol{p}^{\boldsymbol{\alpha}}, \qquad \text{(A63)}$$

where $\langle \boldsymbol{a}, \boldsymbol{b} \rangle \doteq \boldsymbol{a}^{\mathsf{T}} \boldsymbol{b}$ denotes the inner product between two vectors $\boldsymbol{a}, \boldsymbol{b} \in \mathbb{R}^n$. Similarly, we write $h_j \left[\boldsymbol{x}\right]_2, h_j \in \boldsymbol{h}^\theta$ as:

$$h_j \left[\boldsymbol{x}\right]_2 = \sum_{\boldsymbol{\alpha} \in 2\mathcal{B}} \boldsymbol{w}_{\boldsymbol{\alpha}}^{\theta_j} \boldsymbol{p}^{\boldsymbol{\alpha}}, \qquad \text{(A64)}$$

where $\boldsymbol{w}_{\boldsymbol{\alpha}}^{\theta_j} \in \mathbb{R}^{m_n(2)}$ is a vector of coefficients indexed by $\left[\boldsymbol{x}\right]_2$, whose entries are defined as:

$$\left[\boldsymbol{w}_{\boldsymbol{\alpha}}^{\theta_j}\right]_{\boldsymbol{p}^{\boldsymbol{\alpha}_1}} = c_{h_j^\theta}(\boldsymbol{\alpha} - \boldsymbol{\alpha}_1), \qquad \text{(A65)}$$

where $c_{h_j^\theta}(\boldsymbol{\alpha} - \boldsymbol{\alpha}_1)$ is the coefficient of $h_j \in \boldsymbol{h}^\theta$ corresponding to the monomial $\boldsymbol{p}^{\boldsymbol{\alpha}-\boldsymbol{\alpha}_1}$. Using the expression in (A64), we can write $q_j^\theta \doteq h_j \left[\boldsymbol{x}\right]_2^{\mathsf{T}} \boldsymbol{\lambda}_j^\theta$ as:

$$q_j^\theta \doteq h_j \left[\boldsymbol{x}\right]_2^{\mathsf{T}} \boldsymbol{\lambda}_j^\theta = \left\langle \boldsymbol{\lambda}_j^\theta, h_j \left[\boldsymbol{x}\right]_2 \right\rangle = \left\langle \boldsymbol{\lambda}_j^\theta, \sum_{\boldsymbol{\alpha} \in 2\mathcal{B}} \boldsymbol{w}_{\boldsymbol{\alpha}}^{\theta_j} \boldsymbol{p}^{\boldsymbol{\alpha}} \right\rangle = \sum_{\boldsymbol{\alpha} \in 2\mathcal{B}} \left\langle \boldsymbol{\lambda}_j^\theta, \boldsymbol{w}_{\boldsymbol{\alpha}}^{\theta_j} \right\rangle \boldsymbol{p}^{\boldsymbol{\alpha}}. \qquad \text{(A66)}$$

Eq. (A63) and (A66) have written the left-hand side of (6) as a sum of products, where each product is between a monomial $\boldsymbol{p}^{\boldsymbol{\alpha}}$ and a coefficient, $\left\langle \boldsymbol{\lambda}_j^x, \boldsymbol{w}_{\boldsymbol{\alpha}}^{x_j} \right\rangle$ or $\left\langle \boldsymbol{\lambda}_j^\theta, \boldsymbol{w}_{\boldsymbol{\alpha}}^{\theta_j} \right\rangle$, parametrized by the unknown vectors $\boldsymbol{\lambda}_j^x, j = 1, \ldots, |\boldsymbol{h}^x|$, and $\boldsymbol{\lambda}_j^\theta, j = 1, \ldots, |\boldsymbol{h}^\theta|$, where $|\boldsymbol{h}^x|$ and $|\boldsymbol{h}^\theta|$ denotes the cardinality of the sets $\boldsymbol{h}^x$ and $\boldsymbol{h}^\theta$, respectively.

**Obtaining the Compact SDP** (7). Combining the left-hand side (eq. (A60), (A63) and (A66)) and the right-hand side (eq. (A56) and (A59)) of eq. (6), we are ready to write down the final expression for the compact SDP (7). To do so, we first concatenate all the independent unknown variables into a single vector, called the dual variable:

$$\boldsymbol{d} = [(\boldsymbol{\lambda}_1^x)^{\mathsf{T}}, \ldots, (\boldsymbol{\lambda}_{|\boldsymbol{h}^x|}^x)^{\mathsf{T}}, (\boldsymbol{\lambda}_1^\theta)^{\mathsf{T}}, \ldots, (\boldsymbol{\lambda}_{|\boldsymbol{h}^\theta|}^\theta)^{\mathsf{T}}, \text{svec}\left(\boldsymbol{S}_1\right)^{\mathsf{T}}, \ldots, \text{svec}\left(\boldsymbol{S}_{l_g}\right)^{\mathsf{T}}, \text{svec}\left(\boldsymbol{S}_0\right)^{\mathsf{T}}]^{\mathsf{T}} \in \mathbb{R}^{m_d}, \text{(A67)}$$

whose dimension is:

$$m_d = \underbrace{|\boldsymbol{h}^x| \cdot m_{\tilde{n}}(2) + |\boldsymbol{h}^\theta| \cdot m_n(2)}_{m_{d_1}} + \underbrace{l_g \cdot \frac{m_{\tilde{n}}(1)\left(m_{\tilde{n}}(1)+1\right)}{2}}_{m_{d_2}} + \underbrace{\frac{m(\mathcal{B})\left(m(\mathcal{B})+1\right)}{2}}_{m_{d_3}}, \qquad \text{(A68)}$$

where $m_{d_1}$ is the dimension of the free variables $\boldsymbol{\lambda}^x$ and $\boldsymbol{\lambda}^\theta$, $m_{d_2}$ is the dimension of the PSD variables $\boldsymbol{S}_k, k = 1, \ldots, l_g$, $m_{d_3}$ is the dimension of the PSD variable $\boldsymbol{S}_0$, and we use symmetric vectorization to save storage. Then it is obvious that the dual variable $\boldsymbol{d}$ lives in a convex cone $\mathcal{K}$ defined by:

$$\mathcal{K} \doteq \mathbb{R}^{m_{d_1}} \times \underbrace{\mathcal{S}_+^{m_{\tilde{n}}(1)} \times \ldots \times \mathcal{S}_+^{m_{\tilde{n}}(1)}}_{l_g} \times \mathcal{S}_+^{m(\mathcal{B})}. \qquad \text{(A69)}$$

Additionally, the dual variable $\boldsymbol{d}$ must satisfy the equality constraint in (6):

$$f(\boldsymbol{p}) - \hat{f} - \sum_{j=1}^{|\boldsymbol{h}^x|} q_j^x - \sum_{j=1}^{|\boldsymbol{h}^\theta|} q_j^\theta = s_0 + \sum_{k=1}^{l_g} s_k, \forall \boldsymbol{p}. \qquad \text{(A70)}$$

Now using the expressions in eq. (A60), (A63), (A66), (A56), and (A59), we obtain the following linear constraints for each monomial $\boldsymbol{p}^{\boldsymbol{\alpha}}$:

$$c_f(\boldsymbol{\alpha}) = \sum_{j=1}^{|\boldsymbol{h}^x|} \left\langle \boldsymbol{\lambda}_j^x, \boldsymbol{w}_{\boldsymbol{\alpha}}^{x_j} \right\rangle + \sum_{j=1}^{|\boldsymbol{h}^\theta|} \left\langle \boldsymbol{\lambda}_j^\theta, \boldsymbol{w}_{\boldsymbol{\alpha}}^{\theta_j} \right\rangle + \sum_{k=1}^{l_g} \left\langle \text{svec}\left(\boldsymbol{S}_k\right), \text{svec}\left(\boldsymbol{W}_{\boldsymbol{\alpha}}^k\right) \right\rangle + \left\langle \text{svec}\left(\boldsymbol{S}_0\right), \text{svec}\left(\boldsymbol{W}_{\boldsymbol{\alpha}}^0\right) \right\rangle, \text{(A71)}$$

where we have used the fact that $\langle \boldsymbol{A}, \boldsymbol{B} \rangle = \langle \operatorname{svec}(\boldsymbol{A}), \operatorname{svec}(\boldsymbol{B}) \rangle$ for any two symmetric matrices $\boldsymbol{A}, \boldsymbol{B} \in \mathcal{S}^n$. The linear constraint (A71) can be written compactly as:

$$\boldsymbol{a}_{\boldsymbol{\alpha}}^{\mathsf{T}} \boldsymbol{d} = c_f(\boldsymbol{\alpha}), \tag{A72}$$

where $\boldsymbol{a}_{\boldsymbol{\alpha}} \in \mathbb{R}^{m_d}$ is a vector of constants that is only related to the equality and inequality constraints $h_j$ and $g_k$ of the POP (2):

$$\boldsymbol{a}_{\boldsymbol{\alpha}} = \left[ (\boldsymbol{w}_{\boldsymbol{\alpha}}^{x_1})^{\mathsf{T}}, \ldots, \left( \boldsymbol{w}_{\boldsymbol{\alpha}}^{x|\boldsymbol{h}^x|} \right)^{\mathsf{T}}, (\boldsymbol{w}_{\boldsymbol{\alpha}}^{\theta_1})^{\mathsf{T}}, \ldots, \left( \boldsymbol{w}_{\boldsymbol{\alpha}}^{\theta|\boldsymbol{h}^\theta|} \right)^{\mathsf{T}}, \operatorname{svec}(\boldsymbol{W}_{\boldsymbol{\alpha}}^1)^{\mathsf{T}}, \ldots, \operatorname{svec}\left( \boldsymbol{W}_{\boldsymbol{\alpha}}^{l_g} \right)^{\mathsf{T}}, \operatorname{svec}(\boldsymbol{W}_{\boldsymbol{\alpha}}^0)^{\mathsf{T}} \right]^{\mathsf{T}}. \tag{A73}$$

All the linear constraints, one for each $\boldsymbol{p}^{\boldsymbol{\alpha}}, \boldsymbol{\alpha} \in 2\mathcal{B}$, assembled together, define an affine subspace:

$$\mathcal{A} \doteq \left\{ \boldsymbol{d} : \underbrace{\begin{bmatrix} \vdots \\ \boldsymbol{a}_{\boldsymbol{\alpha}}^{\mathsf{T}} \\ \vdots \end{bmatrix}}_{\boldsymbol{A} \in \mathbb{R}^{m(2\mathcal{B}) \times m_d}} \boldsymbol{d} = \underbrace{\begin{bmatrix} \vdots \\ c_f(\boldsymbol{\alpha}) \\ \vdots \end{bmatrix}}_{\boldsymbol{b} \in \mathbb{R}^{m(2\mathcal{B})}} \right\}. \tag{A74}$$

Therefore, problem (5) is equivalent to:

$$\text{find } \boldsymbol{d} \in \mathbb{R}^{m_d}, \quad \text{s.t.} \quad \boldsymbol{d} \in \mathcal{K} \cap \mathcal{A}, \tag{A75}$$

with the convex cone $\mathcal{K}$ defined in (A69) and the affine subspace defined in (A74).

**Partial Orthogonality**. Finally, we state a property, namely *partial orthogonality* [28], of the matrix $\boldsymbol{A} \in \mathbb{R}^{m(2\mathcal{B}) \times m_d}$ that defines the affine subspace $\mathcal{A}$ in (A74).

**Theorem A9** (**Partial Orthogonality of $\boldsymbol{A}$**). *Let $\boldsymbol{A} = [\boldsymbol{A}_1, \boldsymbol{A}_2, \boldsymbol{A}_3]$ be the column-wise partition of $\boldsymbol{A}$ according to the dimension defined in (A68), i.e., $\boldsymbol{A}_1 \in \mathbb{R}^{m(2\mathcal{B}) \times m_{d_1}}, \boldsymbol{A}_2 \in \mathbb{R}^{m(2\mathcal{B}) \times m_{d_2}}$ and $\boldsymbol{A}_3 \in \mathbb{R}^{m(2\mathcal{B}) \times m_{d_3}}$, then $\boldsymbol{A}_3 \boldsymbol{A}_3^{\mathsf{T}}$ is an invertible diagonal matrix.*

*Proof.* From the partition, we know that $\boldsymbol{A}_3$ corresponds to the columns of $\boldsymbol{A}$ indexed by $\operatorname{svec}(\boldsymbol{S}_0)$. Therefore, according to (A73), which shows the row of $\boldsymbol{A}$ corresponding to a monomial $\boldsymbol{p}^{\boldsymbol{\alpha}}$, we can write $\boldsymbol{A}_3$ as:

$$\boldsymbol{A}_3 = \begin{bmatrix} \vdots \\ \operatorname{svec}(\boldsymbol{W}_{\boldsymbol{\alpha}}^0)^{\mathsf{T}} \\ \vdots \end{bmatrix}. \tag{A76}$$

Now we can compute the $(i, j)$-th entry of $\boldsymbol{A}_3 \boldsymbol{A}_3^{\mathsf{T}}$ for $i \neq j$:

$$\left[ \boldsymbol{A}_3 \boldsymbol{A}_3^{\mathsf{T}} \right]_{i,j} = \operatorname{svec}(\boldsymbol{W}_{\boldsymbol{\alpha}_i}^0)^{\mathsf{T}} \operatorname{svec}(\boldsymbol{W}_{\boldsymbol{\alpha}_j}^0) = \left\langle \boldsymbol{W}_{\boldsymbol{\alpha}_i}^0, \boldsymbol{W}_{\boldsymbol{\alpha}_j}^0 \right\rangle = 0, \tag{A77}$$

where $\left\langle \boldsymbol{W}_{\boldsymbol{\alpha}_i}^0, \boldsymbol{W}_{\boldsymbol{\alpha}_j}^0 \right\rangle = 0$ holds due to the definition of the indicator matrix in (A55) (if $\boldsymbol{\alpha}_1 + \boldsymbol{\alpha}_2 = \boldsymbol{\alpha}_i$, then $\boldsymbol{\alpha}_1 + \boldsymbol{\alpha}_2 \neq \boldsymbol{\alpha}_j$ when $\boldsymbol{\alpha}_i \neq \boldsymbol{\alpha}_j$). The diagonal entries of $\boldsymbol{A}_3 \boldsymbol{A}_3^{\mathsf{T}}$ are nonzero because:

$$\left[ \boldsymbol{A}_3 \boldsymbol{A}_3^{\mathsf{T}} \right]_{i,i} = \operatorname{svec}(\boldsymbol{W}_{\boldsymbol{\alpha}_i}^0)^{\mathsf{T}} \operatorname{svec}(\boldsymbol{W}_{\boldsymbol{\alpha}_i}^0) = \left\langle \boldsymbol{W}_{\boldsymbol{\alpha}_i}^0, \boldsymbol{W}_{\boldsymbol{\alpha}_i}^0 \right\rangle \geq 1. \tag{A78}$$

Since $\left[ \boldsymbol{A}_3 \boldsymbol{A}_3^{\mathsf{T}} \right]_{i,j} = 0$ for any $i \neq j$, and $\left[ \boldsymbol{A}_3 \boldsymbol{A}_3^{\mathsf{T}} \right]_{i,i} \geq 1$, $\boldsymbol{A}_3 \boldsymbol{A}_3^{\mathsf{T}}$ is diagonal and invertible. ∎

In Section A5, we will see the partial orthogonality property of $\boldsymbol{A}$ allows efficient computation of the projection map onto the affine subspace $\mathcal{A}$. ∎

## A5  Proof of Theorem 9 (DRS for Optimality Certification)

*Proof.* We will first prove that DRS iterates converge to a solution of the feasibility SDP (7) if it is feasible. We will then show how to compute a suboptimality bound from each iteration of the DRS update. Finally, we will discuss how to implement the projection maps in the DRS iterates.

**Convergence**. We first prove (i), *i.e.,* the DRS iterates (8), with $0 < \gamma_\tau < 2$, converge to a solution of the SDP (7) if it is feasible. To do so, we write the SDP (7) equivalently as:

$$\min_{\boldsymbol{d}} \quad \mathbb{1}_{\mathcal{K}}(\boldsymbol{d}) + \mathbb{1}_{\mathcal{A}}(\boldsymbol{d}), \tag{A79}$$

where $\mathbb{1}_{\mathcal{K}}(\boldsymbol{d})$ (resp. $\mathbb{1}_{\mathcal{A}}(\boldsymbol{d})$) is the indicator function of the set $\mathcal{K}$ (resp. the set $\mathcal{A}$), *i.e.,* $\mathbb{1}_{\mathcal{K}}(\boldsymbol{d}) = 0$ if $\boldsymbol{d} \in \mathcal{K}$ and $\mathbb{1}_{\mathcal{K}}(\boldsymbol{d}) = \infty$ if $\boldsymbol{d} \notin \mathcal{K}$. It is clear that if the SDP (7) is feasible, then the optimal cost of problem (A79) is 0; while if the SDP (7) is infeasible, then the optimal cost of problem (A79) is $\infty$. Douglas-Rachford Splitting is designed to solve problems of the following type:

$$\min_{\boldsymbol{d}} \quad f(\boldsymbol{d}) + g(\boldsymbol{d}), \tag{A80}$$

where $f$ and $g$ are convex functions of $\boldsymbol{d}$. Now let $f = \mathbb{1}_{\mathcal{K}}(\boldsymbol{d})$, $g = \mathbb{1}_{\mathcal{A}}(\boldsymbol{d})$, and observe that the proximal operator for an indicator function $\mathbb{1}_{\mathcal{K}}$ is exactly the projection onto the set $\mathcal{K}$,[6] we obtain the DRS iterates of (8) from [6, Algorithm 4.2]. In addition, [6, Proposition 4.3] tells us the DRS iterates converge to a solution of (A79). This implies the sequence $\{\boldsymbol{d}_\tau\}_{\tau \geq 0}$ converges to a point inside $\mathcal{K} \cap \mathcal{A}$ when the intersection is nonempty.[7]

**Suboptimality Bound**. We then prove (ii), *i.e.,* the DRS iterates provide valid suboptimality bounds $\bar{\varepsilon}_\tau$ at each iteration. In particular, this suboptimality bound can be computed from $\boldsymbol{d}_\tau^{\mathcal{A}}$. To show this, we note that any $\boldsymbol{d}_\tau^{\mathcal{A}}$ satisfies the equality constraint in (6) because $\boldsymbol{d}_\tau^{\mathcal{A}} \in \mathcal{A}$. Therefore, let svec $(\boldsymbol{S}_0^\tau)$ and svec $(\boldsymbol{S}_k^\tau)$, $k = 1, \ldots, l_g$ be the sub-vectors in $\boldsymbol{d}_\tau^{\mathcal{A}}$, then for any $\boldsymbol{p} \in \mathcal{P}$, eq. (6) tells us:

$$f(\boldsymbol{p}) - \hat{f} = [\boldsymbol{p}]_{\mathcal{B}}^\mathsf{T} \boldsymbol{S}_0^\tau [\boldsymbol{p}]_{\mathcal{B}}^\mathsf{T} + \sum_{k=1}^{l_g} g_k [\boldsymbol{p}]_1^\mathsf{T} \boldsymbol{S}_k^\tau [\boldsymbol{p}]_1 \geq \lambda_1(\boldsymbol{S}_0^\tau) M_0^2 + \sum_{k=1}^{l_g} \min\{0, \lambda_1(\boldsymbol{S}_k^\tau)\} M_1^2, \tag{A81}$$

where $M_0$ and $M_1$ are upper bounds on the $\ell_2$-norm of the monomial bases $[\boldsymbol{p}]_{\mathcal{B}}$ and $[\boldsymbol{p}]_1$:

$$\|[\boldsymbol{p}]_{\mathcal{B}}\| \leq M_0, \quad \|[\boldsymbol{p}]_1\| \leq M_1, \quad \forall \boldsymbol{p} \in \mathcal{P}. \tag{A82}$$

In (A81), we have used the fact that $g_k(\boldsymbol{p}) \leq 1$ for any $\boldsymbol{p} \in \mathcal{P}$ from the POP (2). Now to obtain the suboptimality bound $\bar{\varepsilon}_\tau$, let $\boldsymbol{p} = \boldsymbol{p}^\star$ be the global minimizer in (A81), we have $f(\boldsymbol{p}^\star) = f^\star$ and:

$$f^\star - \hat{f} \geq \lambda_1(\boldsymbol{S}_0^\tau) M_0^2 + \sum_{k=1}^{l_g} \min\{0, \lambda_1(\boldsymbol{S}_k^\tau)\} M_1^2 \tag{A83}$$

$$\implies \frac{\hat{f} - f^\star}{\hat{f}} \leq \frac{-\lambda_1(\boldsymbol{S}_0^\tau) M_0^2 - \sum_{k=1}^{l_g} \min\{0, \lambda_1(\boldsymbol{S}_k^\tau)\} M_1^2}{\hat{f}} := \bar{\varepsilon}_\tau. \tag{A84}$$

We now give the expressions for the upper bounds $M_0$ and $M_1$ for Examples 1-4.

**Example 1 (Single Rotation Averaging)**. Recall $\boldsymbol{x} = \boldsymbol{r} = \text{vec}(\boldsymbol{R})$ with $\|\boldsymbol{r}\|^2 = 3$, so:

$$\|[\boldsymbol{p}]_1\|^2 = 1 + \|\boldsymbol{r}\|^2 + \|\boldsymbol{\theta}\|^2 = 4 + N := M_1^2, \tag{A85}$$

$$\|[\boldsymbol{p}]_{\mathcal{B}}\|^2 = 1 + \|\boldsymbol{r}\|^2 + \|\boldsymbol{\theta}\|^2 + \|(\boldsymbol{r})_2\|^2 + \|\boldsymbol{\theta} \otimes \boldsymbol{r}\|^2 = 4N + 13 := M_0^2. \tag{A86}$$

**Example 2 (Shape Alignment)**. Recall $\boldsymbol{x} = \bar{\boldsymbol{r}} = \text{vec}(s\Pi\boldsymbol{R})$ with $\|\bar{\boldsymbol{r}}\|^2 \leq 2\bar{s}^2$, so:

$$\|[\boldsymbol{p}]_1\|^2 = 1 + \|\bar{\boldsymbol{r}}\|^2 + \|\boldsymbol{\theta}\|^2 \leq 1 + 2\bar{s}^2 + N := M_1^2, \tag{A87}$$

$$\|[\boldsymbol{p}]_{\mathcal{B}}\|^2 = 1 + \|\bar{\boldsymbol{r}}\|^2 + \|\boldsymbol{\theta}\|^2 + \|(\bar{\boldsymbol{r}})_2\|^2 + \|\boldsymbol{\theta} \otimes \boldsymbol{r}\|^2 \leq (1 + 2\bar{s}^2)(1 + N) + 4\bar{s}^4 := M_0^2. \tag{A88}$$

**Example 3 (Point Cloud Registration).** Recall $\boldsymbol{x} = [\boldsymbol{r}^\mathsf{T}, \boldsymbol{t}^\mathsf{T}]^\mathsf{T} = [\mathrm{vec}\,(\boldsymbol{R})^\mathsf{T}, \boldsymbol{t}^\mathsf{T}]^\mathsf{T}$ with $\|\boldsymbol{r}\|^2 = 3$, $\|\boldsymbol{t}\|^2 \le T^2$, so:

$$\|[\boldsymbol{p}]_1\|^2 = 1 + \|\boldsymbol{r}\|^2 + \|\boldsymbol{t}\|^2 + \|\boldsymbol{\theta}\|^2 \le 4 + T^2 + N := M_1^2, \quad \text{(A89)}$$

$$\|[\boldsymbol{p}]_\mathcal{B}\|^2 = 1 + \|\boldsymbol{r}\|^2 + \|\boldsymbol{t}\|^2 + \|\boldsymbol{\theta}\|^2 + \|(\boldsymbol{r})_2\|^2 + \|(\boldsymbol{t})_2\|^2 + \|\boldsymbol{r} \otimes \boldsymbol{t}\|^2 + \|\boldsymbol{\theta} \otimes \boldsymbol{r}\|^2 + \|\boldsymbol{\theta} \otimes \boldsymbol{t}\|^2 \quad \text{(A90)}$$

$$\le 13 + 4N + 4T^2 + T^4 + NT^2 := M_0. \quad \text{(A91)}$$

**Example 4 (Mesh Registration).** Same as point cloud registration.

**Projection Maps.** To carry out the DRS iterates (8), we need to implement the projection onto the convex cone $\mathcal{K}$, $\mathrm{proj}_\mathcal{K}$, and the projection onto the affine subspace $\mathcal{A}$, $\mathrm{proj}_\mathcal{A}$. The projection onto the PSD cone has a closed-form solution, due to Higham [11].

**Lemma A10 (Projection onto $\mathcal{S}_+^n$ [11]).** *Given any matrix $\boldsymbol{S} \in \mathcal{S}^n$, let $\boldsymbol{S} = \boldsymbol{U}\mathrm{diag}\,(\lambda_1, \ldots, \lambda_n)\,\boldsymbol{U}^\mathsf{T}$ be its spectral decomposition, then the projection of $\boldsymbol{S}$ onto the PSD cone $\mathcal{S}_+^n$ is:*

$$proj_{\mathcal{S}_+^n}(\boldsymbol{S}) = \boldsymbol{U}\mathrm{diag}\,(\max(0, \lambda_1), \ldots, \max(0, \lambda_n))\,\boldsymbol{U}^\mathsf{T}. \quad \text{(A92)}$$

Using this Lemma and the expression of the convex cone $\mathcal{K}$ in eq. (A69), the projection of $\boldsymbol{d}$ onto $\mathcal{K}$ can be performed component-wise:

$$\mathrm{proj}_\mathcal{K}(\boldsymbol{d}) = \left[\boldsymbol{\lambda}^\mathsf{T}, \mathrm{svec}\left(\mathrm{proj}_{\mathcal{S}_+^{m_{\tilde{n}}(1)}}(\boldsymbol{S}_1)\right), \ldots, \mathrm{svec}\left(\mathrm{proj}_{\mathcal{S}_+^{m_{\tilde{n}}(1)}}(\boldsymbol{S}_{l_g})\right), \mathrm{svec}\left(\mathrm{proj}_{\mathcal{S}_+^{m(\mathcal{B})}}(\boldsymbol{S}_0)\right)\right]^\mathsf{T} \quad \text{(A93)}$$

where $\boldsymbol{\lambda} \in \mathbb{R}^{m_{d_1}}$ are the unconstrained variables in $\boldsymbol{d}$ (*cf.* eq. (A67)).

For the affine subspace $\mathcal{A} = \{\boldsymbol{d} : \boldsymbol{A}\boldsymbol{d} = \boldsymbol{b}\}$, the projection onto $\mathcal{A}$ is [10]:

$$\mathrm{proj}_\mathcal{A}(\boldsymbol{d}) = \boldsymbol{d} - \boldsymbol{A}^\mathsf{T}\left(\boldsymbol{A}\boldsymbol{A}^\mathsf{T}\right)^{-1}\left(\boldsymbol{A}\boldsymbol{d} - \boldsymbol{b}\right). \quad \text{(A94)}$$

The next theorem states that the inverse $\left(\boldsymbol{A}\boldsymbol{A}^\mathsf{T}\right)^{-1}$ can be computed efficiently using the Matrix Inversion Lemma [8].

**Theorem A11 (Efficient Matrix Inversion).** *Let $\boldsymbol{A} = [\boldsymbol{A}_1, \boldsymbol{A}_2, \boldsymbol{A}_3]$ be the partition of $\boldsymbol{A}$ as in Theorem A9. Denote $\boldsymbol{A}_{12} = [\boldsymbol{A}_1, \boldsymbol{A}_2]$, and $\boldsymbol{D} = \boldsymbol{A}_3\boldsymbol{A}_3^\mathsf{T}$ as the invertible diagonal matrix. Then the inverse of $\boldsymbol{A}\boldsymbol{A}^\mathsf{T}$ is:*

$$\left(\boldsymbol{A}\boldsymbol{A}^\mathsf{T}\right)^{-1} = \boldsymbol{D}^{-1} - \boldsymbol{D}^{-1}\boldsymbol{A}_{12}\left(\mathbf{I}_{m_{d_1}+m_{d_2}} + \boldsymbol{A}_{12}^\mathsf{T}\boldsymbol{D}^{-1}\boldsymbol{A}_{12}\right)^{-1}\boldsymbol{A}_{12}^\mathsf{T}\boldsymbol{D}^{-1}. \quad \text{(A95)}$$

*Proof.* Write $\boldsymbol{A}\boldsymbol{A}^\mathsf{T} = \boldsymbol{D} + \boldsymbol{A}_{12}\boldsymbol{A}_{12}^\mathsf{T}$, and invoke the Matrix Inversion Lemma. ∎

The computational benefit brought by the partial orthogonality property of $\boldsymbol{A}$ is that, in eq. (A95), only a matrix of size $m_{d_1} + m_{d_2}$ needs to be inverted (the inversion of the diagonal matrix $\boldsymbol{D}$ is cheap), although the matrix $\boldsymbol{A}\boldsymbol{A}^\mathsf{T}$ has size $m(2\mathcal{B})$, which is typically much larger. Partial orthogonality has been exploited in several works to design scalable first-order solvers for solving SOS programs [28, 2].

Another computational advantage is, we can rewrite $\mathrm{proj}_\mathcal{A}(\boldsymbol{d})$ in eq. (A94) as:

$$\mathrm{proj}_\mathcal{A}(\boldsymbol{d}) = \left(\mathbf{I}_{m_d} - \boldsymbol{A}^\mathsf{T}\left(\boldsymbol{A}\boldsymbol{A}^\mathsf{T}\right)^{-1}\boldsymbol{A}\right)\boldsymbol{d} + \boldsymbol{A}^\mathsf{T}\left(\boldsymbol{A}\boldsymbol{A}^\mathsf{T}\right)^{-1}\boldsymbol{b}. \quad \text{(A96)}$$

Because the matrix $\boldsymbol{A}$ only depends on the constraints of the POP (2) and is unrelated to the visual measurements $\boldsymbol{y}_i$, both $\mathbf{I}_{m_d} - \boldsymbol{A}^\mathsf{T}\left(\boldsymbol{A}\boldsymbol{A}^\mathsf{T}\right)^{-1}\boldsymbol{A}$ and $\boldsymbol{A}^\mathsf{T}\left(\boldsymbol{A}\boldsymbol{A}^\mathsf{T}\right)^{-1}$ can be computed offline. Hence, during online optimality certification, only matrix-vector multiplications are required to perform the projection onto the affine subspace. ∎

## A6 Chordal Sparse Initialization

In theory, one can start DRS (8) at any initial condition $\boldsymbol{d}_0$. However, to speed up DRS, we compute the initial point $\boldsymbol{d}_0$ by solving a cheap SOS program with *chordal sparsity* [21, 14].

**Proposition A12** (**Chordal Sparse Initialization**). *Define* $[\boldsymbol{p}]_{\mathcal{B}_i} \doteq [1, \boldsymbol{x}^\mathsf{T}, \theta_i, \theta_i \boldsymbol{x}^\mathsf{T}]^\mathsf{T} \in \mathbb{R}^{2n+2}$, $[\boldsymbol{p}]_{1i} \doteq [1, \boldsymbol{x}^\mathsf{T}, \theta_i]^\mathsf{T} \in \mathbb{R}^{n+2}$, *as the sparse monomial bases only in* $\boldsymbol{x}$ *and* $\theta_i, i = 1, \ldots, N$. *Let the solution of the following SOS program (SDP):*

$$\max \qquad \zeta \in \mathbb{R} \tag{A97}$$

$$s.t. \quad f(\boldsymbol{p}) - \zeta - \sum_{h_j \in \boldsymbol{h}^x} h_j \cdot \left([\boldsymbol{p}]_2^\mathsf{T} \boldsymbol{\lambda}_j^x\right) - \sum_{h_j \in \boldsymbol{h}^\theta} h_j \cdot \left([\boldsymbol{x}]_2^\mathsf{T} \boldsymbol{\lambda}_j^\theta\right) =$$

$$\sum_{i=1}^N [\boldsymbol{p}]_{\mathcal{B}_i}^\mathsf{T} \boldsymbol{S}_{0i} [\boldsymbol{p}]_{\mathcal{B}_i} + \sum_{k=1}^{l_g} g_k \cdot \left(\sum_{i=1}^N [\boldsymbol{p}]_{1i}^\mathsf{T} \boldsymbol{S}_{ki} [\boldsymbol{p}]_{1i}\right), \forall \boldsymbol{p}, \tag{A98}$$

$$\boldsymbol{\lambda}_j^x \in \mathbb{R}^{m_{\tilde{n}}(2)}, \boldsymbol{\lambda}_j^\theta \in \mathbb{R}^{m_n(2)}, \boldsymbol{S}_{0i} \in \mathcal{S}_+^{2n+2}, \boldsymbol{S}_{ki} \in \mathcal{S}_+^{n+2}, \tag{A99}$$

*be* $\boldsymbol{\lambda}_j^{x\star}, \boldsymbol{\lambda}_j^{\theta\star}, \boldsymbol{S}_{0i}^\star$ *and* $\boldsymbol{S}_{ki}^\star$, *then* $\boldsymbol{d}_0$ *can be constructed as:*

$$\boldsymbol{d}_0 = [(\boldsymbol{\lambda}_1^{x\star})^\mathsf{T}, \ldots, (\boldsymbol{\lambda}_{|\boldsymbol{h}^x|}^{x\star})^\mathsf{T}, (\boldsymbol{\lambda}_1^{\theta\star})^\mathsf{T}, \ldots, (\boldsymbol{\lambda}_{|\boldsymbol{h}^\theta|}^{\theta\star})^\mathsf{T}, svec\left(\bar{\boldsymbol{S}}_1^\star\right)^\mathsf{T}, \ldots, svec\left(\bar{\boldsymbol{S}}_{l_g}^\star\right)^\mathsf{T}, svec\left(\bar{\boldsymbol{S}}_0^\star\right)^\mathsf{T}]^\mathsf{T} \tag{A100}$$

*where* $\bar{\boldsymbol{S}}_k^\star \in \mathcal{S}^{\mathbb{R}^{m_{\tilde{n}}(1)}}, k = 1, \ldots, l_g$, *and* $\bar{\boldsymbol{S}}_0^\star \in \mathcal{S}^{m(\mathcal{B})}$ *satisfy:*

$$[\boldsymbol{p}]_1^\mathsf{T} \bar{\boldsymbol{S}}_k^\star [\boldsymbol{p}]_1 = \sum_{i=1}^N [\boldsymbol{p}]_{1i}^\mathsf{T} \boldsymbol{S}_{ki}^\star [\boldsymbol{p}]_{1i}, \forall \boldsymbol{p}, \tag{A101}$$

$$[\boldsymbol{p}]_{\mathcal{B}}^\mathsf{T} \bar{\boldsymbol{S}}_0^\star [\boldsymbol{p}]_{\mathcal{B}} = \sum_{i=1}^N [\boldsymbol{p}]_{\mathcal{B}_i}^\mathsf{T} \boldsymbol{S}_{0i}^\star [\boldsymbol{p}]_{\mathcal{B}_i}, \forall \boldsymbol{p}. \tag{A102}$$

The chordal sparse SDP (A97) is different from the SDP (5) in two aspects. First, we have relaxed the large PSD constraints into multiple much smaller PSD constraints with fixed sizes (independent of the number of measurements $N$). For example, $\boldsymbol{S}_0 \in \mathcal{S}_+^{m(\mathcal{B})}$ has been divided into $\boldsymbol{S}_{0i} \in \mathcal{S}_+^{2n+2}, i = 1, \ldots, N$, where each $\boldsymbol{S}_{0i}$ is associated with a sparse monomial basis $[\boldsymbol{p}]_{\mathcal{B}_i}$ of fixed size. Similarly, each $\boldsymbol{S}_k \in \mathcal{S}_+^{m_{\tilde{n}}(1)}$ has been divided into $N$ smaller PSD constraints $\boldsymbol{S}_{ki} \in \mathcal{S}_+^{n+2}, i = 1, \ldots, N$. Second, instead of trying to certify $\hat{f}$ is the global minimum of $f(\boldsymbol{p})$, we turn to maximize a lower bound $\zeta$ of $f(\boldsymbol{p})$. The reason is, by relaxing the large PSD constraints into multiple smaller constraints (*i.e.,* by requiring the SOS polynomials in the feasibility SDP (5) to admit chordal sparse decompositions as in (A101) and (A102)), problem (A97) is more restrictive than problem (5) and in general its optimum $\zeta^\star$ cannot certify the global optimality of $\hat{f}$ (*i.e.,* $\zeta^\star < p_{\mathcal{B}}^\star \leq f^\star \leq \hat{f}$).[8] However, the chordal sparse SOS program (A97) scales to large $N$. Therefore, we compute $\boldsymbol{d}_0$ by solving this cheap SOS program (A97) using an IPM-based SDP solver and then refine $\boldsymbol{d}_0$ by running DRS for the more powerful (but more expensive) SOS program (5).

## A7 Details of Experiments

### A7.1 Details of Experimental Setup

We test primal relaxation and dual certification on *random* problem instances of Examples 1-4. At each Monte Carlo run, we generate inliers and outliers as follows. In single rotation averaging (SRA), we first randomly generate a ground-truth 3D rotation $\boldsymbol{R}^\circ$, then inliers are generated by $\boldsymbol{R}_{\text{in}} = \boldsymbol{R}^\circ \boldsymbol{R}_\epsilon$, where $\boldsymbol{R}_\epsilon$ is generated by randomly sampling a unit-norm rotation axis $\boldsymbol{\Psi} \in \mathbb{R}^3$ and a rotation angle $\phi \sim \mathcal{N}(0, \sigma^2)$ with $\sigma = 3°$; outliers are arbitrary random rotations. In shape alignment (SA), we first randomly generate a 3D shape $\{\boldsymbol{B}_i\}_{i=1}^N$, where each $\boldsymbol{B}_i \sim \mathcal{N}(\boldsymbol{0}, \mathbf{I}_3)$, and then scale the shape such that its diameter (*i.e.,* maximum distance between two points) is 4. We then generate a random ground-truth scale $s^\circ \in [0.5, 2]$ and a random ground-truth rotation $\boldsymbol{R}^\circ$. Inlier 2D measurements are generated by $\boldsymbol{b}_{\text{in}} = s^\circ \Pi \boldsymbol{R}^\circ \boldsymbol{B} + \epsilon$, where $\epsilon \sim \mathcal{N}(\boldsymbol{0}, \sigma^2 \mathbf{I}_2)$ with $\sigma = 0.01$, and outliers are arbitrary

2D vectors $\boldsymbol{b}_{\text{out}} \sim \mathcal{N}(\boldsymbol{0}, \mathbf{I}_2)$. In point cloud registration (PCR), we first generate $\{\boldsymbol{a}_i\}_{i=1}^N$ in the same way as generating $\{\boldsymbol{B}_i\}_{i=1}^N$ in SA. Then we sample a random rotation $\boldsymbol{R}^\circ$ and a random translation $\boldsymbol{t}^\circ$ with $\|\boldsymbol{t}^\circ\| \le T = 1$. Inlier 3D points are generated by $\boldsymbol{b}_{\text{in}} = \boldsymbol{R}^\circ \boldsymbol{a} + \boldsymbol{t}^\circ + \boldsymbol{\epsilon}$, where $\boldsymbol{\epsilon} \sim \mathcal{N}(\boldsymbol{0}, \sigma^2 \mathbf{I}_3)$ with $\sigma = 0.01$; and outliers are arbitrary random vectors $\boldsymbol{b}_{\text{out}} \in \mathcal{N}(\boldsymbol{0}, \mathbf{I}_3)$. In mesh registration (MR), we first generate a random mesh by sampling unit normals $\{\boldsymbol{u}_i\}_{i=1}^N$ and points $\{\boldsymbol{a}_i\}_{i=1}^N$ the same way as in SA. Then we generate a random rotation $\boldsymbol{R}^\circ$ and translation $\boldsymbol{t}^\circ, \|\boldsymbol{t}^\circ\| \le T = 1$. Inlier normals are generated by $\boldsymbol{v}_{\text{in}} = (\boldsymbol{R}^\circ \boldsymbol{u} + \boldsymbol{\epsilon})/\|\boldsymbol{R}^\circ \boldsymbol{u} + \boldsymbol{\epsilon}\|$, where $\boldsymbol{\epsilon} \sim \mathcal{N}(\boldsymbol{0}, \sigma^2 \mathbf{I}_3)$ with $\sigma = 0.01$. Inlier points are generated by $\boldsymbol{b}_{\text{in}} = \boldsymbol{R}^\circ(\boldsymbol{a} + \boldsymbol{u} \times \boldsymbol{\Phi}) + \boldsymbol{t}^\circ + \boldsymbol{\epsilon}$, where $\boldsymbol{\Phi} \sim \mathcal{N}(\boldsymbol{0}, \mathbf{I}_3)$ and $\boldsymbol{\epsilon} \sim \mathcal{N}(\boldsymbol{0}, \sigma^2 \mathbf{I}_3)$ with $\sigma = 0.01$ (note that $\boldsymbol{a} + \boldsymbol{u} \times \boldsymbol{\Phi}$ generates a random point on the face defined by $(\boldsymbol{a}, \boldsymbol{u})$). Outlier normals are randomly generated unit-norm 3D vectors $\boldsymbol{v}_{\text{out}}$ and outlier points are randomly generated $\boldsymbol{b}_{\text{out}} \sim \mathcal{N}(\boldsymbol{0}, \mathbf{I}_3)$. The relative weight between point-to-plane distance and normal-to-normal distance is set to be $w_i = 1, i = 1, \ldots, N$. In problem (TLS), $\bar{c} = 1$ for all applications, and $\beta_i^2 = \beta^2, i = 1, \ldots, N$, is set as follows. In SRA, $\beta^2 = (2\sqrt{2}\sin(3\sigma/2))^2$. In SA, $\beta^2 = \sigma^2 \cdot \text{chi2inv}(2, 0.99)$. In PCR, $\beta^2 = \sigma^2 \cdot \text{chi2inv}(3, 0.99)$. In MR, $\beta^2 = 2\sigma^2 \cdot \text{chi2inv}(3, 0.99)$, where $\text{chi2inv}(d, p)$ computes the quantile of the $\chi^2$ distribution with $d$ degrees of freedom and lower tail probability equal to $p$ (see [24] for a probabilistic interpretation).

### A7.2 Dense vs. Sparse Moment Relaxation

We compare the performance of the dense moment relaxation (3) and the sparse moment relaxation (4) with $N = 10$ measurements, because the dense relaxation is too large to be solved by IPM solvers at $N = 20$. Fig. A1 boxplots the rotation estimation error (left axis) and the relative duality gap (right axis) averaged over 30 Monte Carlo runs for the four Examples 1-4. For single rotation averaging (Fig. A1(a)), both the dense and spare relaxations are tight up to $80\%$ outlier measurements (relative duality gap always below $10^{-5}$), and both of them return accurate rotation estimations (rotation error always below 5 degrees). For shape alignment and point cloud registration (Fig. A1(b)(c)), both the dense and sparse relaxations produce occasional non-tight solutions (especially at high-outlier regime). However, we see that the rotation estimations are still quite accurate. We observed that the relaxation becomes tighter for increasing $N$. Indeed, the results in the paper shows improved performance for $N = 20$. Hence, we conjecture that, when $N$ is small, the estimation problem is more "ambiguous" for the relaxations, in the sense that inliers do not form a dominating consensus set as strong as when $N$ is large. This is similar to human perception: we recognize the patterns more easily when we see dense visual measurements (*e.g.*, a dense point cloud vs. a sparse point cloud of only a few points). For mesh registration (Fig. A1(d)), the relaxations are always tight, and significantly better than the case of point cloud registration. This echoes our previous conjecture: adding surface normals to the visual measurements provides more cues and makes the estimation less "ambiguous". Finally, it is also interesting to see that at $80\%$ outlier rate (there are only 2 inliers), there are two runs where the relaxations produce the globally optimal solutions (because the relative duality gap is below $10^{-5}$), but the globally optimal solutions are far away from the ground-truth solutions (the rotation errors are 90 and 180 degrees). We suspect the reason is the possible symmetry in the randomly generated problems, as also observed in [26].

### A7.3 Results for Point Cloud Registration

Fig. A2 shows the performance of primal relaxation and dual certification on point cloud registration, and the results look qualitatively the same as the results for mesh registration in the main text.

### A7.4 Details of Satellite Pose Estimation

The neural network in [5] learns a 3D model of the Tango satellite consisting of 11 keypoints $\{\boldsymbol{B}_i\}_{i=1}^{11}$, shown in Fig. A3(a). It can also output 11 2D landmark detections for a given 2D image, $\{\boldsymbol{b}_i\}_{i=1}^{11}$, shown in Fig. A3(b). We assume a weak perspective camera model[9] and the inlier 3D keypoints and 2D landmarks satisfy the following generative model:

$$\boldsymbol{b}_i = s\boldsymbol{\Pi}\boldsymbol{R}\boldsymbol{B}_i + \boldsymbol{t} + \boldsymbol{\epsilon}_i, \tag{A103}$$

Figure A1: Dense momemt relaxation vs. sparse moment relaxation on (a) Single Rotation Averaging, (b) Shape Alignment, (c) Point Cloud Registration, and (d) Mesh Registration. Left axis: rotation estimation error; right axis: relative duality gap. $N = 10$ and statistics are plotted over 30 Monte Carlo runs.

Figure A2: Certifiable point cloud registration. (a) Sparse moment Relaxation, (b) Dual optimality certification and (c) Convergence of suboptimality.

where $t \in \mathbb{R}^2$ is a 2D translation and $\epsilon_i$ models an unknown but bounded additive noise that satisfies $\|\epsilon_i\| \le \delta_i$. Then the *pairwise relative* 3D keypoints and 2D landmarks will satisfy the shape alignment model used in Example 2:

$$\underbrace{b_i - b_j}_{\bar{b}_{ij}} = s\Pi R \underbrace{(B_i - B_j)}_{\bar{B}_{ij}} + \underbrace{(\epsilon_i - \epsilon_j)}_{\bar{\epsilon}_{ij}}, \tag{A104}$$

because the translation $t$ cancels out due to the subtraction, and $\|\bar{\epsilon}_{ij}\| \le \delta_i + \delta_j$ models the updated noise. Because there are 11 keypoints and landmarks, we have $K = \binom{11}{2} = 55$ pairwise measurements $\{\bar{B}_k\}_{k=1}^{K}$ and $\{\bar{b}_k\}_{k=1}^{K}$. Using the $K$ pairwise measurements, we can first estimate $s$ and $R$ using the certifiable algorithms discussed in the main text, and then estimate the translation using the adaptive voting method in [23]. The full 6D camera pose can be recovered as:

$$R^{3D} = R, \quad t^{3D} = \frac{\left[t^{\mathsf{T}}, 1\right]^{\mathsf{T}}}{s}. \tag{A105}$$

When spoiling outliers, we replace $l$ landmarks $\boldsymbol{b}_i$'s with random 2D pixels, which implies that the outlier rate should be computed as:

$$1 - \frac{\binom{11-l}{2}}{55}, \tag{A106}$$

where $\binom{11-l}{2}$ is the number of inlier pairwise relative measurements (a pairwise measurement $\bar{\boldsymbol{b}}_{ij}$ is an inlier if and only if both $\boldsymbol{b}_i$ and $\boldsymbol{b}_j$ are inliers). Using the formula in eq. (A106), the outlier rates are $0\%$, $18\%$, $35\%$, $49\%$, $62\%$ and $73\%$ for $l = 0, 1, 2, 3, 4, 5$.

Extra results and visualizations are provided in Fig. A3. These results were certified as correct by the dual optimality certifiers presented in the main text.

(a) 3D wireframe model of Tango

(b) $l = 3$ (49% outlier rate)

(c) $l = 4$ (62% outlier rate)

(d) $l = 4$ (62% outlier rate)

(e) $l = 5$ (73% outlier rate)

(f) $l = 5$ (73% outlier rate)

Figure A3: Satellite pose estimation on the SPEED dataset [19].

## A7.5 Comparison to Primal Baselines

Fig. A4(a) compares the performance of our primal solver (SDP: Basis Reduction) versus two state-of-the-art baselines: (1) GNC (best heuristics, no optimality guarantees) [27] and (2) SDP: Chordal Sparse (an efficient SDP relaxation that exploits correlative sparsity) [21] on single rotation averaging. Our primal relaxation is significantly tighter than chordal sparse relaxation, and the accuracy and robustness of our estimates dominate both baselines.

Figure A4: (a) Comparison to primal baselines. (b) Comparison to certification baselines. (c) Adversarial outliers.

## A7.6 Comparison to Certification Baselines

Our DRS approach is the first mathematically rigorous approach for verifying solution correctness. We compare it with a heuristic method that performs Kolmogorov–Smirnov (KS) test [17] on the squared residuals with a $\chi^2$ distribution (i.e., tests normality of the residuals classified as inliers). Fig. A4(b) shows that KS test has many false positives/negatives, while ours has zero, for single rotation averaging.

## A7.7 Adversarial Outliers

We performed tests with an adversarial outlier model (where outliers follow a different model and are consistent with each other) and test our algorithm (SDP: Basis Reduction) against two state-of-the-art baselines. Fig. A4(c) shows our method dominates both baselines, is insensitive to adversarial outliers until the maximum breakdown point 50%. Note that our relaxation is still tight at 50% outlier rate, certifying that the globally optimal solution is obtained. However, due to the presence of adversarial outliers, the globally optimal solution may not be the ground-truth solution (if we assume the ground-truth solution has a larger set of consistent measurements).

## Footnotes

[1]The same set of quadratic constraints hold when $\boldsymbol{r}_i^\mathsf{T} \in \mathbb{R}^3, i = 1, \ldots, 3$ denotes the $i$-th row of $\boldsymbol{R}$.

[2]Note that due to the division by $2\bar{s}^2$ in eq. (A6), $0 \leq g^{\bar{r}} \leq 1$ is satisfied.

[3]Because the constraint polynomials $\boldsymbol{h}$ and $\boldsymbol{g}$ have degree 2, the localizing matrices of order $\kappa - 1$ is used to make sure every moment appearing in the localizing matrices also appears in the moment matrix $\boldsymbol{M}_\kappa(\boldsymbol{z}_{2\kappa})$. In a more general setting where the constraint polynomials $h_i$ (or $g_i$) have degree $2v_i$ or $2v_i - 1$, then the localizing matrices of degree $\kappa - v_i$ should be used.

[4]In the general case, suppose $\mathcal{P}$ are defined by polynomials with degree $2v_i$ or $2v_i - 1, i = 1, \ldots, l_h + l_g$, then denote $v = \max_i v_i$, the sufficient condition becomes $\mathrm{rank}\left(\boldsymbol{M}_{\kappa-v}\left(\boldsymbol{z}_{2\kappa-2v}\right)\right) = \mathrm{rank}\left(\boldsymbol{M}_\kappa\left(\boldsymbol{z}_{2\kappa}\right)\right)$.

[5]The uniqueness of the solution comes from the fact that Interior Point Methods solvers (*e.g.,* SeDuMi) output an optimal solution of *maximum rank* if the SDP has more than one optimal solutions [22]. Therefore, if $\boldsymbol{p}^\star$ is not unique, then the SDP will have multiple optimal solutions and the rank of the solution will be larger than 1 (*cf.* Theorem 6.18 in [15]).

[6]The proximal operator of a function $f$ is defined as: $\text{prox}_f(\boldsymbol{x}_0) \doteq \arg\min_{\boldsymbol{x}} \frac{1}{2}\|\boldsymbol{x} - \boldsymbol{x}_0\|^2 + f(\boldsymbol{x})$. When $f = \mathbb{1}_{\mathcal{K}}$ is an indicator function, then $\text{prox}_f(\boldsymbol{x}_0) = \arg\min_{\boldsymbol{x}} \frac{1}{2}\|\boldsymbol{x} - \boldsymbol{x}_0\|^2 + \mathbb{1}_{\mathcal{K}}(\boldsymbol{x}) := \text{proj}_{\mathcal{K}}(\boldsymbol{x}_0)$.

[7]In fact, more generally, even if the intersection is empty, it is known that, if $\gamma_\tau = 1$, then the sequences $\{\boldsymbol{d}_\tau^{\mathcal{K}}\}_{\tau \geq 0}$ and $\{\boldsymbol{d}_\tau^{\mathcal{A}}\}_{\tau \geq 0}$ converge to a solution of the optimization: $\min_{\boldsymbol{d}_1 \in \mathcal{K}, \boldsymbol{d}_2 \in \mathcal{A}} \|\boldsymbol{d}_1 - \boldsymbol{d}_2\|$, *i.e.,* a pair of points $\bar{\boldsymbol{d}}_1 \in \mathcal{K}$ and $\boldsymbol{d}_2 \in \mathcal{A}$ that achieves the minimum distance between set $\mathcal{K}$ and set $\mathcal{A}$ [12, 1].

[8]From a different perspective, $\boldsymbol{S}_k \succeq 0$ and $\boldsymbol{S}_0 \succeq 0$ imply that there must exist smaller PSD decompositions $\boldsymbol{S}_{ki} \succeq 0$ and $\boldsymbol{S}_{0i} \succeq 0$. However, $\boldsymbol{S}_{ki} \succeq 0$ and $\boldsymbol{S}_{0i} \succeq 0$ do not necessarily mean $\boldsymbol{S}_k \succeq 0$ and $\boldsymbol{S}_0 \succeq 0$.

[9]Weak perspective camera model is a good approximation of the full perspective camera model when the object is far away from the camera center [29, 25].