[Reviews · NeurIPS 2020]

Review 1

Summary and Contributions: The paper studies SDP relaxations for robust geometric perception in the presence of outliers. They use the standard truncated loss formulation, and provide an SDP relaxation via the Lassere hierarchy. While this SDP formulation isn't computationally tractable, they provide an alternate dual formulation and Douglas Radford splitting algorithm to solve the dual problem and obtain a certificate in an efficient manner.

Strengths: The paper address both the issue of certifiably optimal solution to the geometric perception problem with outliers, but also provides a computationally efficient algorithm to do so. As I am not very familiar with this exact area, I cannot comment on the significance of the problem. Empirical evaluation is restricted to mostly synthetic settings (simulations) for showing the generality of approach. From a quick study on other papers cited in related work in this space of geometric perception, it seems like this is the standard. The first two components: Lassere based SDP relaxation and sparse moment relaxation seem to have been used in prior work as well (Yang and Carlone CVPR 2020) and hence do not seem novel. While the DR splitting based algorithm seems like a standard application of known techniques, it could be the first application to this setting. Since I am unfamiliar with all related work in this space, I am not sure about this.

Weaknesses: My main concern is that the paper does not contextualize the related work appropriately in a self-contained manner. It is unclear to the reader what the main technical challenges are, what are the novel insights that are used to solve the problems, and why previous techniques fail or are insufficient. The theorem statements are somewhat abstract and a bit hard to follow due to heave notation. While this could be inherent to the approach, from the current exposition, it seems like the paper mainly uses standard results in optimization literature. Some discussion on whether it is just a straightforward application, or whether there is some more general technical novelty that could be useful independent of this exact application setting would be useful. However, overall the paper was well-written and guides the reader through the different components in the method. The empirical evaluation could be improved: for e.g., what are relevant baselines? What are computationally efficient (but maybe not certifiably optimal) methods and how does this work compare to those? This is helpful to ground the current results. Otherwise, it's unclear what to make of the numbers/results. Is the sophisticated machinery even necessary? It also seems like more "real" datasets could be employed (beyond SPEED), but unfortunately I am not familiar with this space to provide concrete suggestions.

Correctness: While i haven't read the proofs in detail, the theorem statements seem correct and empirical methodology is sound (upto insufficient baselines).

Clarity: The paper is overall well-written but the technical results seem a bit opaque (refer weaknesses above).

Relation to Prior Work: As a person not familiar with this space, I found it hard to understand how this work differs from existing work from a technical perspective See discussion in strengths and weaknesses above.

Reproducibility: Yes

Additional Feedback:


Review 2

Summary and Contributions: This paper studies the truncated least squares (TLS) cost function with an application to robust geometric perception. Its main contributions are - Exhibiting that for a large class of geometric perception problems, the TLS objective can be written as a low-dimensional optimization problem. - Exploit the problem structure (sparsity in constraints) to reduce effective size of SDP to be solved. - Provide a dual certification algorithm based on Douglas-Rachford splitting which can cmpute sub-optimality gap of a proposed solution. Update (Post-Rebuttal): I would like to thank the authors for addressing my concerns in their rebuttal. While the paper overall provides a nice unifying analysis for robust perception problems, there are a few things worth highlighting which can be improved with the current draft: (i) The draft should highlight the novelty of the presented results and compare it with existing literature. (ii) The draft right now is a bit dense and it would benefit from adding in more discussions around the main presented results as well as the formulations. I think that the presentation could be improved if the authors mention whether the results follow directly from known techniques or if there were certain technical challenges in obtaining these. (iii) As mentioned by the other reviewers, I think the draft could benefit by adding more detailed experiments showcasing the advantages of this framework. Having said this, I still liked the overall contribution and am inclined to keep my rating the same.

Strengths: This work provides a principled view to approach robust geometric perception problem (specific problem instances of total least squares) through the lens of polynomial optimization and sums-of-squares relaxations. The theoretical claims provided in the paper build on existing literature on convex relaxations via SDPs and the connections with perception problem seems novel and interesting.

Weaknesses: I would like to highlight a few questions/remarks regarding the results: 1. The paper considers the truncated least squares cost where errors saturate after some level \bar{c}. A similar robust estimator, by the name of Least Trimmed Squares (LTS) proposed in [1] only minimizes the error on a subset of the data points, and sets the contribution of the largest residual points to zero. Can the authors comment if the proposed techniques could indeed be extended to handle such classes of estimators? 2. Line 131-132 mentions that the TLS is known to be robust to outliers. Could the authors mention precisely what are the breakdown points and under what kind of assumptions? 3. The proposed framework seems to run only in the "small N (O(100))" regime. Can the authors comment if there are ways of scaling up these primal relaxations (even approximately at the cost of optimality). Also, what are ranges of N which are often used by practitioners for the class of perception problems discussed here? 4. Comparison with existing baselines: In the experiments section, it would be helpful to compare the proposed optimization methods with existing heuristics in terms of performance accuracy, time taken for completion as well as memory requirements on all 5 tasks studied. 5. Outlier generation for experiments: The current set of experiments work with random perturbations independent of the parameter x. It would be useful to look at the performance of these methods with different adversarial perturbations (say generated by another model trained to attack the proposed model). The random perturbations for the outliers studied here seem a bit restrictive. [1] Rousseeuw PJ. Least median of squares regression. Journal of the American statistical association. 1984.

Correctness: Both the theoretical claims and the experimental setup look correct. The experiments, though, could be improved with further additions (see points above).

Clarity: Overall, I found the paper a bit dense to read but this can be attributed to a page limit restriction. The paper is quite well-motivated and has a nice logical flow to it with the problem formulation followed by the primal-dual analysis.

Relation to Prior Work: The paper does a good job of linking its contributions to existing works in the literature and providing relevant references for material which had to be skipped from the paper.

Reproducibility: Yes

Additional Feedback:


Review 3

Summary and Contributions: The paper presents an optimization framework for robust geometric perception problems which contain large amounts of outliers. The authors show that the considered problem class with truncated least squares can be posed as an optimization over the ring of polynomials. Further contributions include further polynomial basis reduction to reduce the problem size and optimality certifiers. The generality of the method is demonstrated by solving four different problem instances, namely: single rotation averaging, shape alignment, mesh registration, and satellite pose estimation.

Strengths: - The paper introduces a novel mathematical well-founded optimization framework. - The content is very relevant to the community since effective outlier estimation is a central problem in many practical perception problems.

Weaknesses: - Although the papers’ focus is on the theory, the experimental evaluation could be stronger by comparing to established, as well as state-of-the-art baseline methods on outlier filtering, e.g. RANSAC, M-estimation methods or specialized global solvers like branch-and-bound-based consensus maximization. - The paper is very densely written and difficult to follow.

Correctness: The proofs for all theorems and proofs have been presented in the supplementary material. The proposed theory seems to be sound and well-founded.

Clarity: The paper is well written and structured.

Relation to Prior Work: The paper is very comprehensive regarding citations of related works and provides a nicely partitioned overview. Nevertheless, the discussion of related work focuses on mentioning algorithm properties, but lacks discussing differences and similarities to the proposed approach.

Reproducibility: Yes

Additional Feedback:


Review 4

Summary and Contributions: This paper presents a general certifiable global optimal framework for solving geometric model fitting problem under the presence of outliers. It formulates the task as optimization over polynomial rings defined by the Lasserre's hierarchy of convex relaxations. Both basis reduction and SOS based techniques and analyses are presented in the paper.

Strengths: The geometric perception tasks addressed in this paper is very relevant to both computer vision and optimization communities. This paper represents a systematic treatment of multiple different geometric problem under the influences of outliers. The proposed certifiable optimal solutions are valid and useful.

Weaknesses: While paper can be used a useful reference, one drawback is over the (lack of) theoretic originality of the techniques used. Lasserre's hierarchical convex relaxation is a well known technique in various fields, including numerical computation, optimal control (see numerous papers by D. Henrion ), and computer vision (see ICCV 2005 paper by F. Kahl and Henrion). So has been the case for sparse SDP basis reduction, as well as for SoS (sum of squares). The claimed novelty of unifying multiple problems under the the same POP formulation ( i.e one rings to rule them all) seems to be a slight over-statement of the contribution, as this is a straightforward consequence of plugging the TLS loss function to Lasserre's moment based relaxation pipeline. In addition, the TLS loss is only one of the choices for robust estimators, and a slightly restrictive choice. My second concern is the rather weak experiment section (which imo weaker than sufficient). Many of the simulated tests (monte carlo) are somewhat artificial, failing to validate either the claims that were made in proceeding sections, or demonstrate the benefit of having a certified optimal solution. In other words, I see there is a clear mismatch between the (over) claims and the experiment validation.

Correctness: Seems correct to me, though have not checked the proofs completely.

Clarity: Yes, the paper is well written.

Relation to Prior Work: The work is built upon a large amount of prior works, and it presents itself as a good summary, though it falls short in bringing in original contribution other than a carefully executed review and compilation.

Reproducibility: Yes

Additional Feedback: Main drawbacks: (1) what is really new in this paper ? (2) more convincing experiments.

[Author Response · NeurIPS 2020]



**(a) Comparison to baselines for solving primal TLS problem. Top: rotation error, bottom: duality gap.**

**(b) Comparison to baselines for verifying solution correctness. Top: DRS (ours), bottom: KS test.**

**(c) Performance w.r.t. increasing adversarial outliers. Top: rotation error, bottom: duality gap.**

**Novelty and Significance (R1, R4).** (i) **Motivation:** *Outliers* are ubiquitous in computer vision and robotics [101]. (ii) **Hardness:** *Globally optimal* outlier-robust geometric perception is NP-hard and intractable [28]. (iii) **Limitation of SOTA:** SOTA algorithms are divided into *fast heuristics* (real-time but no guarantees) and *global solvers* (optimal but exponential time). (iv) **Significance of this paper:** (1) *Theoretical:* The *first polynomial-time* method for solving generic outlier-robust geometric perception with *a posteriori* global optimality guarantees. The tightness of the relaxation discovers a class of non-convex problems that admits *hidden convexity*, which has significant potential for theoretical study.[1] (2) *Algorithmic:* The *first* tractable method for designing *dual optimality certifiers* that run orders of magnitude faster than SOTA SDP solvers (*cf.* Table 1).[2] (3) **Broader Impact:** Optimality guarantees (*e.g.,* rejecting wrong solutions as in Fig. 1(b)) are crucial for safety-critical applications. (v) **Novelties:** (1) First to reformulate TLS as POP with structured sparsity (*cf.* Prop. 5(i)(ii)); (2) First to empirically show Lasserre's hierarchy is (surprisingly) tight for *outlier-robust* problems (Kahl'07 IJCV, Yang'20 CVPR assume outlier-free) with *binary variables* (vs. MAX-CUT relaxation is not tight); (3) First to use *basis reduction* to improve efficiency but keep tightness (vs. chordal relaxation [91] is not tight (Fig. (a))). (4) First to propose scalable certifiers using DRS and chordal initialization.

**Comparison with Baselines (R1-4).** (i) Fig. (a) compares the performance of our primal solver (SDP: Basis Reduction) versus two (SOTA) baselines, GNC (best heuristics) [97] and SDP: Chordal Sparse (efficient SDP relaxation) [91]. Our primal relaxation is significantly tighter than chordal sparse relaxation [91], and the accuracy and robustness of our estimates dominates both baselines. (ii) Our DRS approach is the *first* mathematically rigorous approach for verifying solution correctness. We compare it with a heuristic method that performs Kolmogorov–Smirnov (KS) test on the squared residuals with a $\chi^2$ distribution (*i.e.,* tests normality of the residuals classified as inliers). Fig. (b) shows that KS test has many false positives/negatives, while ours has zero. *All results are for Single Rotation Averaging, similar results hold for the other applications considered in the paper.*

**Adversarial Outliers (R2).** We performed tests with an adversarial outlier model (where outliers follow a different model and are consistent with each other) and test our algorithm (SDP: Basis Reduction) against two SOTA baselines. Fig. (c) shows our method dominates both baselines, is insensitive to adversarial outliers until the maximum breakdown point $50\%$ (the tightness of the relaxation implies the optimal solution may not be the ground-truth solution at $50\%$).

**LTS (R2).** LTS minimizes sum of the $K$ smallest squared residuals: $\min_{\boldsymbol{x}\in\mathcal{X}} \sum_{i=1}^{K} r_i^2(\boldsymbol{x})$, with $r_1^2 \leq \cdots \leq r_N^2$. This is equivalent to: $\min_{\boldsymbol{x}\in\mathcal{X},\theta_i^2=1} \sum_{i=1}^{N} \frac{1+\theta_i}{2} r_i^2(\boldsymbol{x})$, subject to: $\sum_{i=1}^{N} \theta_i = 2K - N$, which ensures the number of $\theta_i$'s with value $+1$ is exactly $K$ (smallest $K$). Therefore, LTS can be written as a POP, and our framework can be applied. However, tightness of performing relaxation for LTS is not guaranteed and basis reduction may need extra care.

**Others.** (i) Theoretical breakdown of TLS is $50\%$. Empirical robustness can be higher if outliers are not adversarial ([99], over $95\%$). [101] establishes breakdown for a specific problem. (ii) $N = 100$ is common for real problems. But surely there is still room for scalability improvements (*e.g.,* BM factorization [18]). (iii) Main paper is rigorous thus hard to follow. Supplementary provides details for non-expert readers, and we will open source our implementation.

## Footnotes

[1]Moreover, the tightness of the relaxations further motivates developing fast SDP solvers, which is a major line of research on its own (many related work in NeurIPS). As SDP solvers become more efficient, these problems eventually can be solved in real-time.

[2]The goal of this paper is NOT to *replace* existing fast heuristics, but to *enhance* them with a fast *certification* that allows asserting the quality of their estimates and rejecting failure cases (*cf.* Fig. 1(b)), for safety-critical applications.


[Meta-Review · NeurIPS 2020]

The paper proposes an SDP based framework for studying outlier robust optimization problems. Reviewers believe that the ideas in the paper are interesting and the results are novel. But the paper requires significant rewriting to ensure that novelty of the approach vis-a-vis existing works is clear and the implications of the results are also clearly spelled out.